# When Data-Free Knowledge Distillation Meets Non-Transferable Teacher: Escaping Out-of-Distribution Trap is All You Need

**Ziming Hong** [1]  **Runnan Chen** [1]  **Zengmao Wang** [2]  **Bo Han** [3]  **Bo Du** [2]  **Tongliang Liu** [1]

## Abstract

Data-free knowledge distillation (DFKD) transfers knowledge from a teacher to a student without access the real in-distribution (ID) data. Its common solution is to use a generator to synthesize fake data and use them as a substitute for real ID data. However, existing works typically assume teachers are trustworthy, leaving the robustness and security of DFKD from untrusted teachers largely unexplored. In this work, we conduct the first investigation into distilling non-transferable learning (NTL) teachers using DFKD, where the transferability from an ID domain to an out-of-distribution (OOD) domain is prohibited. We find that NTL teachers fool DFKD through divert the generator's attention from the useful ID knowledge to the misleading OOD knowledge. This hinders ID knowledge transfer but prioritizes OOD knowledge transfer. To mitigate this issue, we propose Adversarial Trap Escaping (`ATEsc`) to benefit DFKD by identifying and filtering out OOD-like synthetic samples. Specifically, inspired by the evidence that NTL teachers show stronger adversarial robustness on OOD samples than ID samples, we split synthetic samples into two groups according to their robustness. The fragile group is treated as ID-like data and used for normal knowledge distillation, while the robust group is seen as OOD-like data and utilized for forgetting OOD knowledge. Extensive experiments demonstrate the effectiveness of `ATEsc` for improving DFKD against NTL teachers. Code is released at https://github.com/tmllab/2025_ICML_ATEsc.

## 1. Introduction

Recently, data-free knowledge distillation (DFKD) (Micaelli & Storkey, 2019; Luo et al., 2020; Fang et al., 2021b; Yu et al., 2023; Tran et al., 2024) has attracted appealing attention as it deals with distilling valuable knowledge from a well-trained teacher model $f$ to a student model $S$ *without requiring to access the real in-distribution (ID) training data*. As shown in Figure 1 (a), since the real ID data $\mathcal{D}_{id}$ is unavailable, a common solution is to introduce a generator $G$ to synthesize fake samples via an *adversarial exploration* way. These synthetic samples are then used as proxies to guide the knowledge transfer from teacher to student (Micaelli & Storkey, 2019; Fang et al., 2019). More specifically, $G$ and $S$ are alternatively optimized during the training procedure: (i) the $G$ is trained to synthesize fake samples $\mathcal{D}_s$ causing disagreement between student $S$ and teacher $f$, thus expanding the coverage of the data distribution; (ii) the $S$ is trained to reach teacher-student agreement on samples synthesized by $G$, thus learning teacher's knowledge. As a result, the synthetic data $\mathcal{D}_s$ is similar to the real ID domain $\mathcal{D}_{id}$, and the student $S$ obtains teacher's knowledge by reaching similar accuracy on real ID data $\mathcal{D}_{id}$.

Although significant progress has been made in this field, existing works always assume that pre-trained teachers are trustworthy. Recently, Hong et al. (2023) conducted the first study to explore DFKD in the context of untrusted teachers. However, their work primarily focuses on teachers with backdoor (Wu et al., 2022; Gu et al., 2019). The robustness and security of DFKD from untrusted teachers beyond backdoor remain largely unexplored. In this paper, we conduct the first investigation on the situation of *DFKD against non-transferable learning (NTL) teachers* (Wang et al., 2022). Specifically, in NTL teacher's training, transferability from an ID domain $\mathcal{D}_{id}$ to an out-of-distribution (OOD) domain[1] $\mathcal{D}_{ood}$ is deliberately restricted by incorporating an additional regularization term into the standard ID-domain supervised learning (SL) objective, where the regularization term encourages the teacher's output to differ between the two domains.

---

[1]Sydney AI Centre, The University of Sydney [2]School of Computer Science, Wuhan University [3]Department of Computer Science, Hong Kong Baptist University. Correspondence to: Zengmao Wang <wangzengmao@whu.edu.cn>, Tongliang Liu <tongliang.liu@sydney.edu.au>.

*Proceedings of the $42^{nd}$ International Conference on Machine Learning*, Vancouver, Canada. PMLR 267, 2025. Copyright 2025 by the author(s).

---

[1]The $\mathcal{D}_{ood}$ can be both close-set (i.e., share the same label space (Bengio et al., 2011; Tzeng et al., 2017)) and open-set (i.e., disjoint label space (Liang et al., 2017; Wei et al., 2022)) to the $\mathcal{D}_{id}$.

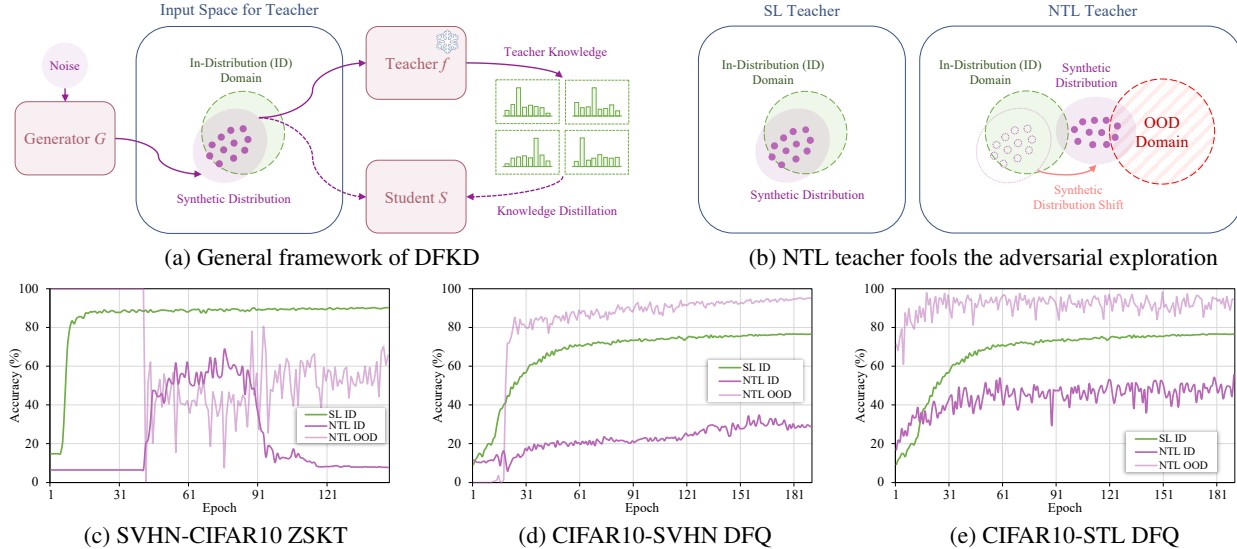

(a) General framework of DFKD

(b) NTL teacher fools the adversarial exploration

(c) SVHN-CIFAR10 ZSKT

(d) CIFAR10-SVHN DFQ

(e) CIFAR10-STL DFQ

*Figure 1.* **(a)** The adversarial exploration framework of data-free knowledge distillation (DFKD). **(b)** For the SL teacher pre-trained on ID domain $\mathcal{D}_{id}$, the synthetic distirbution $\mathcal{D}_s$ is close to the real $\mathcal{D}_{id}$. However, for the NTL teacher pre-trained on both $\mathcal{D}_{id}$ and out-of-distribution data $\mathcal{D}_{ood}$, *OOD trap effect* leads to the shift of $\mathcal{D}_s$ from $\mathcal{D}_{id}$ toward $\mathcal{D}_{ood}$. **(c-e)** A comparison of training dynamics of distilling SL and NTL teachers with DFKD baselines. We present three metrics for student: ID accuracy in distilling a SL teacher (SL-ID) and an NTL teacher (NTL-ID), and OOD accuracy[2] in distilling an NTL teacher (NTL-OOD). More results are shown in Appendix C.

Through empirical investigations (Section 3), we find NTL teachers fool the training of both generator and student via a commonly raised ***OOD trap effect***. As illustrated in Figure 1 (b), this effect means that when distilling NTL teachers, misleading knowledge from the OOD domain replaces the role of ID domain knowledge, acting as a *trap* that diverts the DFKD's primary attention. Two key phenomenon, as shown in Figure 1 (c-d), are directly linked to this effect: **(i)** *degradation of ID knowledge transfer*: students exhibit lower NTL-ID accuracies compared to distillation from SL teachers, and **(ii)** *misleading OOD knowledge transfer*: students inherit teachers' OOD knowledge, reaching high NTL-OOD accuracies[2]. Furthermore, we identify two intrinsic reasons that contribute to the *OOD trap effect*: **(i)** *ID-to-OOD synthetic distribution shift*: the synthetic distribution $\mathcal{D}_s$ shifts from the ID domain $\mathcal{D}_{id}$ toward the OOD domain $\mathcal{D}_{ood}$; **(ii)** *ID-OOD learning task conflicts*: the learning targets of student on ID-like and OOD-like synthetic samples have significant differences. These factors together become the primary reasons why NTL teachers can fool DFKD, leading to misguided student learning.

To mitigate the OOD trap effect imposed by NTL teachers, we propose Adversarial Trap Escaping (ATEsc), a novel plug-and-play approach to benefit DFKD by identifying and filtering out misleading OOD-like synthetic samples according to their adversarial robustness. ATEsc is inspired from the perspective that the adversarial vulnerability of a data

point is closely related to its margin, i.e., the distance from the data point to the decision boundary (Xu et al., 2023; Zhu et al., 2021a). By analyzing the margin of the ID and OOD domains, we verify that an NTL teacher exhibits a significant difference in adversarial vulnerability between the two domains, demonstrating strong robustness in the OOD domain while remaining significantly more vulnerable in the ID domain[3]. Accordingly, ATEsc split synthetic samples into a fragile group and a robust group based on their adversarial robustness against untargeted projected gradient descent (PGD) attack (Madry, 2017). Then, ATEsc only uses the fragile samples (regarded as ID-like) to guide the student in learning teacher's knowledge. Therefore, the student's input distribution is close to the real ID domain, and the learning targets consist solely of ID knowledge, thus effectively alleviating the negative impact from *ID-to-OOD synthetic distribution shift* and *ID-OOD learning task conflicts*. Furthermore, OOD-like robust samples, which demonstrate high resilience to adversarial perturbations, are used as negative examples. The student's outputs on these samples are optimized to be distinct from the teacher's outputs, thus further suppressing the transfer of misleading OOD knowledge from teacher to student.

Empirically, we conduct experiments on NTL teachers across three OOD domain configurations (close-set, open-set, and backdoor-trigger) on eight dataset pairs, three DFKD baseline methods, and multiple architecture combinations. The results demonstrate that ATEsc consistently

---

[2]NTL-OOD accuracy is calculated between the prediction and the specific OOD domain label $y_{ood}$ (see Section 2.1). A higher NTL-OOD indicates more misleading OOD knowledge transfer.

[3]Further discussion and verification are shown in Section 4.1.

helps different DFKD baselines escape the OOD trap, significantly enhancing the student's ID performance while effectively suppressing the transfer of misleading OOD knowledge. Our main contributions are organized as three-fold:

- For the first time, we identify the commonly occurred *OOD trap effect* in distilling NTL teachers by DFKD (exists across tasks, architectures, and DFKD methods). We further identify its intrinsic causes: *ID-to-OOD synthetic distribution shift* and *ID-OOD learning task conflicts*.

- We propose a novel plug-and-play approach ATEsc to enhance DFKD in transferring ID knowledge from NTL teachers. ATEsc mitigates the OOD trap effect via harnessing synthetic samples based on the difference of NTL teachers' adversarial robustness on ID and OOD domains.

- We validate the effectiveness of ATEsc via extensive experiments across different OOD domain configurations, datasets, network architectures, and DFKD baselines.

## 2. Preliminary

### 2.1. Non-Transferable Learning

The original Non-Transferable Learning (NTL) (Wang et al., 2022) was proposed to restrict a model's transferability from an in-distribution (ID) domain to a *close-set* out-of-distribution (OOD) domain (i.e., share the same label space with the ID domain (Bengio et al., 2011; Ganin & Lempitsky, 2015; Tzeng et al., 2017; Huang et al., 2023a;b; 2024)) by maximizing the discrepancy between the representations of the two domains. Here, we slightly expand the original NTL to a more general purpose. Specifically, we consider an ID domain $\mathcal{D}_{\text{id}} = \{(\boldsymbol{x}_i, y_i)\}_{i=1}^{N_{\text{id}}}$ and an OOD domain $\mathcal{D}_{\text{ood}} = \{(\boldsymbol{x}_i)\}_{i=1}^{N_{\text{ood}}}$. The ID domain is labeled and expected to be the utility domain of the NTL model. The OOD domain can be unlabeled and applicable to both *close-set* and *open-set* (i.e., disjoint label space) (Liang et al., 2017; Liu et al., 2020; Wei et al., 2022) to the ID domain. The learning objectives on ID and OOD domains are shown as follows:

**Learning on ID domain.** Considering a neural network $f : \mathcal{X} \to \mathcal{Y}$. In general, the $f$ is the combination of a feature extractor $f_e$ and a classifier $f_c$, i.e., $f(\boldsymbol{x}) = f_c(f_e(\boldsymbol{x}))$. The learning objective on the ID domain is to minimize the loss:

$$\mathcal{L}_{\text{id}} = \mathbb{E}_{(\boldsymbol{x},y) \sim \mathcal{D}_{\text{id}}} \left[ D_{\text{KL}}(f(\boldsymbol{x}), y) \right], \tag{1}$$

where $D_{\text{KL}}$ is the Kullback-Leibler (KL) divergence.

**Learning on OOD domain.** The main idea of NTL is to maximize the discrepancy of both the output logits and feature representations between ID and OOD domain data. For the output level, NTL directly maximizes the KL divergence between the output logits of the two domains, and thus, the loss $\mathcal{L}_{\text{out}}$ can be formulated as:

$$\mathcal{L}_{\text{out}} = \mathbb{E}_{\boldsymbol{x}_u \sim \mathcal{D}_{\text{id}}, \boldsymbol{x}_o \sim \mathcal{D}_{\text{ood}}} \left[ D_{\text{KL}}(f(\boldsymbol{x}_o), f(\boldsymbol{x}_u)) \right], \tag{2}$$

For the feature level, NTL introduces Maximum Mean Discrepancy (MMD) to measure the distance between features from two domains, which can be formulated as:

$$\mathcal{L}_{\text{feat}} = \mathbb{E}_{\boldsymbol{x}_u \sim \mathcal{D}_{\text{id}}, \boldsymbol{x}_o \sim \mathcal{D}_{\text{ood}}} \left[ \text{MMD}(f_e(\boldsymbol{x}_o), f_e(\boldsymbol{x}_u)) \right]. \tag{3}$$

The output-level and feature-level terms are combined by multiplying and then used as a regularization on the ID domain learning objective. Therefore, the complete NTL objective $\mathcal{L}_{\text{NTL}}$ is derived as:

$$\mathcal{L}_{\text{NTL}} = \mathcal{L}_{\text{id}} - \min(1, \alpha \cdot \mathcal{L}_{\text{out}} \cdot \mathcal{L}_{\text{feat}}), \tag{4}$$

where $\alpha$ is a trade-off weight, and $\min(1, \cdot)$ upper bound the term $\alpha \cdot \mathcal{L}_{\text{out}} \cdot \mathcal{L}_{\text{feat}}$ by 1 for training stability. By minimizing Equation 4, the $f$ will still perform as normal supervised learning (SL) model on the ID domain. In contrast, OOD domain data are mapped by $f$ into a distinct cluster that lies far from the ID domain in both the output and intermediate feature spaces (Wang et al., 2022; Hong et al., 2024a), resulting in *ID-to-OOD non-transferability*.

**Class controlling of OOD data.** In fact, the OOD cluster in the output space is often predicted as a single class (Wang et al., 2022); however, this class is not fixed and can vary due to random factors. However, controlling the class assignment of OOD data is crucial in scenarios where the accuracy of OOD predictions for a specific label must be assessed. A notable example is poison-based backdoor attack and defense (Li et al., 2022; Hong et al., 2023). Therefore, an additional constraint can be introduced to enforce OOD data predictions toward a designated OOD class:

$$\mathcal{L}_{\text{cls}} = \mathbb{E}_{\boldsymbol{x} \sim \mathcal{D}_{\text{ood}}} \left[ \mathcal{L}_{\text{CE}}(f(\boldsymbol{x}), y_{\text{ood}}) \right], \tag{5}$$

where $\mathcal{L}_{\text{CE}}$ is the cross-entropy loss, and $y_{\text{ood}}$ is the designated label for the OOD-class. By using $\mathcal{L}_{\text{cls}}$ as a regularization, we get the total loss of class-specific NTL[4]:

$$\mathcal{L}_{\text{NTL-cls}} = \mathcal{L}_{\text{NTL}} + \lambda_{\text{cls}} \cdot \mathcal{L}_{\text{cls}}, \tag{6}$$

where $\lambda_{\text{cls}}$ is hyper-parameter to balance the weight.

### 2.2. Data-Free Knowledge Distillation

The goal of data-free knowledge distillation (DFKD) is to transfer a teacher's knowledge to a student without using any real samples drawn from the teacher's ID domain. Its adversarial exploration framework is shown in Figure 1 (a), where we have a generator $G$ (with parameters $\theta_G$), a student $S$ (with parameters $\theta_S$), and the targeted teacher $f$.

The generator $G$ and the student $S$ will be optimized alternately with the teacher's parameters being frozen. Specifically, the generator $G$ is optimized to cause disagreement between student $S$ and teacher $f$, thus expanding the coverage of the synthetic distribution (i.e., adversarial exploration):

$$\max_{\theta_G} \left\{ \mathcal{L}_{\text{syn}} = \mathbb{E}_{\boldsymbol{n} \sim \mathcal{D}_{\text{noise}}} \left[ D_{\text{KL}}(S(G(\boldsymbol{n})), f(G(\boldsymbol{n}))) \right] \right\}, \tag{7}$$

---

[4]The algorithm for training NTL is shown in Appendix B.1.

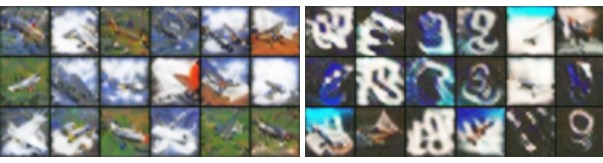

(a) SL on CIFAR10      (b) NTL on CIFAR10→Digits

*Figure 2.* Synthetic samples of CMI on SL and NTL teacher.

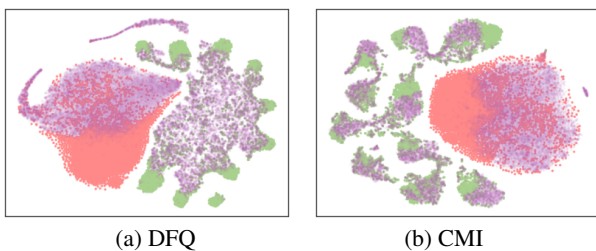

(a) DFQ      (b) CMI

*Figure 3.* Feature distribution (NTL: SVHN→CIFAR10). Green dots and red dots represent real ID and OOD domain samples, respectively. Purple transparent dots represent the synthetic samples.

where $D_{\mathrm{KL}}$ is the KL divergence to measure the distribution difference, $\boldsymbol{n}$ represents a noise vector sampled from a predefined noise distribution $\mathcal{D}_{\mathrm{noise}}$. Then, the synthesized samples are leveraged to teach the student $S$ to mimic the teacher's outputs, thus obtaining teacher's knowledge:

$$\min_{\theta_S} \left\{ \mathcal{L}_{\mathrm{kd}} = \mathbb{E}_{\boldsymbol{n} \sim \mathcal{D}_{\mathrm{noise}}} \left[ D_{\mathrm{KL}}(S(G(\boldsymbol{n})), f(G(\boldsymbol{n}))) \right] \right\}. \quad (8)$$

In addition to the basic adversarial exploration losses, state-of-the-art (SOTA) DFKD methods introduce different diversity strategy (Choi et al., 2020; Fang et al., 2021b; Yu et al., 2023; Fang et al., 2021b; Patel et al., 2023; Tran et al., 2024) for improving synthesizing quality and training stability. An essential and common regularization used in SOTA DFKD methods (Choi et al., 2020; Fang et al., 2021b; Tran et al., 2024) is the Batch-Normalization (BN) loss, which was first proposed in (Yin et al., 2020). The BN loss in DFKD is usually represented as the KL divergence between the feature statistics of current inputs and the stored BN statistics:

$$\mathcal{L}_{\mathrm{bn}} = \sum_l D_{\mathrm{KL}}(\mathcal{N}(\mu_l(\{G(\boldsymbol{n})\}), \sigma_l^2(\{G(\boldsymbol{n})\})), \\ \mathcal{N}(\mu_l, \sigma_l^2)), \quad (9)$$

where $\mathcal{N}(\mu_l(\{G(\boldsymbol{n})\}), \sigma_l^2(\{G(\boldsymbol{n})\}))$ is the $l$-th layer's feature statistics of synthetic samples $\{G(\boldsymbol{n})\}$, and $\mathcal{N}(\mu_l, \sigma_l^2)$ is the BN statistics for $l$-th layer. Minimizing the BN loss in the generator $G$'s training stage makes the synthetic samples follow similar statistical information of teacher's training samples, thus improving the synthetic sample quality.

The complete algorithm for training DFKD baseline with BN loss is shown in Appendix B.2. More related works and techniques of DFKD are illustrated in Appendix A.

## 3. When DFKD Meets NTL Teachers

In this section, we investigate the impact of NTL teachers on DFKD. We analyze four DFKD baselines: the basic adversarial exploration ZSKT (Micaelli & Storkey, 2019) and its enhanced version DFQ (Choi et al., 2020) with sample diversity loss and BN loss, and two SOTA methods with an additional memory bank: CMI (Fang et al., 2021b) and NAYER (Tran et al., 2024). Experiments are conducted on both close- and open-set dataset pairs: SVHN (Netzer et al., 2011), CIFAR10 (Krizhevsky et al., 2009), STL10 (Coates et al., 2011), Digits (Wang et al., 2022). Typical results are shown in Figure 1 (c-e), where we plot the training dynamic to compare distilling SL and NTL teachers. Specifically, the extent of knowledge transfer on ID and OOD domain during distillation are evaluated by the ID domain accuracy to its real labels and the OOD domain accuracy corresponding to the OOD-class label in Equation 5, respectively. More results refer to Appendices C and D.

### 3.1. NTL Teachers Fool DFKD via OOD Trap Effect

We find that the intrinsic *ID-to-OOD non-transferability* in NTL teachers generally fools the training of both generator $G$ and student $S$ in DFKD, with key phenomenons in Figure 1 (c-e) directly related to this:

- *Degradation of ID knowledge transfer*: the student exhibits lower NTL-ID accuracy compared to distillation an SL teacher, along with more unstable training dynamics.

- *Misleading OOD knowledge transfer*: the student inherits teacher's misleading OOD knowledge, resulting in a high NTL-OOD accuracy.

We refer to the combination of the above two phenomena as the **OOD trap effect** imposed by NTL teachers on DFKD. Intuitively, as illustrated in Figure 1 (b), misleading knowledge from the OOD domain replaces the role of ID domain knowledge when distilling NTL teachers, acting as a trap that diverts DFKD's primary attention. In addition, we conduct visualization in both *image space* and NTL teacher's *feature space* to show the impact of OOD trap effect for DFKD. **(i)** *In synthetic image space*, we visualize the synthetic images by distilling a SL teacher and an NTL teacher using DFKD. As shown in Figure 2, SL results resemble real CIFAR10 samples (see Figure 9), while NTL results integrate both the ID domain (CIFAR10) and OOD domain (Digits) semantics. **(ii)** *In feature space*, as shown in Figure 3, we plot the distribution of the NTL teacher's features for synthetic samples, real ID samples, and real OOD samples using t-SNE visualization (Van der Maaten & Hinton, 2008). We find that parts of synthetic samples's features overlap with real OOD domain features, which means that these samples are more similar to real OOD samples and can activate NTL teacher's OOD misleading knowledge.

### 3.2. Why Does OOD Trap Effect Occur

We argue that the OOD trap effect is primarily caused by two underlying reasons: **(i)** *ID-to-OOD synthetic distribution shift* and **(ii)** *ID-OOD learning task conflicts*.

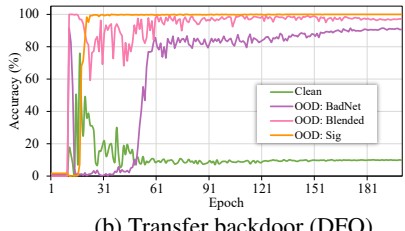
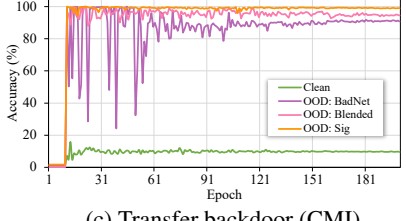

(a) Defend against DFME       (b) Transfer backdoor (DFQ)       (c) Transfer backdoor (CMI)

*Figure 4.* Downstream applications of NTL's OOD trap effect to DFKD: **(a)** NTL teachers defend against data-free model extraction (DFME); **(b-c)** NTL teachers transfer backdoor to students via data-free knowledge distillation.

**ID-to-OOD synthetic distribution shift** *means the synthetic distribution $\mathcal{D}_s$ will occur shift from the ID domain $\mathcal{D}_{id}$ toward the OOD domain $\mathcal{D}_{ood}$*, resulting in mix ID and OOD characteristics of synthetic samples. This is caused by the joint effect of the adversarial exploration loss and the BN loss when training the generator $G$ in DFKD. Specifically:

- *The BN loss (Equation 9) lets the synthetic data statistics to follow the statistics of NTL teacher's training data.* When training NTL teachers, each batch contains a mixture of ID and OOD samples. This results in the fact that BN layers in well-trained NTL teachers record statistical information for the mixture distribution of real ID and OOD domains. As such, when conducting DFKD, the BN loss constraints the generator $G$ to synthesize data which follows the statistics of NTL teacher's training data (i.e., the mixture statistics of real ID and OOD data).

- *The adversarial exploration loss (Equation 7) lets the generator $G$ to maximize the discrepancy between student and NTL teacher.* Accordingly, even if we assume that the generator $G$ only synthesize ID-like samples and the student $S$ only has ID domain knowledge in some stage, the generator $G$ will still be trained to synthesize OOD-like data. This is because in such situation, the student's output distribution on ID data $P(S(\mathcal{D}_{id}))$ are similar to teacher's output distribution on ID data $P(f(\mathcal{D}_{id}))$. However, when training the NTL teacher $f$, the discrepancies between its outputs on ID and OOD domain were intentionally maximized (Equation 2). Therefore, the current student's outputs on the OOD domain $P(S(\mathcal{D}_{ood}))$ will also have a large discrepancy with the teacher outputs on the OOD domain $P(f(\mathcal{D}_{ood}))$. As such, we have $D_{KL}(S(\mathcal{D}_{id}), f(\mathcal{D}_{id})) < D_{KL}(S(\mathcal{D}_{ood}), f(\mathcal{D}_{ood}))$. Combining with the constraint from the BN loss, the $G$ will be optimized toward synthesizing OOD-domain-like samples to satisfy the maximization of the distribution discrepancy between the student $S$ and the NTL teacher $f$.

**ID-OOD learning task conflicts** *means the significant difference of the learning targets on ID-like and OOD-like synthetic samples when distillation.* Due to the ID-to-OOD shift of synthetic distributions as discussed in above, the synthetic distribution can be seen as intermediate distribution and containing merge information (i.e., both ID-like and OOD-like) of the two domains. Recalling the NTL pretraining objective

in Equation 2, the term of $\max D_{KL}(f(\boldsymbol{x}_o), f(\boldsymbol{x}_u))$ leads to the significant difference of teacher's outputs on ID data and OOD data. Thus, in distillation stage (Equation 8), the student's learning targets on ID- and OOD-like synthetic samples have a huge difference. This introduce a severe task conflict (Misra et al., 2016; Yu et al., 2020; Liu et al., 2021a; Yadav et al., 2024; Yang et al., 2024) in the knowledge distillation stage, degrading the student's ID performance.

In summary, the generator's *ID-to-OOD synthetic distribution shift* results in the OOD knowledge transfer in DFKD. This causes the student's *ID-OOD learning task conflicts* and results in the degradation of ID knowledge transfer[5].

### 3.3. Application of OOD Trap Effect

Furthermore, we show both benign and malign downstream applications of NTL teachers' OOD trap effect for DFKD.

**Benign: defend against data-free model extraction.** Data-free model extraction (DFME) (Kariyappa et al., 2021; Truong et al., 2021) seeks to replicate a model's functionality without access to its original training data, posing a threat to its intellectual property (Wang et al., 2024c). We show that a model owner can effectively defend against DFME if their model is trained by NTL. As shown in Figure 4 (a), we train NTL models with SVHN as ID domain and consider both close-set (Digits, MNIST) and open-set (CIFAR10) OOD domains. Compared to the undefended SL model, NTL models effectively degrade the ID knowledge transfer, with even nearly no model functions being stolen.

**Malign: transfer backdoor via data-free knowledge distillation.** A backdoor (Wu et al., 2022) pre-implanted in a model can alter its behavior when triggered by inputs with pre-designed triggers, posing a high risk of malicious manipulation. Our experiments in Figure 4 (b-c) provide the evidence that backdoor pre-implanted by NTL (i.e., clean samples as the ID domain and clean sample with triggers as the OOD domain) can be transferred to student via DFKD due to the mechanism of NTL's OOD trap effect, with the

---

[5]It is notably that without the BN loss, the generator $G$ does not need to synthesize samples that resemble those from the OOD domain. Unfortunately, their ID performance will be influenced by low quality of synthetic samples, limiting the useability of the learned student. More analysis are shown in Appendix C.4.

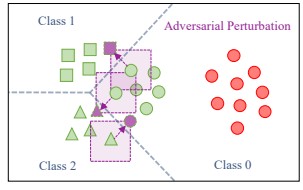 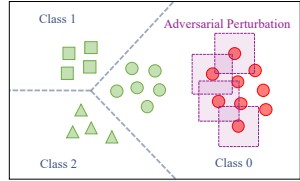

(a) ID domain         (b) OOD domain

*Figure 5.* We use green and red denotes ID and OOD domain data. (a): The decision boundaries cannot separate the $l_p$-balls around the data points of ID domain. (b) The $l_p$-balls around the OOD data points are far away from the decision boundaries.

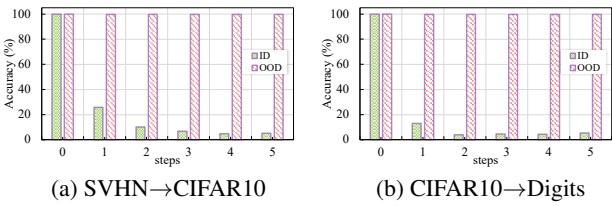

(a) SVHN→CIFAR10      (b) CIFAR10→Digits

*Figure 6.* The difference of adversarial robustness (PGD with $\epsilon = 8$) of the ID and OOD domain under different attack steps.

students inheriting high attack success rate (ASR). Notably, the risk of backdoor transfer to students via DFKD was first observed in ABD (Hong et al., 2023), although their study focused on conventional backdoor teachers instead of NTL-based ones. In Appendix E.5, we demonstrate that their defense strategy struggles to mitigate backdoor transfer when the backdoored teacher is NTL-based.

## 4. Adversarial Trap Escaping

To mitigate the OOD trap effect in NTL teachers, in this section, we propose Adversarial Trap Escaping (ATEsc), which serves as a plug-and-play module for DFKD. In Section 4.1, we identify the vulnerability difference of ID and OOD domain samples against adversarial perturbation. Accordingly, in Section 4.2, we propose to split synthetic samples into two groups via their robustness against untargeted projected gradient descent attack. Then, in Section 4.3 and Section 4.4, we illustrate how will ATEsc treats the fragile group to conduct knowledge distillation and use the robust group for OOD misleading knowledge forgetting. The overall pipeline of ATEsc is presented in Section 4.5.

### 4.1. Adversarial Robustness Difference

We start from the premise that a data point's adversarial vulnerability is closely linked to its **margin**, i.e., its distance to the decision boundary (Xu et al., 2023; Zhu et al., 2021a). *We argue that NTL teachers exhibit different vulnerability to adversarial perturbation on ID and OOD domain samples.*

For the ID domain, the training objective (Equation 1) of NTL teachers are identical to that of standard SL models (i.e., learning correct classification), suggesting that NTL teachers have complex decision boundaries that lie within the ID domain, as illustrated in Figure 5 (a). This implies

that small adversarial perturbations applied to ID samples can easily generate adversarial examples with predictive labels being different from their original labels, i.e., the fragile adversarial robustness of NTL teachers on ID domain.

For the OOD domain, recalling the previously illustration in Section 2.1, NTL achieves *ID-to-OOD non-transferability* by maximizing the distance between the features and predictions of OOD data and those of ID data (Equations 2 and 3). As such, OOD domain data are mapped into a distinct cluster that lies extremely far from the ID domain, resulting in a large distance from the decision boundaries as well. Accordingly, small adversarial perturbations on OOD data may still not exceed the decision boundaries of their original class, as illustrated in Figure 5 (b). Therefore, NTL teachers demonstrate strong adversarial robustness on OOD samples compared to ID samples.

To empirically verify the adversarial robustness difference on ID and OOD domains, we conduct experiments across different datasets and architectures to compare the robustness on each domain. Typical results are shown in Figure 6, where the accuracy reflects the *consistency of predictive labels before and after untargeted adversarial attack*. It can be observed that predictive labels for ID domain data are easily changed after attacks, which aligns with the conclusions of existing adversarial attack studies for SL models (Madry, 2017; Huang et al., 2017). In contrast, OOD samples exhibit strong robustness against adversarial perturbations even after multiple steps and higher perturbation bound.

Additional validations of the adversarial robustness of NTL teachers are provided in Appendix C.5. Furthermore, in Appendices C.6 and C.7, we analyze the impact of NTL training objectives on adversarial robustness.

### 4.2. Synthetic Data Grouping

Motivated by the adversarial robustness difference, we propose Adversarial Trap Escaping (dubbed as ATEsc) to benefit DFKD in distilling NTL teachers. ATEsc intervenes in DFKD after generator's optimization in each epoch. Specifically, ATEsc first splits synthetic samples into a fragile group and a robust group according to NTL teacher's adversarial robustness on these samples. Formally, to assess the robustness of the NTL teacher $f$ against synthetic samples $\mathcal{D}_s = \{\boldsymbol{x}_i = G(\boldsymbol{n}_i)\}_{i=1}^B$, ATEsc conducts untargeted projected gradient descent (PGD) attack (Madry, 2017) on each sample to find the worst-case perturbation that can disrupt its original predictions. Untargeted PGD is formulated as:

$$\hat{\boldsymbol{x}}_i = \Pi_{\mathcal{B}_\epsilon^p(\boldsymbol{x}_i)}(\boldsymbol{x}_i + \alpha \cdot \text{sign}(\nabla_{\boldsymbol{x}_i}\mathcal{L}_{\text{CE}}(f(\boldsymbol{x}_i), \tilde{y}_i))), \quad (10)$$

where $\hat{\boldsymbol{x}}_i$ denotes the perturbed variant of the original sample $\boldsymbol{x}_i$ with the original predictive label $\tilde{y}_i$. $\mathcal{B}_\epsilon^p(\boldsymbol{x}_i) = \{\boldsymbol{x}_i : \|\boldsymbol{x} - \boldsymbol{x}_i\|_p \leq \epsilon\}$ and $\epsilon$ is the perturbation bound. $\Pi$ represents the projection operation.

Based on the observation that perturbations easily push ID

data points beyond their original decision boundaries, the synthetic data $x_i$ is considered ID-domain-like if the predictive label of the attack result $\hat{x}_i$ changes. Conversely, samples whose predictive labels remain unchanged are regarded as OOD-like. Accordingly, each batch of synthetic data $\mathcal{D}_s$ is split into two groups: an ID-domain-like fragile group $G_f$ and an OOD-domain-like robust group $G_r$:

$$x_i \in \begin{cases} G_f, & \text{if } \arg\max f(\hat{x}_i) \neq \tilde{y}_i, \\ G_r, & \text{if } \arg\max f(\hat{x}_i) = \tilde{y}_i. \end{cases} \quad (11)$$

### 4.3. Calibrated Knowledge Distillation

After the synthetic data grouping, we propose to only use the *fragile samples* (i.e., $G_f$) to teach the student $S$ learning teacher's knowledge. Accordingly, the learning objective of DFKD's knowledge distillation stage can be calibrated by revising Equation 8 as:

$$\mathcal{L}_{\text{ckd}} = \mathbb{E}_{x \sim \mathcal{D}_{G_f}} \left[ D_{\text{KL}}(S(x), f(x)) \right]. \quad (12)$$

Since fragile samples are considered ID-like, the teacher's outputs on these samples are more likely to contain valuable ID-domain knowledge. As such, learning from the teacher only on $G_f$ helps mitigate the task conflicts caused by ID-OOD discrepancies, thus enhancing ID knowledge transfer.

### 4.4. Misleading Knowledge Forgetting

Although calibrated knowledge distillation can filter out OOD-domain-like samples, some intermediate samples inevitably exist. These samples are not robust but can still activate the teacher's misleading OOD knowledge, enabling a certain degree of OOD knowledge transfer from NTL teacher to student. This may not fully align with expectations in certain scenarios (such as defending backdoor transfer from NTL teachers to DFKD students (Hong et al., 2023)). To further suppress misleading OOD knowledge transfer, we use *robust samples* (i.e., $G_r$), which exhibit extremely high resilience to adversarial perturbations, as negative references. The student's outputs on these samples are optimized to be distinct from the NTL teacher's outputs. Formally, we use KL divergence to measure the distance between student's and teacher's outputs:

$$\mathcal{L}_{\text{forget}} = \mathbb{E}_{x \sim \mathcal{D}_{G_r}} \left[ D_{\text{KL}}(S(x), f(x)) \right]. \quad (13)$$

By maximizing $\mathcal{L}_{\text{forget}}$, the student's outputs on OOD-like robust samples are encouraged to diverge from NTL teacher's OOD outputs $f(x)$, thereby further reducing the transfer of misleading knowledge from the OOD domain.

### 4.5. Overall Pipeline

We obtain the complete ATEsc loss by combining both the calibrated knowledge distillation and the forgetting term:

$$\mathcal{L}_{\text{ATEsc}} = \mathcal{L}_{\text{ckd}} - \min(1, \lambda \cdot \mathcal{L}_{\text{forget}}), \quad (14)$$

where $\lambda$ control the weight of the forget loss, $\min(1, \cdot)$ set an upper bound for stable training of maximization term.

---

**Algorithm 1** DFKD with ATEsc
1: **Input:** an NTL teacher $f$; a generator $G$ with parameters $\theta_G$; a student $S$ with parameters $\theta_S$; total training epochs $E$; synthetic batchsize $B$.
2: **Output:** the DFKD student $S$.
3: **for** $e = 1$ to $E$ **do**
4:     ▷ *Synthesizing*
5:     Updating $\theta_G$ via Equation 7 and Equation 9;
6:     ▷ *Synthetic Data Grouping*
7:     Synthesizing fake samples $\mathcal{B} = \{G(n_i)\}_{i=1}^{B}$ from noise;
8:     Obtain perturbed fake samples $\hat{\mathcal{B}}$ via Equation 10;
9:     Clustering $\mathcal{B}$ into $G_f$ and $G_r$ by Equation 11;
10:    ▷ *Calibrated KD and Forgetting*
11:    Updating $\theta_S$ via Equation 14;
12: **end for**

---

The proposed ATEsc serves as a plug-and-play module for various DFKD baselines, inserted between the synthesizing and distillation stages in each training epoch. Specifically, after generator's training, ATEsc identifies fragile and robust sub-groups from the current synthetic batch samples. Then, the two sub-group samples are used for distillation by revising the original KD loss (see Equation 8) into ATEsc loss (Equation 14). We summarize the pipeline of DFKD with ATEsc in Algorithm 1.

## 5. Experiments

In this section, we empirically validate the effectiveness of ATEsc. We first introduce our experimental setup. Then we present results on NTL teachers across three tasks: closed-set NTL, open-set NTL, and NTL-based backdoor.

**Datasets.** We use the common datasets shared by both NTL and DFKD, including Digits (MNIST (Deng, 2012), USPS (Hull, 1994), SVHN (Netzer et al., 2011), MNIST-M (Ganin et al., 2016), and SYN-D (Roy et al., 2018)), CIFAR10 (Krizhevsky et al., 2009), STL (Coates et al., 2011).

**NTL tasks.** We conduct experiments on NTL teachers trained with three ID/OOD domain configurations:

- **(i) Close-set NTL:** In this setting, ID and OOD domain share the same label space. We choose the dataset pairs of: SVHN→MNIST-M and CIFAR10→STL10;

- **(ii) Open-set NTL:** In this setting, ID and OOD domain have disjoint label space. We conduct experiments on SVHN→CIFAR10 and CIFAR10→Digits.

- **(iii) NTL-based backdoor:** The OOD domain is the ID domain with backdoor triggers. We add four triggers: BadNets(sq), BadNets(grid) (Gu et al., 2019), Blended (Chen et al., 2017), Sig (Barni et al., 2019) on CIFAR10, and treat them individually as distinct OOD domains. The clean CIFAR10 is regarded as the ID domain.

Our aim is to address the malign OOD-Trap effect introduced by NTL teachers in DFKD, and thus, regardless of NTL teacher's tasks, our goal is to transfer only the ID

*Table 1.* Close-set NTL. We report the ID domain accuracy (IAcc) in blue and OOD domain accuracy (OAcc) in red. Results from the original DFKD baselines are highlighted with gray rows. The accuracy drop compared to the pre-trained NTL model is shown in brackets.

| | ID: SVHN OOD: MNIST-M | | | | ID: CIFAR10 OOD: STL | | | |
| | VGG-13→VGG-11 | | ResNet-34→ResNet-18 | | VGG-13→VGG-11 | | ResNet-34→ResNet-18 | |
| | IAcc (%) ↑ | OAcc (%) ↑ | IAcc (%) ↑ | OAcc (%) ↑ | IAcc (%) ↑ | OAcc (%) ↑ | IAcc (%) ↑ | OAcc (%) ↑ |
|---|---|---|---|---|---|---|---|---|
| NTL Teacher | 94.3$_{\pm0.1}$ | 9.9$_{\pm0.0}$ | 94.4$_{\pm0.1}$ | 9.9$_{\pm0.0}$ | 92.7$_{\pm0.1}$ | 10.6$_{\pm0.1}$ | 90.8$_{\pm0.1}$ | 10.4$_{\pm0.0}$ |
| DFQ | 89.2$_{\pm1.6}$ (-5.1) | 9.9$_{\pm0.0}$ (-0.0) | 79.5$_{\pm23.4}$ (-14.9) | 10.2$_{\pm0.2}$ (+0.3) | 42.7$_{\pm3.6}$ (-50.0) | 10.4$_{\pm0.0}$ (-0.2) | 52.5$_{\pm9.5}$ (-38.3) | 11.8$_{\pm1.3}$ (+1.4) |
| + CKD | 88.9$_{\pm0.9}$ (-5.4) | 11.1$_{\pm1.2}$ (+1.2) | 91.7$_{\pm1.5}$ (-2.7) | 10.1$_{\pm0.1}$ (+0.2) | 63.0$_{\pm3.3}$ (-29.7) | 10.4$_{\pm0.0}$ (-0.2) | 68.0$_{\pm10.1}$ (-22.8) | 22.5$_{\pm7.3}$ (+12.1) |
| + ATEsc | 89.3$_{\pm0.6}$ (-5.0) | 12.9$_{\pm3.9}$ (+3.0) | 86.2$_{\pm9.3}$ (-8.2) | 30.8$_{\pm7.2}$ (+20.9) | 62.3$_{\pm2.6}$ (-30.4) | 31.5$_{\pm1.6}$ (+20.9) | 66.2$_{\pm0.7}$ (-24.6) | 51.6$_{\pm6.5}$ (+41.2) |
| CMI | 6.4$_{\pm0.0}$ (-87.9) | 9.9$_{\pm0.0}$ (-0.0) | 60.2$_{\pm3.4}$ (-34.2) | 10.4$_{\pm0.4}$ (+0.5) | 9.9$_{\pm1.8}$ (-82.8) | 11.2$_{\pm2.3}$ (+0.6) | 37.7$_{\pm3.5}$ (-53.1) | 11.0$_{\pm0.4}$ (+0.6) |
| + CKD | 89.8$_{\pm0.1}$ (-4.5) | 29.1$_{\pm2.1}$ (+19.2) | 72.3$_{\pm4.2}$ (-22.1) | 28.6$_{\pm4.4}$ (+18.7) | 53.6$_{\pm3.7}$ (-39.1) | 38.3$_{\pm1.8}$ (+27.7) | 49.8$_{\pm3.6}$ (-41.0) | 33.9$_{\pm2.3}$ (+23.5) |
| + ATEsc | 58.7$_{\pm8.1}$ (-35.6) | 26.7$_{\pm6.4}$ (+16.8) | 76.5$_{\pm3.6}$ (-17.9) | 29.9$_{\pm4.5}$ (+20.0) | 58.6$_{\pm1.1}$ (-34.1) | 32.5$_{\pm2.9}$ (+21.9) | 44.7$_{\pm5.8}$ (-46.1) | 32.9$_{\pm5.5}$ (+22.5) |
| NAYER | 7.2$_{\pm0.3}$ (-87.1) | 10.6$_{\pm0.3}$ (+0.7) | 84.4$_{\pm0.8}$ (-10.0) | 10.0$_{\pm0.1}$ (+0.1) | 10.6$_{\pm0.0}$ (-82.1) | 10.4$_{\pm0.0}$ (-0.2) | 48.7$_{\pm1.1}$ (-42.1) | 10.4$_{\pm0.0}$ (-0.0) |
| + CKD | 89.6$_{\pm0.4}$ (-4.7) | 29.7$_{\pm3.2}$ (+19.8) | 89.2$_{\pm0.9}$ (-5.2) | 39.1$_{\pm1.3}$ (+29.2) | 62.7$_{\pm6.1}$ (-30.0) | 23.7$_{\pm1.1}$ (+13.1) | 64.4$_{\pm0.8}$ (-26.4) | 40.2$_{\pm3.9}$ (+29.8) |
| + ATEsc | 86.1$_{\pm1.8}$ (-8.2) | 33.7$_{\pm3.6}$ (+23.8) | 87.0$_{\pm3.7}$ (-7.4) | 40.0$_{\pm0.3}$ (+30.1) | 69.7$_{\pm5.4}$ (-23.0) | 42.3$_{\pm2.7}$ (+31.7) | 52.7$_{\pm2.4}$ (-38.1) | 39.2$_{\pm1.3}$ (+28.8) |

*Table 2.* Open-set NTL. We report ID domain accuracy (IAcc) in blue and OOD domain accuracy to OOD label (OLAcc) in red. Results from the DFKD baselines are highlighted with gray rows. Accuracy drop compared to the pre-trained model is shown in brackets.

| | ID: SVHN OOD: CIFAR10 | | | | ID: CIFAR10 OOD: Digits | | | |
| | VGG-13→VGG-11 | | ResNet-34→ResNet-18 | | VGG-13→VGG-11 | | ResNet-34→ResNet-18 | |
| | IAcc (%) ↑ | OLAcc (%) ↓ | IAcc (%) ↑ | OLAcc (%) ↓ | IAcc (%) ↑ | OLAcc (%) ↓ | IAcc (%) ↑ | OLAcc (%) ↓ |
|---|---|---|---|---|---|---|---|---|
| NTL Teacher | 93.4$_{\pm0.8}$ | 100.0$_{\pm0.0}$ | 93.0$_{\pm0.1}$ | 99.9$_{\pm0.0}$ | 92.9$_{\pm0.0}$ | 99.5$_{\pm0.3}$ | 91.1$_{\pm0.1}$ | 100.0$_{\pm0.0}$ |
| DFQ | 89.1$_{\pm0.9}$ (-4.3) | 100.0$_{\pm0.0}$ (-0.0) | 89.5$_{\pm0.1}$ (-3.5) | 99.9$_{\pm0.0}$ (-0.0) | 62.8$_{\pm2.3}$ (-30.1) | 99.8$_{\pm0.1}$ (+0.3) | 57.4$_{\pm8.8}$ (-33.7) | 85.8$_{\pm8.6}$ (-14.2) |
| + CKD | 89.0$_{\pm0.4}$ (-4.4) | 99.9$_{\pm0.1}$ (-0.1) | 89.1$_{\pm1.5}$ (-3.9) | 98.8$_{\pm0.6}$ (-1.1) | 68.5$_{\pm1.4}$ (-24.4) | 98.6$_{\pm0.4}$ (-0.9) | 70.1$_{\pm4.0}$ (-21.0) | 5.5$_{\pm3.6}$ (-94.5) |
| + ATEsc | 88.1$_{\pm3.2}$ (-5.3) | 100.0$_{\pm0.0}$ (-0.0) | 88.9$_{\pm0.3}$ (-4.1) | 16.1$_{\pm10.2}$ (-83.8) | 60.3$_{\pm4.5}$ (-32.6) | 78.4$_{\pm20.9}$ (-21.1) | 65.1$_{\pm7.9}$ (-26.0) | 2.5$_{\pm2.9}$ (-97.5) |
| CMI | 6.4$_{\pm0.0}$ (-87.0) | 99.0$_{\pm1.7}$ (-1.0) | 50.0$_{\pm2.0}$ (-43.0) | 99.1$_{\pm0.2}$ (-0.8) | 10.6$_{\pm0.0}$ (-82.3) | 100.0$_{\pm0.0}$ (+0.5) | 34.0$_{\pm1.3}$ (-57.1) | 76.1$_{\pm5.7}$ (-23.9) |
| + CKD | 91.3$_{\pm0.2}$ (-2.1) | 29.3$_{\pm16.0}$ (-70.7) | 74.5$_{\pm4.6}$ (-18.5) | 0.1$_{\pm0.1}$ (-99.8) | 59.4$_{\pm3.9}$ (-33.5) | 30.1$_{\pm7.0}$ (-69.4) | 50.1$_{\pm3.9}$ (-41.0) | 5.4$_{\pm6.4}$ (-94.6) |
| + ATEsc | 72.0$_{\pm2.0}$ (-21.4) | 0.0$_{\pm0.0}$ (-100.0) | 77.4$_{\pm2.2}$ (-15.6) | 0.0$_{\pm0.0}$ (-99.9) | 56.9$_{\pm3.5}$ (-36.0) | 21.1$_{\pm16.2}$ (-78.4) | 43.9$_{\pm6.4}$ (-47.2) | 3.7$_{\pm2.4}$ (-96.3) |
| NAYER | 6.4$_{\pm0.0}$ (-87.0) | 100.0$_{\pm0.0}$ (-0.0) | 54.8$_{\pm30.8}$ (-38.2) | 99.9$_{\pm0.1}$ (-0.0) | 10.6$_{\pm0.0}$ (-82.3) | 100.0$_{\pm0.0}$ (+0.5) | 49.4$_{\pm2.6}$ (-41.7) | 97.9$_{\pm0.8}$ (-2.1) |
| + CKD | 91.2$_{\pm0.2}$ (-2.2) | 67.5$_{\pm10.5}$ (-32.5) | 88.1$_{\pm1.7}$ (-4.9) | 0.2$_{\pm0.1}$ (-99.7) | 70.4$_{\pm1.4}$ (-22.5) | 71.6$_{\pm10.9}$ (-27.9) | 61.2$_{\pm3.9}$ (-29.9) | 6.3$_{\pm1.3}$ (-93.7) |
| + ATEsc | 89.2$_{\pm1.6}$ (-4.2) | 0.0$_{\pm0.0}$ (-100.0) | 85.9$_{\pm1.3}$ (-7.1) | 0.0$_{\pm0.0}$ (-99.9) | 69.8$_{\pm1.9}$ (-23.1) | 2.8$_{\pm3.8}$ (-96.7) | 59.0$_{\pm3.0}$ (-32.1) | 3.6$_{\pm0.6}$ (-96.4) |

*Table 3.* NTL-based backdoor. We report the ID domain accuracy (IAcc) in blue and attack success rate (ASR) in red. Results from the DFKD baselines are highlighted with gray rows. The accuracy drop compared to the pre-trained model is shown in brackets.

| | BadNet(sq) | | BadNet(grid) | | Blended | | Sig | |
| | VGG-13→VGG-11 | | VGG-13→VGG-11 | | ResNet-34→ResNet-18 | | ResNet-34→ResNet-18 | |
| | IAcc (%) ↑ | ASR (%) ↓ | IAcc (%) ↑ | ASR (%) ↓ | IAcc (%) ↑ | ASR (%) ↓ | IAcc (%) ↑ | ASR (%) ↓ |
|---|---|---|---|---|---|---|---|---|
| NTL Teacher | 90.3$_{\pm0.4}$ | 98.3$_{\pm0.1}$ | 90.7$_{\pm0.0}$ | 100.0$_{\pm0.0}$ | 90.5$_{\pm0.2}$ | 100.0$_{\pm0.0}$ | 90.2$_{\pm0.2}$ | 100.0$_{\pm0.0}$ |
| DFQ | 67.6$_{\pm2.5}$ (-22.7) | 71.3$_{\pm48.8}$ (-27.0) | 72.3$_{\pm1.9}$ (-18.4) | 100.0$_{\pm0.0}$ (-0.0) | 52.1$_{\pm26.7}$ (-38.4) | 99.9$_{\pm0.0}$ (-0.1) | 62.5$_{\pm19.8}$ (-27.7) | 99.9$_{\pm0.1}$ (-0.1) |
| + CKD | 70.9$_{\pm2.0}$ (-19.4) | 96.6$_{\pm4.0}$ (-1.7) | 75.2$_{\pm0.2}$ (-15.5) | 100.0$_{\pm0.0}$ (-0.0) | 52.2$_{\pm22.2}$ (-38.3) | 99.9$_{\pm0.1}$ (-0.1) | 68.2$_{\pm18.9}$ (-22.0) | 100.0$_{\pm0.0}$ (-0.0) |
| + ATEsc | 66.6$_{\pm2.2}$ (-23.7) | 2.9$_{\pm3.6}$ (-95.4) | 62.1$_{\pm6.2}$ (-28.6) | 11.4$_{\pm9.1}$ (-88.6) | 57.0$_{\pm4.9}$ (-33.5) | 43.1$_{\pm46.6}$ (-56.9) | 65.8$_{\pm8.8}$ (-24.4) | 65.5$_{\pm29.5}$ (-34.5) |
| CMI | 10.0$_{\pm0.0}$ (-80.3) | 100.0$_{\pm0.0}$ (+1.7) | 10.0$_{\pm0.0}$ (-80.7) | 100.0$_{\pm0.0}$ (-0.0) | 50.5$_{\pm1.1}$ (-40.0) | 99.9$_{\pm0.1}$ (-0.1) | 46.0$_{\pm2.2}$ (-44.2) | 100.0$_{\pm0.0}$ (-0.0) |
| + CKD | 76.3$_{\pm0.8}$ (-14.0) | 96.5$_{\pm0.6}$ (-1.8) | 74.1$_{\pm3.2}$ (-16.6) | 100.0$_{\pm0.0}$ (-0.0) | 68.3$_{\pm8.8}$ (-22.2) | 48.0$_{\pm39.3}$ (-52.0) | 67.1$_{\pm14.3}$ (-23.1) | 100.0$_{\pm0.0}$ (-0.0) |
| + ATEsc | 68.5$_{\pm4.9}$ (-21.8) | 3.6$_{\pm3.5}$ (-94.7) | 62.8$_{\pm4.0}$ (-27.9) | 2.2$_{\pm1.9}$ (-97.8) | 59.7$_{\pm16.4}$ (-30.8) | 19.0$_{\pm5.5}$ (-81.0) | 46.9$_{\pm3.8}$ (-43.3) | 32.2$_{\pm15.1}$ (-67.8) |
| NAYER | 10.0$_{\pm0.0}$ (-80.3) | 100.0$_{\pm0.0}$ (+1.7) | 10.0$_{\pm0.0}$ (-80.7) | 100.0$_{\pm0.0}$ (-0.0) | 70.6$_{\pm2.6}$ (-19.9) | 100.0$_{\pm0.0}$ (-0.0) | 65.8$_{\pm3.5}$ (-24.4) | 100.0$_{\pm0.0}$ (-0.0) |
| + CKD | 80.8$_{\pm3.3}$ (-9.5) | 97.5$_{\pm0.4}$ (-0.8) | 74.3$_{\pm19.7}$ (-16.4) | 100.0$_{\pm0.0}$ (-0.0) | 79.0$_{\pm1.9}$ (-11.5) | 99.2$_{\pm0.9}$ (-0.8) | 70.1$_{\pm15.1}$ (-20.1) | 100.0$_{\pm0.0}$ (-0.0) |
| + ATEsc | 68.9$_{\pm4.7}$ (-21.4) | 34.6$_{\pm47.3}$ (-63.7) | 77.8$_{\pm3.9}$ (-12.9) | 68.5$_{\pm47.1}$ (-31.5) | 67.0$_{\pm3.1}$ (-23.5) | 10.2$_{\pm6.3}$ (-89.8) | 55.8$_{\pm2.1}$ (-34.4) | 56.2$_{\pm17.6}$ (-43.8) |

domain knowledge from NTL teacher to student.

**DFKD baselines.** We choose DFQ (Choi et al., 2020), CMI (Fang et al., 2021b), NAYER (Tran et al., 2024) as DFKD baselines. When conduct DFKD, all NTL teachers are considered as white-box models. For each baseline, we apply two variants of our method as plug-and-play modules: one using only the CKD in Section 4.3 (denoted as + CKD) and the other using the full method (denoted as + ATEsc).

**Implementation details.** For each task, NTL teachers are pre-trained up to 50 epochs (input images resize to 64×64). For DFKD, we train each method up to 200 epochs, and all

hyper-parameters follow original implementations. VGG-13 (Simonyan, 2014) and ResNet-34 (He et al., 2016) are used as teacher architectures, while VGG-11 and ResNet-18 are used as student architectures. We conduct all experiments on a single RTX 4090 GPU (24G). More details are shown in Appendix E.1 due to the limited space of the main paper.

### 5.1. DFKD on Close-set NTL

The results of distilling close-set NTL teachers are shown in Table 1, covering various dataset pairs, network architectures, and DFKD baselines. We use ID domain accuracy (**IAcc**) to measure the performance of ID knowledge transfer.

Since ID and OOD domains share the same label space in this setting, we compute the OOD domain accuracy (**OAcc**) based on the student's predictions on OOD data relative to their ground-truth labels. Higher OAcc means better ID-to-OOD generalization, and also means better resistance to misleading OOD knowledge transfer during DFKD. From the results, each baseline DFKD method performs poorly on both the ID and OOD domains, with the student exhibiting low IAcc and OAcc. For some tasks (such as CMI VGG-13→VGG-11), the DFKD method cannot even transfer valid knowledge to the student. With the addition of CKD, IAcc generally shows significant improvement across tasks, whereas the gains in OAcc are relatively modest. With the full `ATEsc` applied, OAcc further improves, indicating the effectiveness of the forgetting term in mitigating the transfer of misleading OOD knowledge.

### 5.2. DFKD on Open-set NTL

For open-set NTL, the label space of ID and OOD domains is disjoint. For evaluating the transfer of misleading OOD knowledge, we calculate the accuracy between the prediction and the OOD-class label on OOD domain (denoted as **OLAcc**). Lower OLAcc indicates better resistance to OOD knowledge transfer. Moreover, we still use **IAcc** on ID domains. From the results shown in Table 2, the independent effect of CKD contributes to the significant improvements of ID domain knowledge transfer. It also can mitigate the OOD knowledge transfer in some extent. By using the complete `ATEsc`, the OOD knowledge transfer can be further mitigated, while it also sacrify some ID performance.

### 5.3. Defending NTL-based Backdoor in DFKD

We further explore the use of NTL in conducting controllable backdoor attacks (Wu et al., 2022). We use four triggers: BadNets(sq), BadNets(grid) (Gu et al., 2019), Blended (Chen et al., 2017), Sig (Barni et al., 2019) on CIFAR10, and treat them individually as distinct OOD domains. The clean CIFAR10 is regarded as the ID domain. For evaluation, we follow backdoor learning to use clean label accuracy on ID domain (i.e., the same to **IAcc**) and attack success rate (**ASR**) on backdoor data, where the ASR is the accuracy of the prediction and the OOD-class label (i.e., the backdoor target label) on trigger OOD data. From the results shown in Table 3, the full `ATEsc` can alleviate the backdoor transfer from teacher to student, while all the original baselines or only adding the CKD can still inherit backdoor, with the ASR close to the teacher's ASR. Furthermore, we compare `ATEsc` with the defense method ABD (Hong et al., 2023), with the results and analysis presented in Appendix E.5.

### 5.4. Additional Analysis

Due to limited space, additional analysis are shown in appendices, including but not limited to: toy experiments of OOD trap effect (Appendix D), NTL's adversarial difference (Appendices C.5 to C.7), ablation studies (Appendices E.2 to E.4), visualization (Appendix E.7).

## 6. Conclusion

In this work, for the first time, we identify the OOD trap effect from NTL teachers to DFKD, resulting in the degradation of ID knowledge transfer and enabling OOD misleading knowledge transfer. Then, inspired by the difference of NTL model's adversarial robustness on ID and OOD domains, we propose a novel plug-and-play approach `ATEsc` to enhance DFKD in transferring correct knowledge from NTL teachers. The effectiveness of `ATEsc` is validated via extensive experiments across diverse OOD domain configurations, datasets, network architectures, and DFKD baselines.

## Acknowledgements

The authors would like to thank the anonymous reviewers for their insightful and constructive comments. Ziming Hong is supported by JD Technology Scholarship for Postgraduate Research in Artificial Intelligence No. SC4103. Zengmao Wang is supported by NSFC 62476204, 62271357, NSF of Hubei Province 2023BAB072, Wuhan NSF 2024040801020236. Bo Han is supported by RGC Young Collaborative Research Grant No. C2005-24Y and NSFC General Program No. 62376235. Bo Du is supported by NSFC 62225113, Innovative Research Group Project of Hubei Province 2024AFA017. Tongliang Liu is partially supported by the following Australian Research Council projects: FT220100318, DP220102121, LP220100527, LP220200949, and IC190100031.

## Impact Statement

This paper presents work whose goal is to advance the field of data-free knowledge distillation (DFKD). Existing works always assume pre-trained teachers are trustworthy. Therefore, the robustness and security of DFKD from untrusted teachers remain largely unexplored. Our work investigates how DFKD performs when faced with non-transferable learning (NTL) teachers. For the first time, we identify the *OOD trap effect*, a prevalent issue in distilling NTL teachers via DFKD, and further uncover its intrinsic causes: ID-to-OOD synthetic distribution shift and ID-OOD learning task conflicts. To address this issue, we propose a novel plug-and-play approach `ATEsc` to enhance DFKD via handle with the *OOD trap effect*, enabling benign knowledge transfer from NTL teachers via DFKD. Our work takes an important step forward in enhancing the robustness and security of DFKD. However, our method may also raise a challenge for NTL-based model intellectual property (IP) protection, as it provides a data-free inverse solution against NTL models, potentially undermining their intended transferability restrictions. It is important to develop defense strategies against `ATEsc` for NTL-based model intellectual property (IP) protection in the future.

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

## Appendices

## A. Related Work of Data-Free Knowledge Distillation

Data-free knowledge distillation (DFKD) deals with distilling teacher's knowledge to a smaller student without requiring to access the in-distribution (ID) data (Micaelli & Storkey, 2019; Luo et al., 2020; Yu et al., 2023; Liu et al., 2024). Due to the unavailability of the teacher's ID data $\mathcal{D}_{id}$, one needs to find substitution data to conduct knowledge distillation. To this end, previous works try to directly optimize randomly initialized noise images (Yin et al., 2020) under the guidance of a teacher, or leverage unlabeled data from the wild (Fang et al., 2021a). The former one faces efficiency issues due to per-image optimization, and the latter one needs extra data. Generative Adversarial Networks (GANs)-based framework become the mainstream solution, where a generator is introduced to synthesize fake data in an *adversarial exploration* way (Liu et al., 2021b; Micaelli & Storkey, 2019; Fang et al., 2019). Specifically, the generator is trained to synthesize fake samples causing disagreement between student and teacher, thus expanding the coverage of the data distribution. Then, the synthetic samples are used as substitution data to conduct knowledge distillation, thus transferring knowledge from teacher to student.

Early stage DFKD methods such as ZSKT (Micaelli & Storkey, 2019) primarily rely on the adversarial exploration. However, only using adversarial exploration can bring non-stationary distribution problem and loss the diversity of synthesizing. The non-stationary distribution problem causes the catastrophic forgetting of the student models. This issue can be mitigated by memory-bank-based strategies (Fang et al., 2021b; Do et al., 2022; Patel et al., 2023; Binici et al., 2022b; Tran et al., 2024), where old synthetic samples are stored and will be replayed in future epochs. For the diversity problem, following works try to encourage the diversity of the synthetic images by using class-balance loss (Choi et al., 2020), contrastive loss (Fang et al., 2021b), image augmentation (Yu et al., 2023).

DFKD has driven multiple downstream tasks, especially in data-privacy scenarios, such as data-free federated learning (Zhu et al., 2021b; Zhang et al., 2022), data-free meta-learning (Wei et al., 2024; Hu et al., 2023), data-free quantization (Qian et al., 2023; Choi et al., 2020), data-free model fusion (Shi et al., 2024). DFKD may also be used in malicious way: one example is data-free model extraction (DFME) (Kariyappa et al., 2021; Truong et al., 2021; Wang et al., 2024c; Wang et al.).

Although significant progress has been made in the field of DFKD, existing works always assume pre-trained teachers are trustworthy. Hong et al. (2023) conducted the first effort to explore DFKD in the context of untrusted teachers. However, their work primarily focuses on teachers with backdoor (Wu et al., 2022; Gu et al., 2019). Another similar work is DERD (Zhou et al., 2024), while they mainly focus on transferring the adversarial robustness (Yu et al., 2022b;a) from a robust teacher to a student via DFKD. In this work, we conduct the first investigation on the DFKD against non-transferable learning (NTL) teachers (Wang et al., 2022; Hong et al., 2024b; 2025; Xiang et al., 2025; Hong et al., 2024a; Peng et al., 2024; Wang et al., 2023; 2024a; Deng et al., 2024; Zeng & Lu, 2022; Ding et al., 2024).

# B. Algorithms

## B.1. Training NTL

---
**Algorithm 2** Training NTL
---

1: **Input:** an initialized model $f$ with parameters $\theta_f$; total training epochs $E$; labeled ID domain training data $\mathcal{D}_{\text{id}} = \{(\boldsymbol{x}_i^{\text{id}}, y_i^{\text{id}})\}_{i=1}^{N_{\text{id}}}$; unlabeled OOD domain training data $\mathcal{D}_{\text{ood}} = \{(\boldsymbol{x}_i^{\text{ood}})\}_{i=1}^{N_{\text{ood}}}$; batchsize $B$; hyper-parameter $\alpha$.
2: **Output:** the NTL model $f$.
3: **for** $e = 1$ to $E$ **do**
4:      ▷ *Forward*
5:      Sample a batch of ID data $\mathcal{B}_{\text{id}} = \{(\boldsymbol{x}_i^{\text{id}}, y_i^{\text{id}})\}_{i=1}^{B/2}$ from $\mathcal{D}_{\text{id}}$ and a batch of OOD data $\mathcal{B}_{\text{ood}} = \{(\boldsymbol{x}_i^{\text{ood}})\}_{i=1}^{B/2}$ from $\mathcal{D}_{\text{ood}}$.
6:      Concat ID and OOD data into a complete batch $\mathcal{B} = \{\boldsymbol{x}_i\}_{i=1}^{B}$.
7:      Obtain the batch of model prediction $\hat{\mathcal{Y}} = \{\hat{y}_i | \hat{y}_i = f(\boldsymbol{x}_i), \boldsymbol{x}_i \in \mathcal{B}\}_{i=1}^{B}$.
8:      Split the prediction into ID: $\hat{\mathcal{Y}}_{\text{id}} = \{\hat{y}_i^{\text{id}} | \hat{y}_i^{\text{id}} = f(\boldsymbol{x}_i^{\text{id}}), \boldsymbol{x}_i^{\text{id}} \in \mathcal{B}_{\text{id}}\}_{i=1}^{B}$ and OOD: $\hat{\mathcal{Y}}_{\text{ood}} = \{\hat{y}_i^{\text{ood}} | \hat{y}_i^{\text{ood}} = f(\boldsymbol{x}_i^{\text{ood}}), \boldsymbol{x}_i^{\text{ood}} \in \mathcal{B}_{\text{ood}}\}_{i=1}^{B}$.
9:      ▷ *ID loss*
10:     Calculate the ID loss $\mathcal{L}_{\text{id}} = \frac{2}{B} \sum_{i=1}^{B/2} D_{\text{KL}}(f(\boldsymbol{x}_i^{\text{id}}), y_i^{\text{id}})$.
11:     ▷ *OOD loss*
12:     Calculate the OOD loss $\mathcal{L}_{\text{out}} = \frac{2}{B} \sum_{i=1}^{B/2} D_{\text{KL}}(f(\boldsymbol{x}_i^{\text{ood}}), f(\boldsymbol{x}_i^{\text{id}}))$ and $\mathcal{L}_{\text{feat}} = \frac{2}{B} \sum_{i=1}^{B/2} \text{MMD}(f_e(\boldsymbol{x}_i^{\text{ood}}), f_e(\boldsymbol{x}_i^{\text{id}}))$.
13:     Obtain the full NTL loss $\mathcal{L}_{\text{NTL}} = \mathcal{L}_{\text{id}} - \min(1, \alpha \cdot \mathcal{L}_{\text{out}} \cdot \mathcal{L}_{\text{feat}})$.
14:     ▷ *Back Propagation*
15:     Update $\theta_f$ via minimizing $\mathcal{L}_{\text{NTL}}$.
16: **end for**

---

---
**Algorithm 3** Training NTL-cls
---

1: **Input:** an initialized model $f$ with parameters $\theta_f$; total training epochs $E$; labeled ID domain training data $\mathcal{D}_{\text{id}} = \{(\boldsymbol{x}_i^{\text{id}}, y_i^{\text{id}})\}_{i=1}^{N_{\text{id}}}$; unlabeled OOD domain training data $\mathcal{D}_{\text{ood}} = \{(\boldsymbol{x}_i^{\text{ood}})\}_{i=1}^{N_{\text{ood}}}$; batchsize $B$; hyper-parameter $\alpha$ and $\lambda_{\text{cls}}$; the OOD-class index $y_{\text{ood}}$.
2: **Output:** the NTL model $f$.
3: **for** $e = 1$ to $E$ **do**
4:      ▷ *Forward*
5:      Sample a batch of ID data $\mathcal{B}_{\text{id}} = \{(\boldsymbol{x}_i^{\text{id}}, y_i^{\text{id}})\}_{i=1}^{B/2}$ from $\mathcal{D}_{\text{id}}$ and a batch of OOD data $\mathcal{B}_{\text{ood}} = \{(\boldsymbol{x}_i^{\text{ood}})\}_{i=1}^{B/2}$ from $\mathcal{D}_{\text{ood}}$.
6:      Concat ID and OOD data into a complete batch $\mathcal{B} = \{\boldsymbol{x}_i\}_{i=1}^{B}$.
7:      Obtain the batch of model prediction $\hat{\mathcal{Y}} = \{\hat{y}_i | \hat{y}_i = f(\boldsymbol{x}_i), \boldsymbol{x}_i \in \mathcal{B}\}_{i=1}^{B}$.
8:      Split the prediction into ID: $\hat{\mathcal{Y}}_{\text{id}} = \{\hat{y}_i^{\text{id}} | \hat{y}_i^{\text{id}} = f(\boldsymbol{x}_i^{\text{id}}), \boldsymbol{x}_i^{\text{id}} \in \mathcal{B}_{\text{id}}\}_{i=1}^{B}$ and OOD: $\hat{\mathcal{Y}}_{\text{ood}} = \{\hat{y}_i^{\text{ood}} | \hat{y}_i^{\text{ood}} = f(\boldsymbol{x}_i^{\text{ood}}), \boldsymbol{x}_i^{\text{ood}} \in \mathcal{B}_{\text{ood}}\}_{i=1}^{B}$.
9:      ▷ *ID loss*
10:     Calculate the ID loss $\mathcal{L}_{\text{id}} = \frac{2}{B} \sum_{i=1}^{B/2} D_{\text{KL}}(f(\boldsymbol{x}_i^{\text{id}}), y_i^{\text{id}})$.
11:     ▷ *OOD loss*
12:     Calculate the OOD loss $\mathcal{L}_{\text{out}} = \frac{2}{B} \sum_{i=1}^{B/2} D_{\text{KL}}(f(\boldsymbol{x}_i^{\text{ood}}), f(\boldsymbol{x}_i^{\text{id}}))$ and $\mathcal{L}_{\text{feat}} = \frac{2}{B} \sum_{i=1}^{B/2} \text{MMD}(f_e(\boldsymbol{x}_i^{\text{ood}}), f_e(\boldsymbol{x}_i^{\text{id}}))$.
13:     Calculate the class-controlling loss of OOD data $\mathcal{L}_{\text{cls}} = \frac{2}{B} \sum_{i=1}^{B/2} \mathcal{L}_{\text{CE}}(f(\boldsymbol{x}_i^{\text{ood}}), y_{\text{ood}})$.
14:     Obtain the full NTL loss $\mathcal{L}_{\text{NTL-cls}} = \mathcal{L}_{\text{id}} - \min(1, \alpha \cdot \mathcal{L}_{\text{out}} \cdot \mathcal{L}_{\text{feat}}) + \lambda_{\text{cls}} \cdot \mathcal{L}_{\text{cls}}$.
15:     ▷ *Back Propagation*
16:     Update $\theta_f$ via minimizing $\mathcal{L}_{\text{NTL-cls}}$.
17: **end for**

---

## B.2. Training DFKD

---
**Algorithm 4** Training DFKD Baseline
---

1: **Input:** a teacher $f$; a generator $G$ with parameters $\theta_G$; a student $S$ with parameters $\theta_S$; total training epochs $E$; synthetic batchsize $B$; hyper-parameter $\lambda_{\text{bn}}$.
2: **Output:** the DFKD student $S$.
3: **for** $e = 1$ to $E$ **do**
4:      ▷ *Training Generator*
5:      Randomly sample a noise batch $\mathcal{B}_{\text{noise}} = \{\boldsymbol{n}_i | \boldsymbol{n}_i \sim \mathcal{D}_{\text{noise}}\}_{i=1}^{B}$.
6:      Calculate adversarial loss $\mathcal{L}_{\text{syn}} = \frac{1}{B} \sum_{i=1}^{B} D_{\text{KL}}(S(G(\boldsymbol{n}_i)), f(G(\boldsymbol{n}_i)))$.
7:      Calculate BN loss $\mathcal{L}_{\text{bn}} = \sum_l D_{\text{KL}}(\mathcal{N}(\mu_l(G(\mathcal{B}_{\text{noise}})), \sigma_l^2(G(\mathcal{B}_{\text{noise}}))), \mathcal{N}(\mu_l, \sigma_l^2))$.
8:      Updating $\theta_G$ via $\max_{\theta_G} \{\mathcal{L}_{\text{syn}} - \lambda_{\text{bn}} \cdot \mathcal{L}_{\text{bn}}\}$
9:      ▷ *Synthesize Samples and Conduct Knowledge Distillation*
10:     Randomly sample a noise batch $\mathcal{B}_{\text{noise}} = \{\boldsymbol{n}_i | \boldsymbol{n}_i \sim \mathcal{D}_{\text{noise}}\}_{i=1}^{B}$.
11:     Updating $\theta_S$ via $\min_{\theta_S} \left\{ \mathcal{L}_{\text{kd}} = \frac{1}{B} \sum_{i=1}^{B} D_{\text{KL}}(S(G(\boldsymbol{n}_i)), f(G(\boldsymbol{n}_i))) \right\}$;
12: **end for**

---

## C. More Empirically Results on NTL Teachers

In this section, we present more analysis and empirically results for NTL teachers and the OOD trap. In Appendix C.1, we show more training dynamic for distilling NTL teachers via DFKD. In Appendix C.2, we show more synthetic samples and t-SNE visualizations. In Appendix C.3, we present regular KD results. In Appendix C.4, we analyze the influence of BN loss for OOD trap. In Appendices C.5 to C.7, we show more empirical results to demonstrate the difference of adversarial robustness of ID and OOD domain for NTL teachers.

### C.1. Training Dynamic

In this section, we present more comparisons of the DFKD training dynamic between SL teachers and NTL teachers, as shown in Figures 7 and 8. Specifically, in Figure 7, we present three metrics the same as Figure 1: ID accuracy in distilling SL teacher (SL-ID) and NTL teacher (NTL-ID), and OOD accuracy (specific to the OOD-class) in distilling NTL teacher (NTL-OOD). We can see the same phenomenon as we illustrated in the main paper: **(i)** *degradation of ID knowledge transfer*: the student exhibits lower NTL-ID accuracy compared to distillation an SL teacher, and **(ii)** *misleading OOD knowledge transfer*: the student inherits teacher's misleading OOD knowledge, resulting in a high NTL-OOD accuracy.

Moreover, in Figure 8, we present results on closed-set NTL teachers, where we report the OOD accuracy based on the student's predictions to true data labels. This can directly reflect the generalization ability from ID to OOD domain. From the results, the NTL-ID accuracy is consistently lower than the SL-ID accuracy, demonstrating the degradation of ID knowledge transfer during DFKD training. Regarding OOD accuracy, SL teachers enable students to achieve a certain degree of ID-to-OOD generalization. However, NTL-OOD accuracy of DFKD students under NTL teachers cannot improve alongside the increasing NTL-ID accuracy. This indicates that NTL teachers tend to transfer misleading OOD knowledge to students through DFKD.

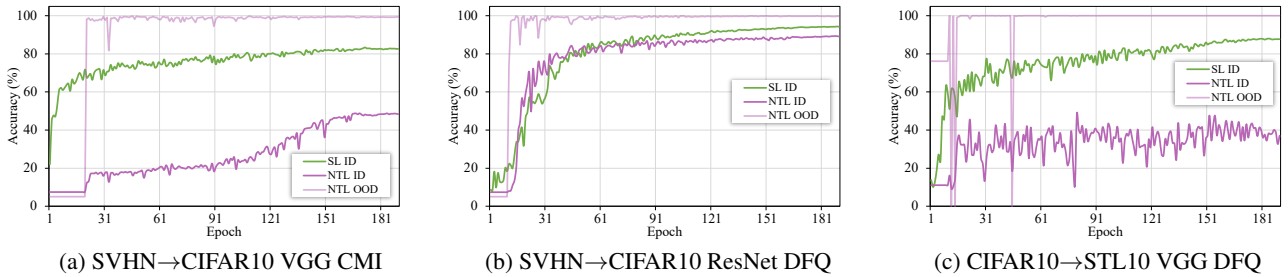

(a) SVHN→CIFAR10 VGG CMI      (b) SVHN→CIFAR10 ResNet DFQ      (c) CIFAR10→STL10 VGG DFQ

*Figure 7.* Comparison of training dynamic of SL and NTL teachers. The OOD accuracy is calculated between the student's predictions and OOD-class labels. This directly show the degree of OOD misleading knowledge transfer from teachers to students.

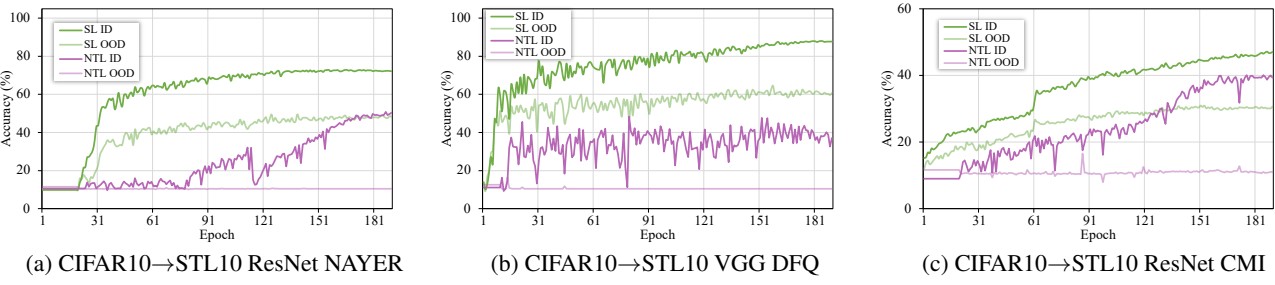

(a) CIFAR10→STL10 ResNet NAYER      (b) CIFAR10→STL10 VGG DFQ      (c) CIFAR10→STL10 ResNet CMI

*Figure 8.* Comparison of the training dynamic of the SL and NTL teachers on close-set tasks. We directly show the OOD accuracy from the student's predictions to true data labels. This can directly reflect the generalization ability from ID to OOD domain.

### C.2. Visualization of Synthetic Samples

In this section, we show the OOD trap effect from NTL teachers to DFKD by visualizing the synthetic samples during DFKD training. First, we show the real image samples from datasets we have involved. Example images from each dataset are shown in Figure 9, and CIFAR10 with four different types of backdoor triggers are shown in Figure 10.

Then, we visualize the synthetic samples during distilling a SL teacher and NTL teachers. Specifically, we use CMI (Fang

et al., 2021b) to distilling a SL teacher (pre-trained on CIFAR10) and three NTL teachers pre-trained on CIFAR10→Digits, CIFAR10→CIFAR10 w/ Sig, and CIFAR10→CIFAR10 w/ Blended. From the results shown in Figure 11, we can observe that the SL results look more like the real CIFAR10 samples in Figure 9 (f), while NTL results consistently integrate both the ID domain (CIFAR10) and OOD domain information (Digits in Figure 9 (a-e), Sig in Figure 9 (e), and Blended in Figure 10 (d)).

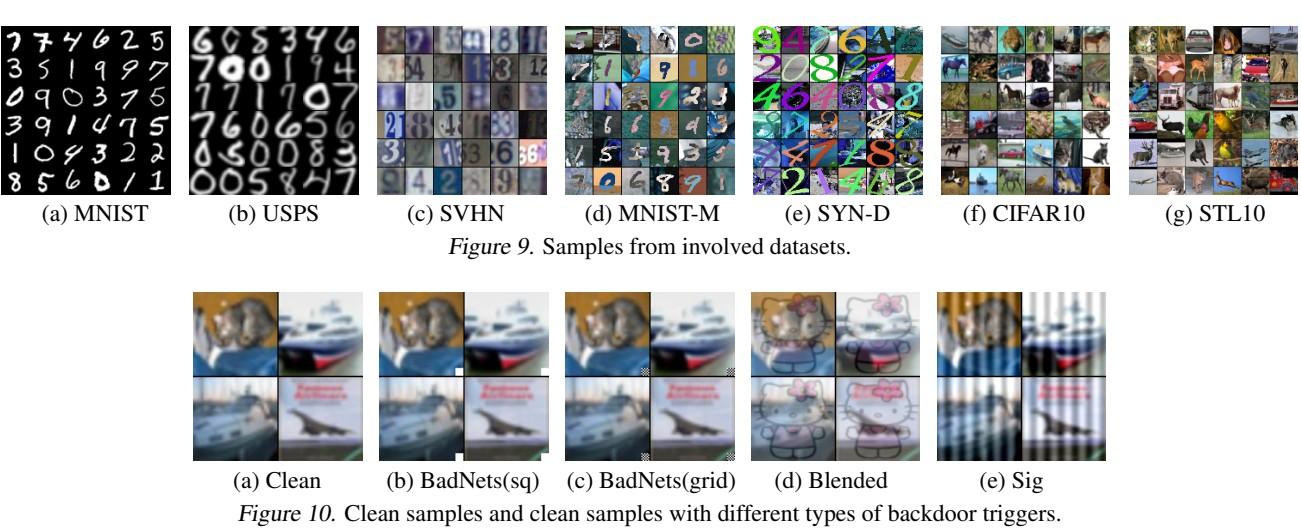

| (a) MNIST | (b) USPS | (c) SVHN | (d) MNIST-M | (e) SYN-D | (f) CIFAR10 | (g) STL10 |

*Figure 9.* Samples from involved datasets.

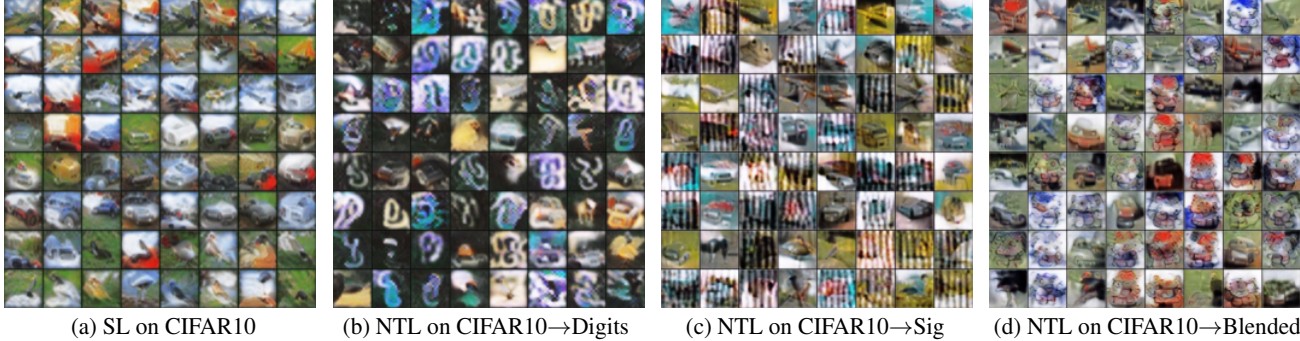

| (a) Clean | (b) BadNets(sq) | (c) BadNets(grid) | (d) Blended | (e) Sig |

*Figure 10.* Clean samples and clean samples with different types of backdoor triggers.

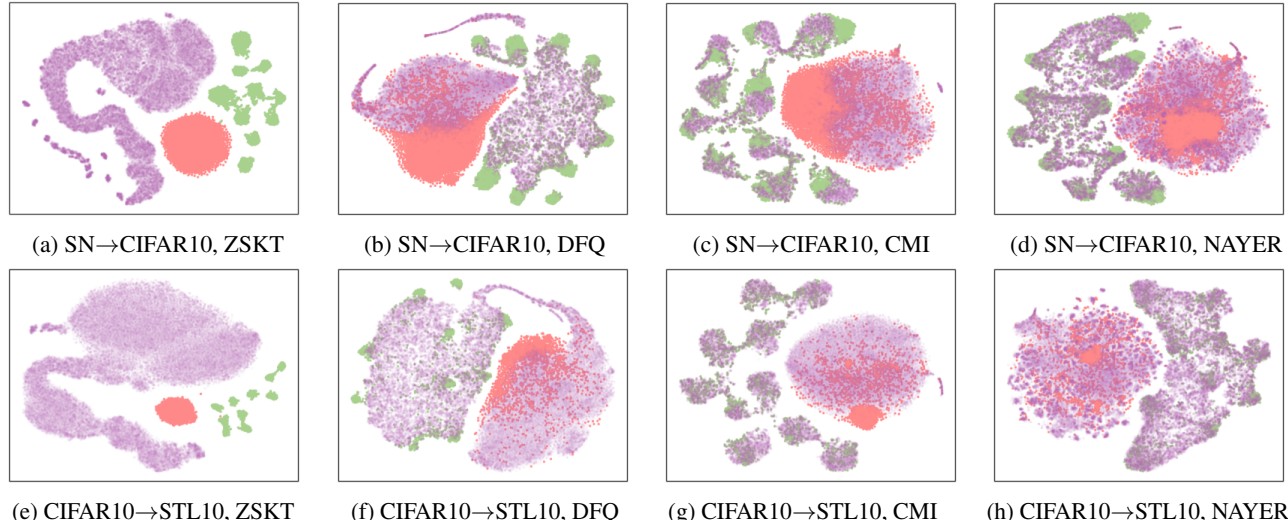

| (a) SL on CIFAR10 | (b) NTL on CIFAR10→Digits | (c) NTL on CIFAR10→Sig | (d) NTL on CIFAR10→Blended |

*Figure 11.* Comparison of synthetic samples (DFKD by CMI. $T$: ResNet34 and $S$: ResNet18).

| (a) SN→CIFAR10, ZSKT | (b) SN→CIFAR10, DFQ | (c) SN→CIFAR10, CMI | (d) SN→CIFAR10, NAYER |
| (e) CIFAR10→STL10, ZSKT | (f) CIFAR10→STL10, DFQ | (g) CIFAR10→STL10, CMI | (h) CIFAR10→STL10, NAYER |

*Figure 12.* t-SNE visualization of synthetic samples ($T$: ResNet34 and $S$: ResNet18). Green dots and red dots represent real ID and OOD domain samples, respectively. Purple transparent dots represent the synthetic samples.

Furthermore, we plot more t-SNE feature visualization results (Van der Maaten & Hinton, 2008) regarding the teacher's features of real ID samples, real OOD samples, and synthetic samples in Figure 12. We observe that for more advanced DFKD methods such as DFQ (Choi et al., 2020), CMI (Fang et al., 2021b), and NAYER (Tran et al., 2024), parts of synthetic samples's features overlap with real OOD domain features. This indicates that these samples are more similar to real OOD samples and can activate NTL teacher's OOD misleading knowledge. In particular, the basic adversarial exploration method ZSKT (Micaelli & Storkey, 2019) fails to approximate the real ID distribution, as its synthetic samples remain distant from ID clusters.

## C.3. Regular Knowledge Distillation Results

Although our work focuses on DFKD, which relies on a generator to synthesize fake samples, we also include regular knowledge distillation (KD) results with NTL teachers for reference, as shown in Table 4. Specifically, we present results of regular KD using ID data only (KD-ID), OOD data only (KD-OOD) and both ID and OOD data (KD-All). The results show that when only ID data is used, the student learns only ID knowledge. However, once OOD data is involved, the student inevitably learns misleading OOD knowledge from the NTL teacher.

Table 4. Regular knowledge distillation. We report the ID domain accuracy in blue and OOD domain accuracy in red. The accuracy drop compared to the pre-trained model is shown in brackets.

| | ID: **CIFAR10** OOD: **STL** | | ID: **CIFAR10** OOD: **Digits** | |
| | **ResNet-34→ResNet-18** | | **ResNet-34→ResNet-18** | |
| | IAcc (%) ↑ | OACC (%) ↑ | IAcc (%) ↑ | OLACC (%) ↓ |
|---|---|---|---|---|
| NTL Teacher | $90.8_{\pm0.1}$ | $10.4_{\pm0.0}$ | $91.1_{\pm0.1}$ | $100.0_{\pm0.0}$ |
| KD-ID | $80.4_{\pm0.2}$ (-10.4) | $58.1_{\pm1.0}$ (+47.7) | $82.2_{\pm0.3}$ (-8.9) | $30.0_{\pm3.7}$ (-70.0) |
| KD-OOD | $10.6_{\pm0.0}$ (-80.2) | $10.4_{\pm0.0}$ (+0.0) | $10.6_{\pm0.0}$ (-80.5) | $100.0_{\pm0.0}$ (-0.0) |
| KD-All | $76.6_{\pm0.4}$ (-14.2) | $10.5_{\pm0.0}$ (+0.1) | $81.8_{\pm0.5}$ (-9.3) | $100.0_{\pm0.0}$ (-0.0) |

## C.4. The Influence of Batch-Normalization Loss

In this section, we discuss the influence of Batch-Normalization (BN) loss (Yin et al., 2020) for the identified OOD trap effect from NTL teachers to DFKD. In DFKD, SOTA methods such as Choi et al. (2020); Fang et al. (2021b); Tran et al. (2024) commonly use BN loss (Equation 9) as regularization for training generator $G$. Specifically,

- As shown in Appendix B.1, when training NTL teachers, each batch contains a mixture of ID and OOD domain samples. This results in that BN layers in pre-trained NTL teachers record statistical information for the mixture distribution of ID and OOD domains (mean $\mu_l$ and var $\sigma_l^2$ for layer $l$).

- During DFKD, when training the generator $G$, BN loss (Equation 9) is represented as the divergence between synthetic feature statistics $\mathcal{N}(\mu_l(x_{syn}), \sigma_l^2(x_{syn}))$ and teacher's BN statistics $\mathcal{N}(\mu_l, \sigma_l^2)$. Minimizing BN loss lets the synthetic samples follow similar statistical information of teacher's training samples.

The joint effects of BN loss (Equation 9) and adversarial exploration loss (Equation 7) constraint the $G$ to: (i) follow the statistics of NTL teacher's training data (i.e., a mixture of real ID and OOD domains), and (ii) maximize the discrepancy between the student $S$ and the NTL teacher. As a result, the generator $G$ is trained to synthesize both ID-like and OOD-like samples (i.e., ID-to-OOD synthetic distribution shift). When all these synthetic samples are used for distillation, the student is inevitably taught to mimic the teacher's behavior on OOD domains, thereby learning misleading OOD knowledge. This also leads to a ID-OOD task conflict, ultimately degrading the student's performance on the ID domain.

Without the BN loss, the generator $G$ does not need to synthesize samples that resemble those from the OOD domain used to train the NTL teacher. However, the student's ID performance will be influenced by low quality of synthetic samples. Actually, when only using the adversarial exploration (such as ZSKT (Micaelli & Storkey, 2019)) to distill NTL teachers, the ID knowledge transfer may be completely blocked in the worst case. This results in the DFKD student exhibiting random-guess-like performance on the ID domain. Figure 1 (c) and a toy example in Appendix D illustrate this phenomenon.

Encouraging *synthesizing diversity* (e.g., class balance in DFQ (Choi et al., 2020), contrastive learning in CMI (Fang et al., 2021b)) can help to mitigate the complete blockage of ID knowledge transfer when using adversarial exploration individually. However, their ID performance is still limited by the poor quality of synthetic samples. Corresponding empirical evidence is shown in Table 5. Synthetic samples w/ and w/o BN are shown in Figures 13 to 16. From the results, when we conduct DFKD w/o BN loss, the synthetic samples do not like real images, and the ID-domain performance is even worse than DFKD w/ BN loss, despite the latter suffering from an ID-to-OOD synthetic distribution shift and ID-OOD task conflict.

*Table 5.* The influence of BN Loss. We report the ID domain accuracy in blue and OOD domain accuracy in red. The accuracy drop compared to the pre-trained model is shown in brackets.

|  | ID: **CIFAR10** OOD: **STL** | | ID: **CIFAR10** OOD: **Digits** | |
| --- | --- | --- | --- | --- |
|  | **ResNet-34→ResNet-18** | | **ResNet-34→ResNet-18** | |
|  | IAcc (%) ↑ | OACC (%) ↑ | IAcc (%) ↑ | OLACC (%) ↓ |
| NTL Teacher | $90.8_{\pm 0.1}$ | $10.4_{\pm 0.0}$ | $91.1_{\pm 0.1}$ | $100.0_{\pm 0.0}$ |
| CMI w/ BN | $37.7_{\pm 3.5}$ (-53.1) | $11.0_{\pm 0.4}$ (+0.6) | $34.0_{\pm 1.3}$ (-57.1) | $76.1_{\pm 5.7}$ (-23.9) |
| CMI w/o BN | $31.1_{\pm 3.2}$ (-59.7) | $27.4_{\pm 3.1}$ (+17.0) | $36.0_{\pm 2.5}$ (-55.1) | $14.7_{\pm 5.6}$ (-85.3) |
| NAYER w/ BN | $48.7_{\pm 1.1}$ (-42.1) | $10.4_{\pm 0.0}$ (+0.0) | $49.4_{\pm 2.6}$ (-41.7) | $97.9_{\pm 0.8}$ (-2.1) |
| NAYER w/o BN | $40.8_{\pm 2.9}$ (-50.0) | $29.3_{\pm 2.2}$ (+18.9) | $39.1_{\pm 3.5}$ (-52.0) | $8.0_{\pm 3.4}$ (-92.0) |

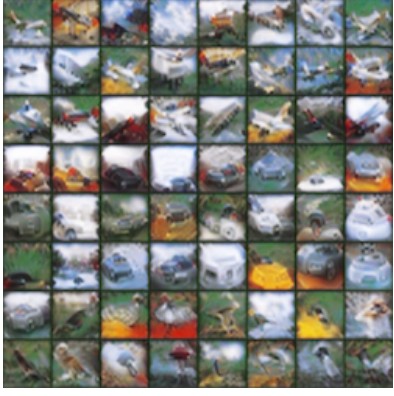 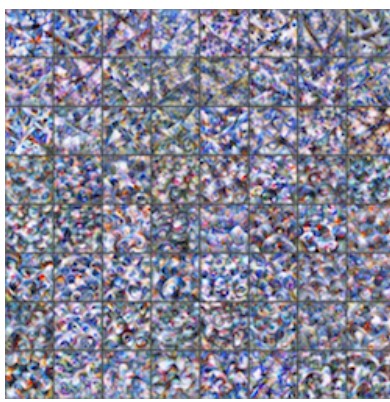

(a) CMI w/ BN Loss                                      (b) CMI w/o BN Loss

*Figure 13.* Synthetic samples of CMI (w/ and w/o BN Loss) on a **SL teacher** (datasets: CIFAR10; archs: ResNet34→ResNet18).

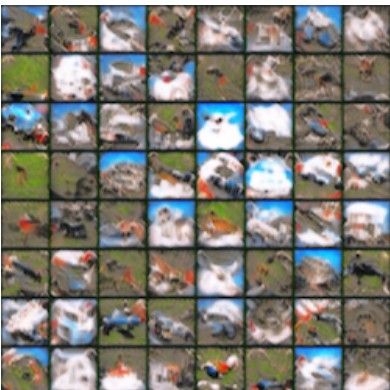 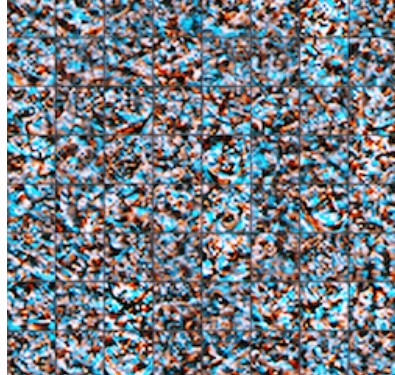

(a) NAYER w/ BN Loss                                      (b) NAYER w/o BN Loss

*Figure 14.* Synthetic samples of NAYER (w/ and w/o BN Loss) on a **SL teacher** (datasets: CIFAR10; archs: ResNet34→ResNet18).

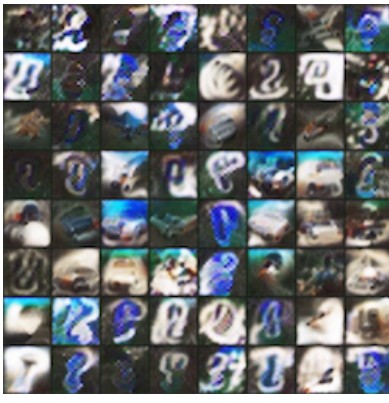 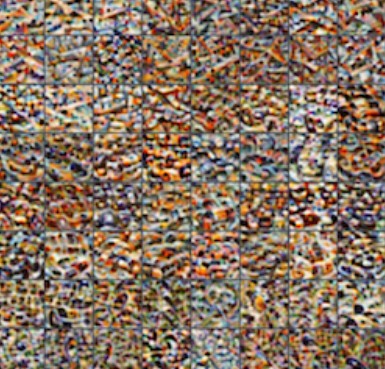

(a) CMI w/ BN Loss                                      (b) CMI w/o BN Loss

*Figure 15.* Synthetic samples of CMI (w/ and w/o BN Loss) on an **NTL teacher** (datasets: CIFAR10→Digits; ResNet34→ResNet18).

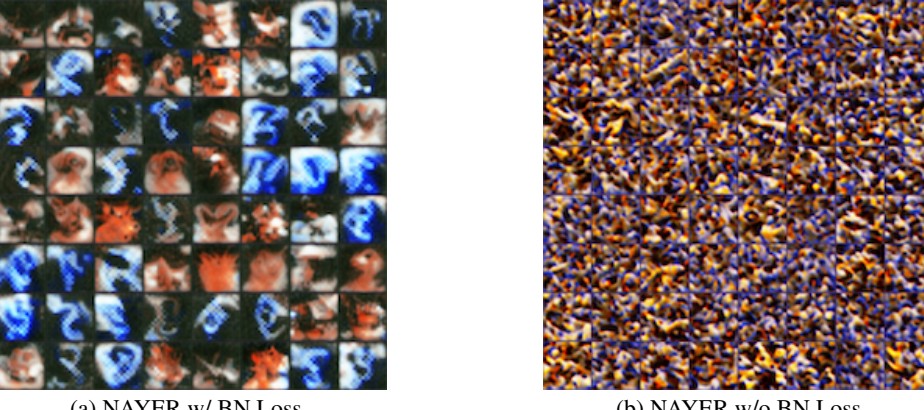

(a) NAYER w/ BN Loss                     (b) NAYER w/o BN Loss

*Figure 16.* Synthetic samples of NAYER (w/ and w/o BN Loss) on an **NTL teacher** (datasets: CIFAR10→Digits; ResNet34→ResNet18).

### C.5. Adversarial Robustness for NTL Teachers

In this section, we present more evidence related to the NTL teachers' adversarial robustness difference on real ID and OOD domains. Results across various datasets and network architectures are shown in Figures 17 to 22, where the accuracy reflects the consistency of predictive labels before and after untargeted PGD (Madry, 2017) attack. We can observe that predictive labels for ID domain data are easily changed after attacks. In contrast, OOD samples exhibit strong robustness against adversarial perturbations.

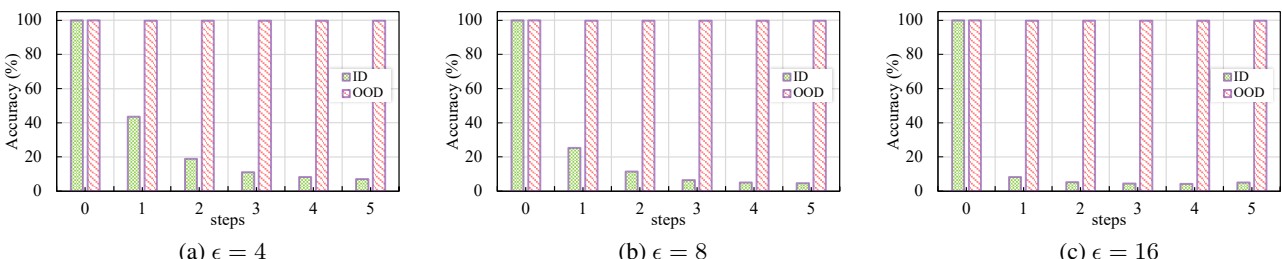

(a) $\epsilon = 4$              (b) $\epsilon = 8$              (c) $\epsilon = 16$

*Figure 17.* Adversarial robustness (PGD) on ID and OOD domain. The NTL teacher is trained on **SVHN→CIFAR10** with **ResNet34**.

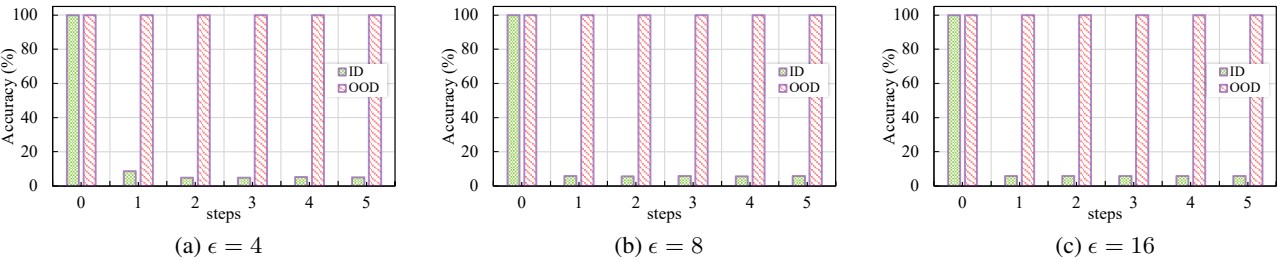

(a) $\epsilon = 4$              (b) $\epsilon = 8$              (c) $\epsilon = 16$

*Figure 18.* Adversarial robustness (PGD) on ID and OOD domain. The NTL teacher is trained on **SVHN→CIFAR10** with **VGG-13**.

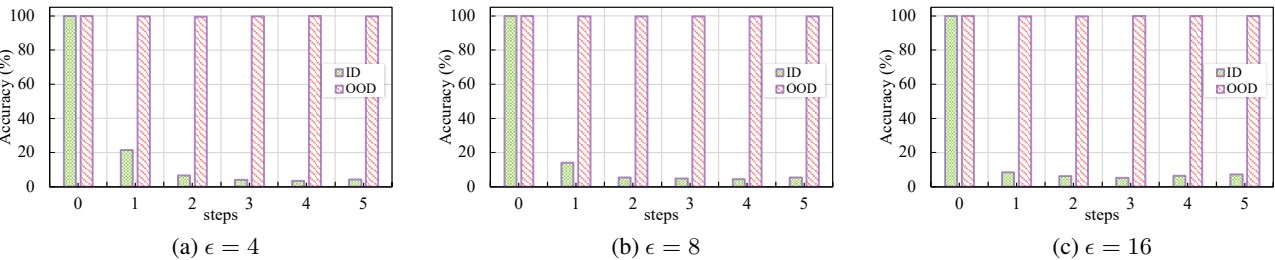

(a) $\epsilon = 4$              (b) $\epsilon = 8$              (c) $\epsilon = 16$

*Figure 19.* Adversarial robustness (PGD) on ID and OOD domain. The NTL teacher is trained on **CIFAR10→Digits** with **ResNet34**.

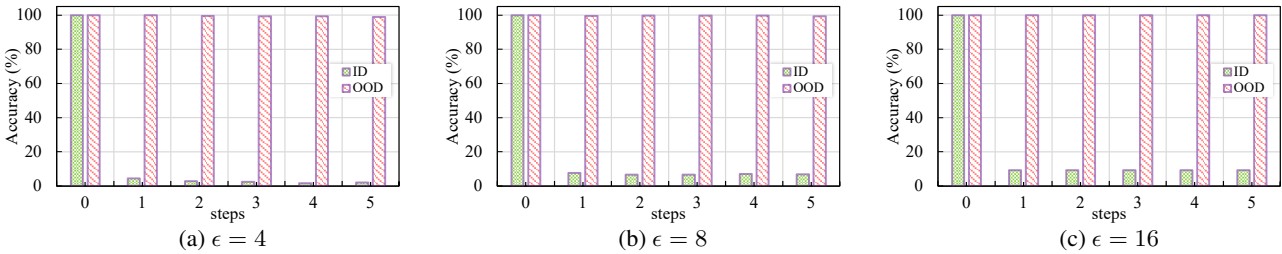

*Figure 20.* Adversarial robustness (PGD) on ID and OOD domain. The NTL teacher is trained on **CIFAR10→Digits** with **VGG-13**.

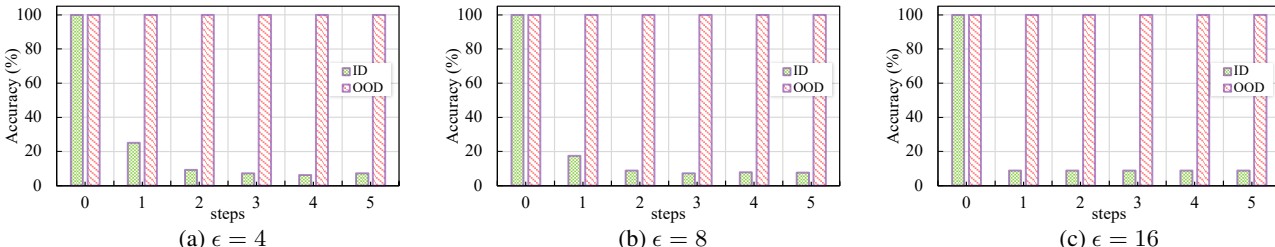

*Figure 21.* Adversarial robustness (PGD) on ID and OOD domain. The NTL teacher is trained on **CIFAR10→STL10** with **ResNet34**.

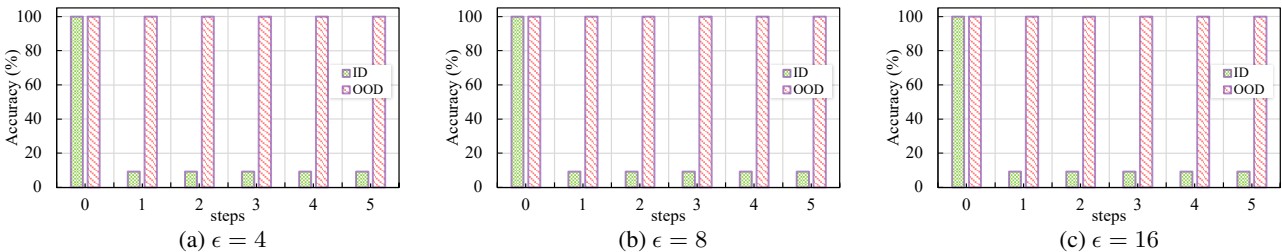

*Figure 22.* Adversarial robustness (PGD) on ID and OOD domain. The NTL teacher is trained on **CIFAR10→STL10** with **VGG-13**.

### C.6. The Influence of NTL Training Objectives to Adversarial Robustness

In this section, we further investigate the influence of each training objective in NTL to the adversarial robustness of NTL teachers. We conduct ablation studies on three variants: **(i)** the vanilla SL on ID domain: $\mathcal{L}_{id}$, **(ii)** the vanilla SL on ID domain and the class control on the OOD domain: $\mathcal{L}_{id} + \mathcal{L}_{cls}$, and **(iii)** the full NTL: $\mathcal{L}_{NTL\text{-}cls}$.

It is notably that the training object of **(ii)** $\mathcal{L}_{id} + \mathcal{L}_{cls}$ is similar to the conventional backdoor learning (Wu et al., 2022; Hong et al., 2023). Compared to **(ii)**, the **(iii)** full NTL $\mathcal{L}_{NTL\text{-}cls}$ has an additional term $(-\min(1, \alpha \cdot \mathcal{L}_{out} \cdot \mathcal{L}_{feat}))$. Therefore, the full NTL $\mathcal{L}_{NTL\text{-}cls}$ not only predicts all OOD-domain samples to the target class $y_{ood}$ (like conventional backdoor learning), but also pushes these OOD-domain samples far from ID-domain samples in both feature and output space.

The adversarial robustness of three NTL variants on ID and OOD domains are shown in Figures 23 to 26. We have:

- **(i)** The vanilla SL models are fragile on both ID and OOD domain. This is because learning correct classification results in relatively **complex decision boundaries** between classes and **small margins**[6] for ID domain data points.

- **(ii)** For the ID-domain SL with class control on the OOD domain (i.e., like conventional backdoor learning (Wu et al., 2022; Hong et al., 2023)), the OOD domain become more robust than ID domain. This is because minimizing $\mathcal{L}_{cls}$ forces all OOD-domain data to be predicted as a single class, which is simple, and **no boundaries** will go through the OOD-domain samples.

- **(iii)** For the full NTL, maximizing the additional term $(-\min(1, \alpha \cdot \mathcal{L}_{out} \cdot \mathcal{L}_{feat}))$ further pushes the OOD-domain cluster far away from ID-domain clusters, resulting in **very large margins** for OOD-domain samples (i.e., no OOD-domain samples will close to decision boundary). As a result, the full NTL teachers exhibit a very strong robustness against adversarial attacks on OOD-domain data.

---

[6]The **margin** is defined as the minimal distance from a data point to decision boundaries (Xu et al., 2023), where a larger margin corresponds to stronger robustness.

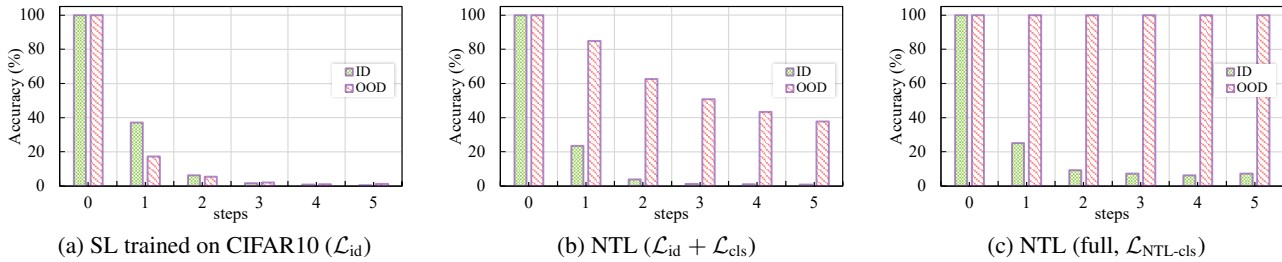

*Figure 23.* Comparison of adversarial robustness of SL and NTL teachers (ResNet-34) against untargeted PGD attack ($\epsilon = 4$) on close-set ID and OOD domain (ID: CIFAR10 and OOD: STL), where the accuracy reflects the consistency of predictive labels before and after untargeted adversarial attack. SL is trained on CIFAR10, NTL with different configurations are trained on ID and OOD domains.

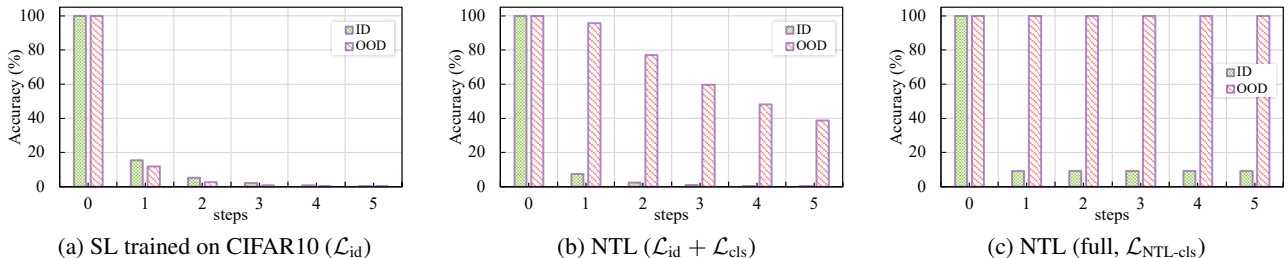

*Figure 24.* Comparison of adversarial robustness of SL and NTL teachers (VGG-13) against untargeted PGD attack ($\epsilon = 4$) on close-set ID and OOD domain (ID: CIFAR10 and OOD: STL), where the accuracy reflects the consistency of predictive labels before and after untargeted adversarial attack. SL is trained on CIFAR10, NTL with different configurations are trained on ID and OOD domains.

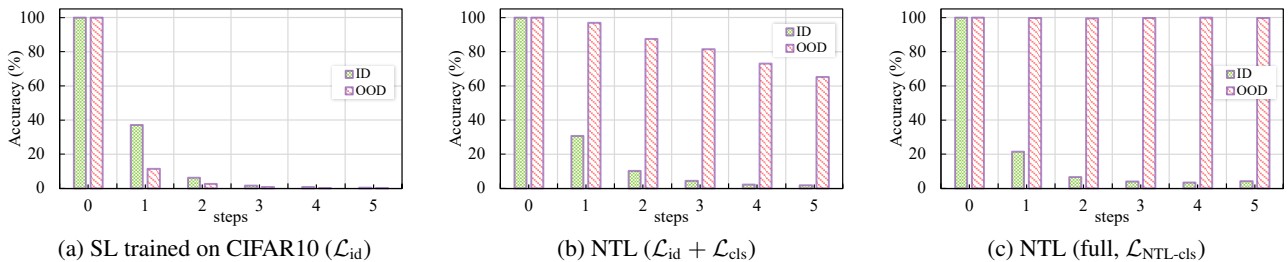

*Figure 25.* Comparison of adversarial robustness of SL and NTL teachers (ResNet-34) against untargeted PGD attack ($\epsilon = 4$) on open-set ID and OOD domain (ID: CIFAR10 and OOD: Digits), where the accuracy reflects the consistency of predictive labels before and after untargeted adversarial attack. SL is trained on CIFAR10, NTL with different configurations are trained on ID and OOD domains.

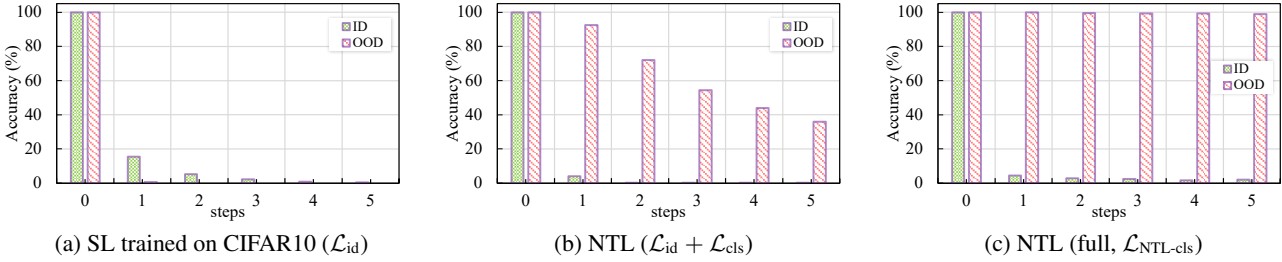

*Figure 26.* Comparison of adversarial robustness of SL and NTL teachers (VGG-13) against untargeted PGD attack ($\epsilon = 4$) on open-set ID and OOD domain (ID: CIFAR10 and OOD: Digits), where the accuracy reflects the consistency of predictive labels before and after untargeted adversarial attack. SL is trained on CIFAR10, NTL with different configurations are trained on ID and OOD domains.

## C.7. Adversarial Robustness for NTL-based Backdoor Teachers

We present the results of NTL-based backdoor teachers trained with the full NTL objective (i.e., $\mathcal{L}_{\text{NTL-cls}}$) in Figure 27. For comparison, we also include results for conventional backdoor teachers trained with $\mathcal{L}_{\text{id}} + \mathcal{L}_{\text{cls}}$ in Figure 28.

The results reveal similar observations discussed in Appendix C.5 and Appendix C.6. Specifically, both NTL-based and conventional backdoor teachers exhibit significantly stronger adversarial robustness on the OOD domain than on

the ID domain. Moreover, NTL-based backdoor teachers demonstrate even greater adversarial robustness compared to conventional backdoor teachers. This enhanced robustness is attributed to the influence of the NTL regularization term: $-\min(1, \alpha \cdot \mathcal{L}_{\text{out}} \cdot \mathcal{L}_{\text{feat}})$.

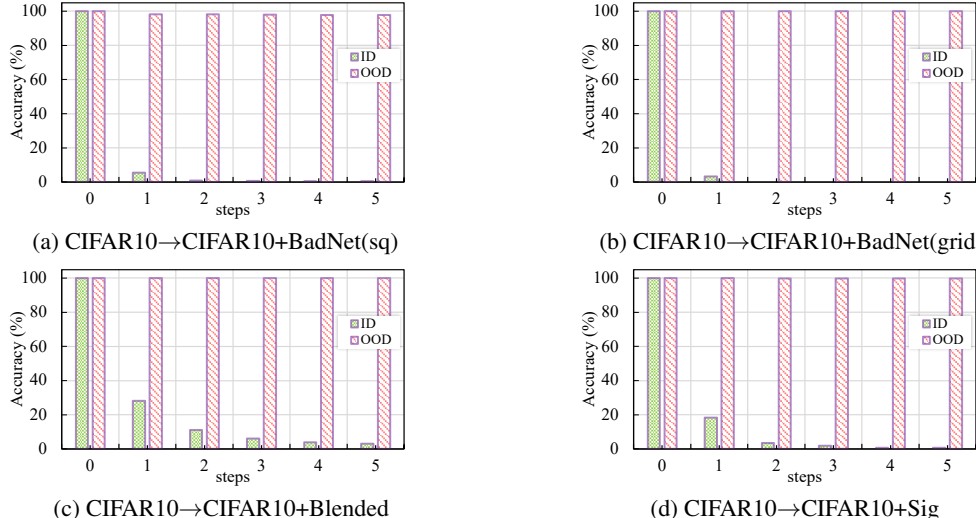

*Figure 27.* The difference of adversarial robustness (PGD with $\epsilon = 4$) on **NTL-based backdoored teachers**. The accuracy reflects the consistency of predictive labels before and after untargeted adversarial attack.

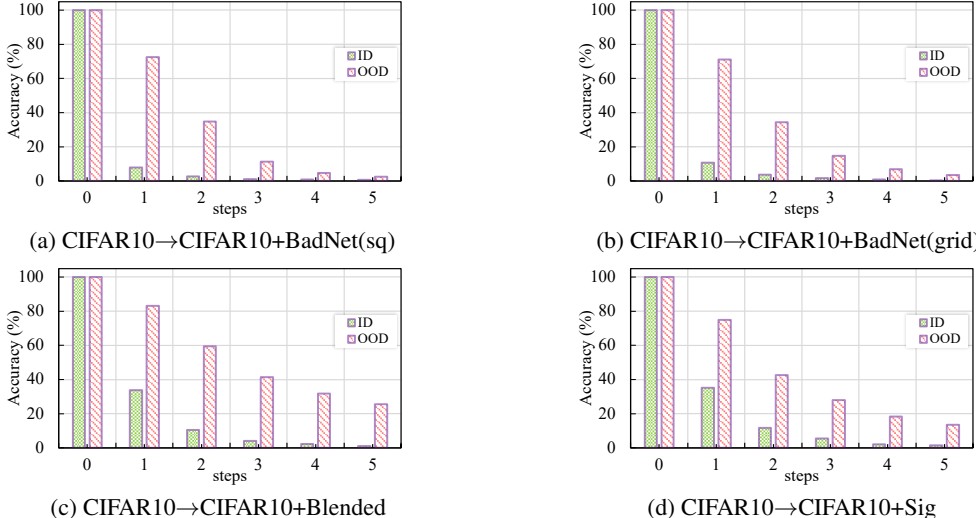

*Figure 28.* The difference of adversarial robustness (PGD with $\epsilon = 4$) on **conventional backdoor teachers**. The accuracy reflects the consistency of predictive labels before and after untargeted adversarial attack.

# D. Toy Experiments for OOD Trap Effect

## D.1. Experiments for Teachers with Batch-Normalization Layers

As shown in Figure 29, we use a toy experiment in 2D space to illustrate the OOD trap effect in NTL teachers. We consider a toy classification task with three classes and two domains with distribution shift (covariate shift). Specifically, we use different colors to denote different classes (red for class 1, green for class 2, and blue for class 3), and we use different symbols to represent different domains: ○ for ID domain and × for OOD domain.

We first train a **supervised learning (SL) teacher** on *ID domain* and conduct DFKD by using the adversarial exploration with BN loss (Equation 7 + Equation 9) to transfer knowledge from the SL teacher to a student model. The teacher and student share the same network architecture (**a three-layer MLP with BN layers**, as shown in Table 8). DFKD results are shown in Figure 30. The first line shows the decision space of the SL teacher, where we use colors corresponding to each class to represent their respective decision

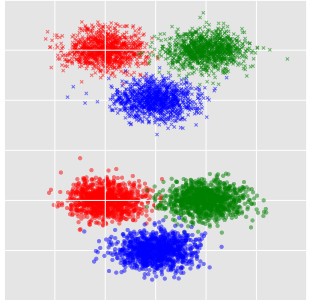

*Figure 29.* A toy experiment.

regions. The second line shows the decision space of the DFKD student. Those yellow crosses with black boundaries represent the synthesized data samples. From the results, the student model can learn correct ID knowledge within the SL teacher after several DFKD training epochs. Besides, the generalization ability from the ID to OOD domain is maintained.

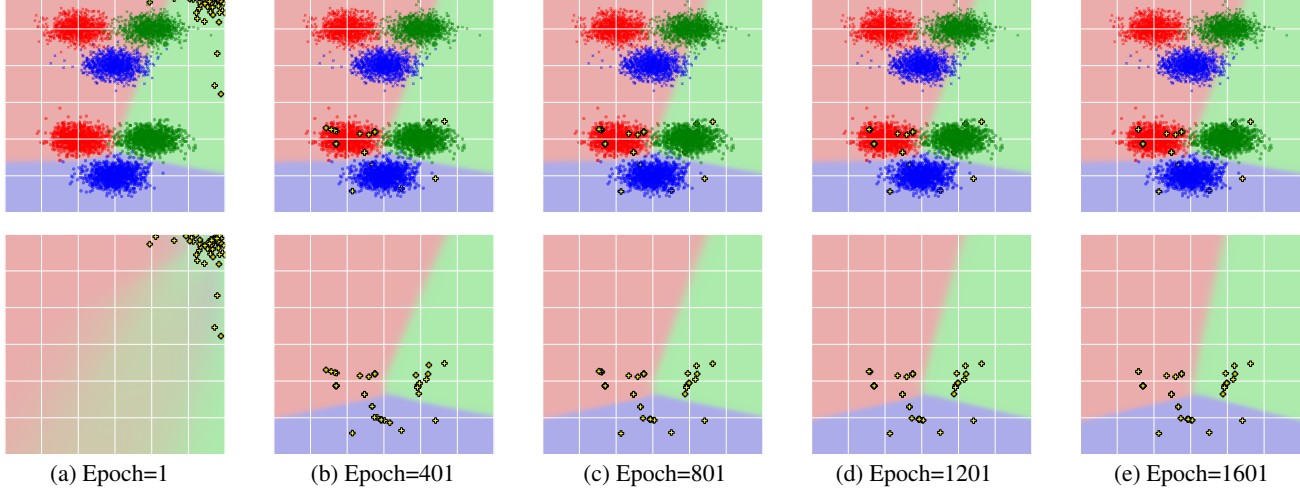

|          |            |            |             |             |
|:--------:|:----------:|:----------:|:-----------:|:-----------:|
| (a) Epoch=1 | (b) Epoch=401 | (c) Epoch=801 | (d) Epoch=1201 | (e) Epoch=1601 |

*Figure 30.* Toy experiment for **data-free knowledge distillation (w/ BN loss)** on a **SL teacher w/ BN layers**. Red represent class 1, green represent class 2, and blue represent class 3 samples. Different symbols represent different domains: ○ for ID domain and × for OOD domain. The first line shows the decision space of the SL teacher, and the second line shows the decision space of the DFKD student.

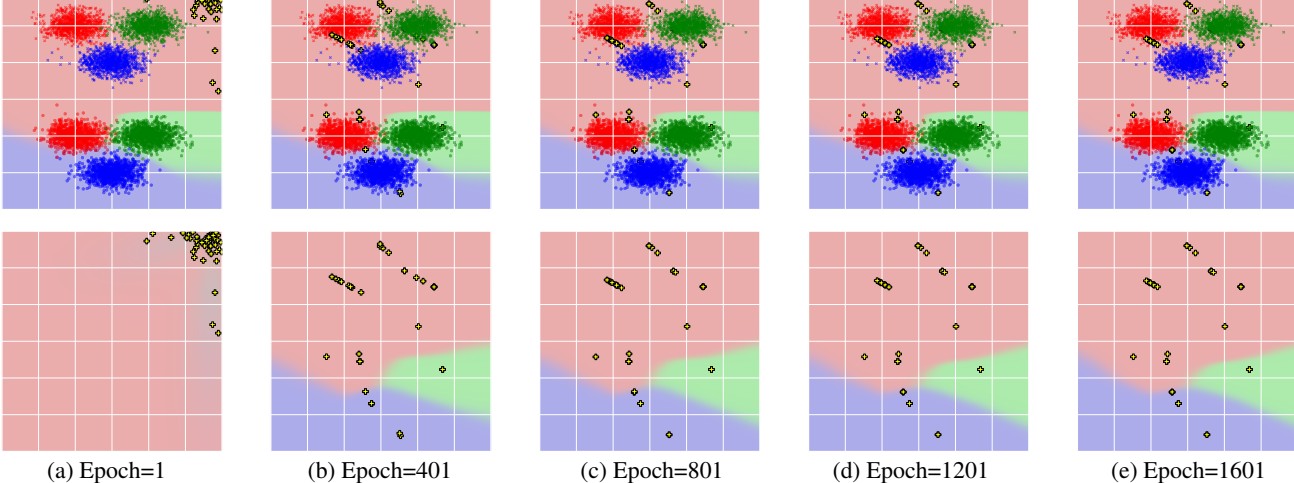

|          |            |            |             |             |
|:--------:|:----------:|:----------:|:-----------:|:-----------:|
| (a) Epoch=1 | (b) Epoch=401 | (c) Epoch=801 | (d) Epoch=1201 | (e) Epoch=1601 |

*Figure 31.* Toy experiment for **data-free knowledge distillation (w/ BN loss)** on an **NTL teacher w/ BN layers**.

Then we train a **non-transferable learning (NTL) teacher** on the ID domain and the OOD domain by minimizing Equation 6 and then conduct DFKD. Results are shown in Figure 31. Compared to the SL results, we can see that the NTL teacher let the DFKD student learning misleading knowledge from the OOD domain. This is caused by the fact that the DFKD generator synthesizing both ID-like and OOD-like samples (i.e., ID-to-OOD synthetic distribution shift).

**The effect of BN loss in DFKD.** To show the effect of the BN loss in DFKD, we conduct DFKD by only using the adversarial exploration loss (i.e., Equation 7). Specifically, we conduct DFKD three times with different initializations of the synthetic data. Results are shown in Figures 32 to 34. Intuitively, without the BN loss, the synthetic samples fail to approach the real distribution of both ID and OOD domains. Moreover, for some runs which initialization are far from the real ID domain (Figures 32 and 33), *the ID domain performance of the DFKD student is significantly degraded to random guess*. In such situations, no ID knowledge is successfully transferred to the DFKD student.

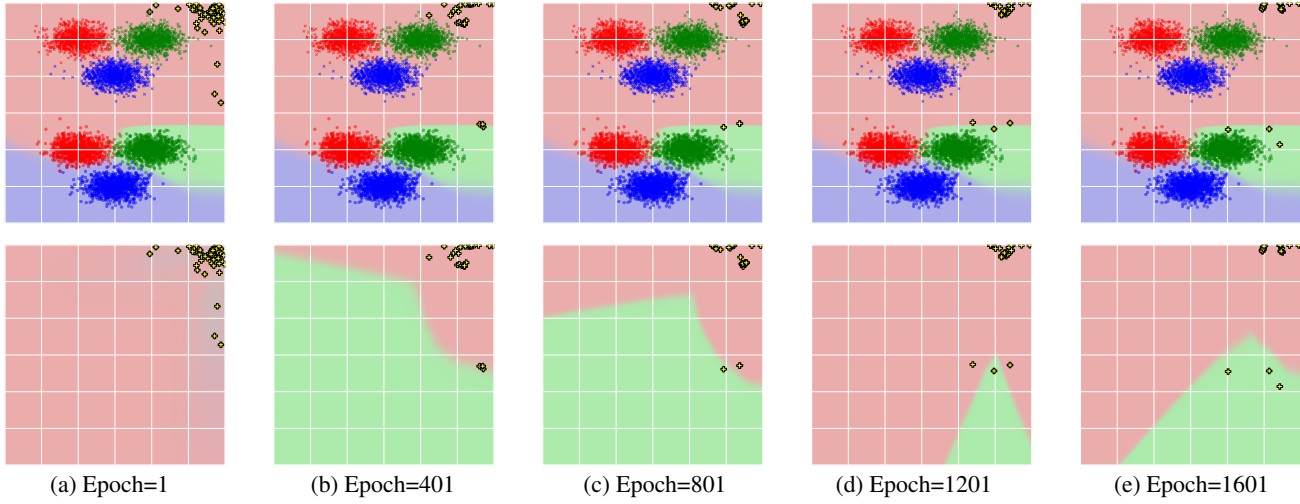

| (a) Epoch=1 | (b) Epoch=401 | (c) Epoch=801 | (d) Epoch=1201 | (e) Epoch=1601 |

*Figure 32.* Toy experiment for **data-free knowledge distillation (w/o BN loss)** on an **NTL teacher w/ BN layers** (initial 1).

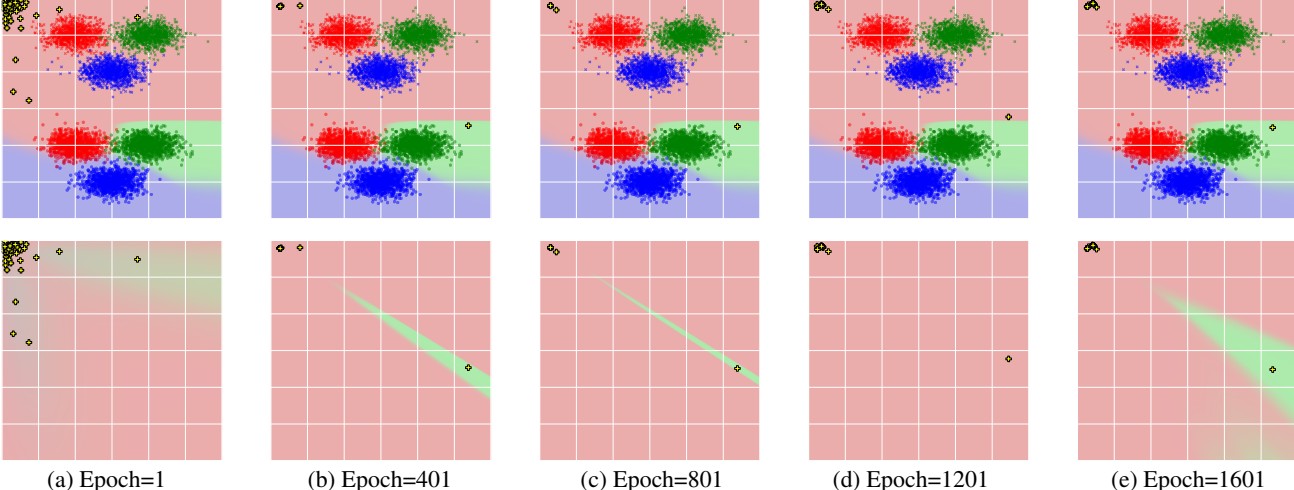

| (a) Epoch=1 | (b) Epoch=401 | (c) Epoch=801 | (d) Epoch=1201 | (e) Epoch=1601 |

*Figure 33.* Toy experiment for **data-free knowledge distillation (w/o BN loss)** on an **NTL teacher w/ BN layers** (initial 2).

**The effect of NTL components.** In addition, we conduct DFKD experiments on different variants of NTL teachers, thus verifying the effect of different component in NTL training objectives. We consider five variants:

- **NTL-cls**: the full NTL with class controlling, as shown in Equation 5.

- **NTL**: the full NTL without class controlling, as shown in Equation 4.

- **SL2domain**: revising **NTL-cls** by remaining the ID domain learning objective and the OOD class controlling term, which

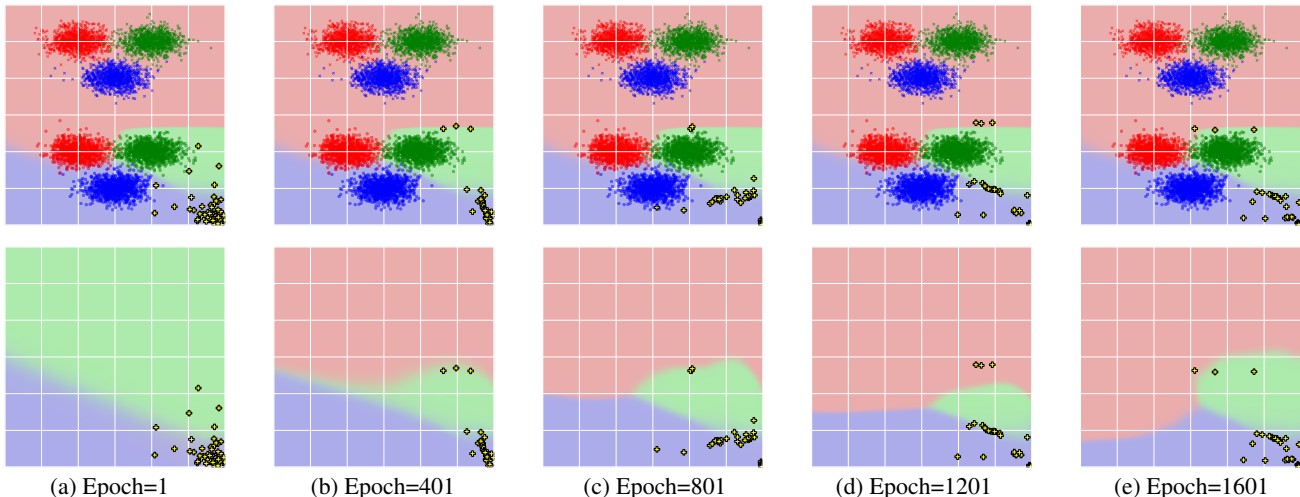

| (a) Epoch=1 | (b) Epoch=401 | (c) Epoch=801 | (d) Epoch=1201 | (e) Epoch=1601 |

*Figure 34.* Toy experiment for **data-free knowledge distillation (w/o BN loss)** on an **NTL teacher w/ BN layers** (initial 3).

can be formulated as:

$$\mathcal{L}_{\text{SL2domain}} = \mathcal{L}_{\text{id}} + \lambda_{\text{cls}} \cdot \mathcal{L}_{\text{cls}}. \tag{15}$$

• **NTL w/o MMD**: revising **NTL-cls** by removing the MMD loss in feature space, which can be formulated as:

$$\mathcal{L}_{\text{NTLw/oMMD}} = \mathcal{L}_{\text{id}} - \min(1, \alpha \cdot \mathcal{L}_{\text{out}}) + \lambda_{\text{cls}} \cdot \mathcal{L}_{\text{cls}}. \tag{16}$$

• **NTL w/o KL**: revising **NTL-cls** by removing the KL loss in output space, which can be formulated as:

$$\mathcal{L}_{\text{NTLw/oKL}} = \mathcal{L}_{\text{id}} - \min(1, \alpha \cdot \mathcal{L}_{\text{feat}}) + \lambda_{\text{cls}} \cdot \mathcal{L}_{\text{cls}}. \tag{17}$$

Visualization results of the SL teacher and NTL variants are shown in Figure 35. The corresponding accuracies on ID and OOD domains are shown in Table 6. From the results, all the five NTL variants can transfer OOD misleading knowledge from teacher to student. However, the full NTL has the most significant influence of the ID knowledge transfer (reflecting on the lowest IAcc). This highlights the importance of the combined effect of feature-level and output-level discrepancy terms in shaping the OOD trap effect.

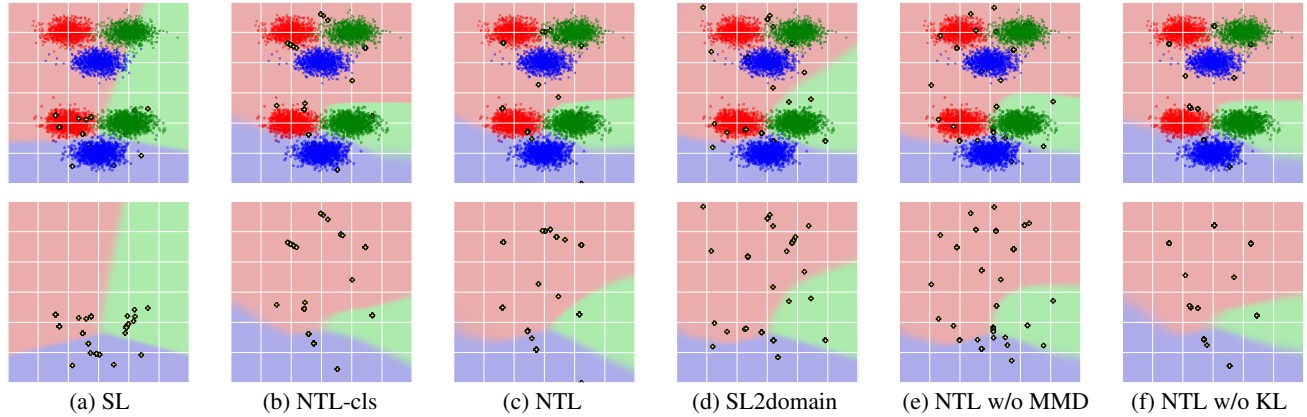

| (a) SL | (b) NTL-cls | (c) NTL | (d) SL2domain | (e) NTL w/o MMD | (f) NTL w/o KL |

*Figure 35.* Comparison of **DFKD w/ BN loss** across variants of **teachers w/ BN layers** (Epoch=1801).

*Table 6.* Results on variants of NTL teachers. We report the ID domain accuracy (IAcc) in blue and OOD domain accuracy (OAcc) in red.

| | SL | | NTL-cls | | NTL | | SL2domain | | NTL w/o MMD | | NTL w/o KL | |
|---|---|---|---|---|---|---|---|---|---|---|---|---|
| | IAcc ↑ | OAcc ↑ | IAcc ↑ | OAcc ↑ | IAcc ↑ | OAcc ↑ | IAcc ↑ | OAcc ↑ | IAcc ↑ | OAcc ↑ | IAcc ↑ | OAcc ↑ |
| Teacher | 99.33% | 45.44% | 98.67% | 34.78% | 99.00% | 34.78% | 99.22% | 34.78% | 99.33% | 34.78% | 98.56% | 34.78% |
| Student | 99.00% | 62.33% | 96.78% | 34.78% | 98.33% | 34.78% | 99.00% | 34.78% | 99.11% | 34.78% | 97.44% | 34.78% |

*Table 7.* The architecture of the toy model w/ BN.

| | |
|---|---|
| **Output** | nn.ReLU |
| | nn.Linear(500, 3) |
| **Hidden** | nn.ReLU |
| | nn.BatchNorm1d(500) |
| | nn.Linear(100, 500) |
| **Input** | nn.ReLU |
| | nn.BatchNorm1d(100) |
| | nn.Linear(2, 100) |

*Table 8.* The architecture of the toy model w/o BN.

| | |
|---|---|
| **Output** | nn.ReLU |
| | nn.Linear(500, 3) |
| **Hidden** | nn.ReLU |
| | nn.Linear(100, 500) |
| **Input** | nn.ReLU |
| | nn.Linear(2, 100) |

## D.2. Experiments for Teachers without Batch-Normalization Layers

We also run the toy experiments by replacing the network architecture from the three-layer MLP with BN layers (Table 7) to a **three-layer MLP without BN layers**, as shown in Table 8. We also train a SL teacher and an NTL teacher, and then, we conduct DFKD by using the **adversarial exploration loss** (Equation 7) individually. The results for SL teacher is shown in Figure 36, and the results for NTL teacher (three runs with different initialization) are shown in Figures 37 to 39.

From the results, although without the BN regularization, the student trained by the adversarial exploration can still learn correct ID knowledge from the SL teacher, with the generalization ability from the ID to OOD domain being maintained. In contrast, when facing the NTL teacher, the individual adversarial exploration still lead to an unexpectedly poor ID performance (random guess in the worst case, like Figures 37 and 38).

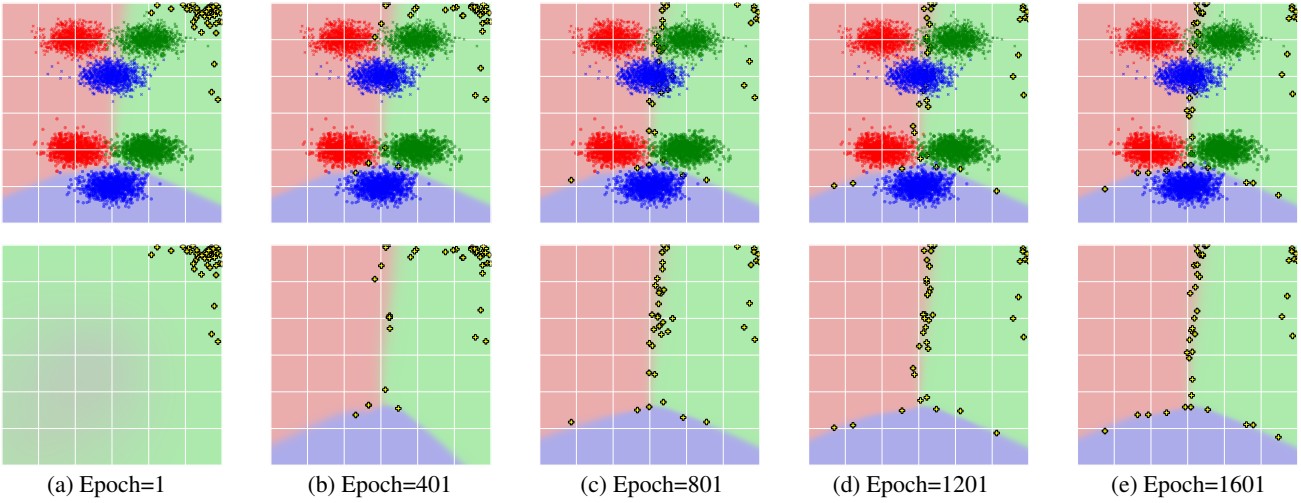

| (a) Epoch=1 | (b) Epoch=401 | (c) Epoch=801 | (d) Epoch=1201 | (e) Epoch=1601 |

*Figure 36.* Toy experiment for **data-free knowledge distillation (w/o BN loss)** on a **SL teacher w/o BN layers**. Red represent class 1, green represent class 2, and blue represent class 3 samples. Different symbols represent different domains: ○ for ID domain and × for OOD domain. The first line shows the decision space of the SL teacher, and the second line shows the decision space of the student.

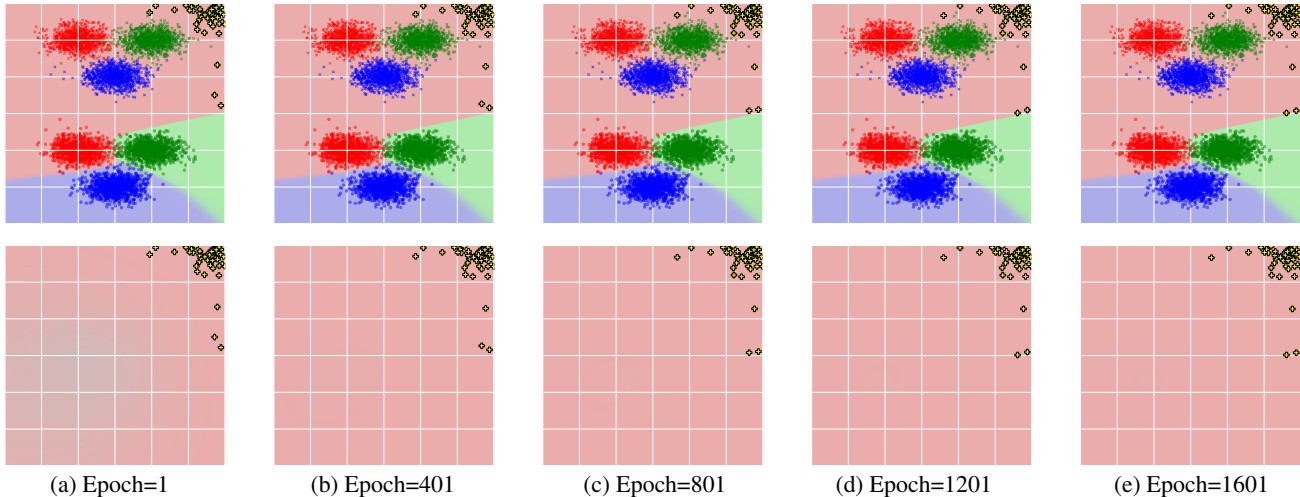

| (a) Epoch=1 | (b) Epoch=401 | (c) Epoch=801 | (d) Epoch=1201 | (e) Epoch=1601 |

*Figure 37.* Toy experiment for **data-free knowledge distillation (w/o BN loss)** on an **NTL teacher w/o BN layers** (initial 1).

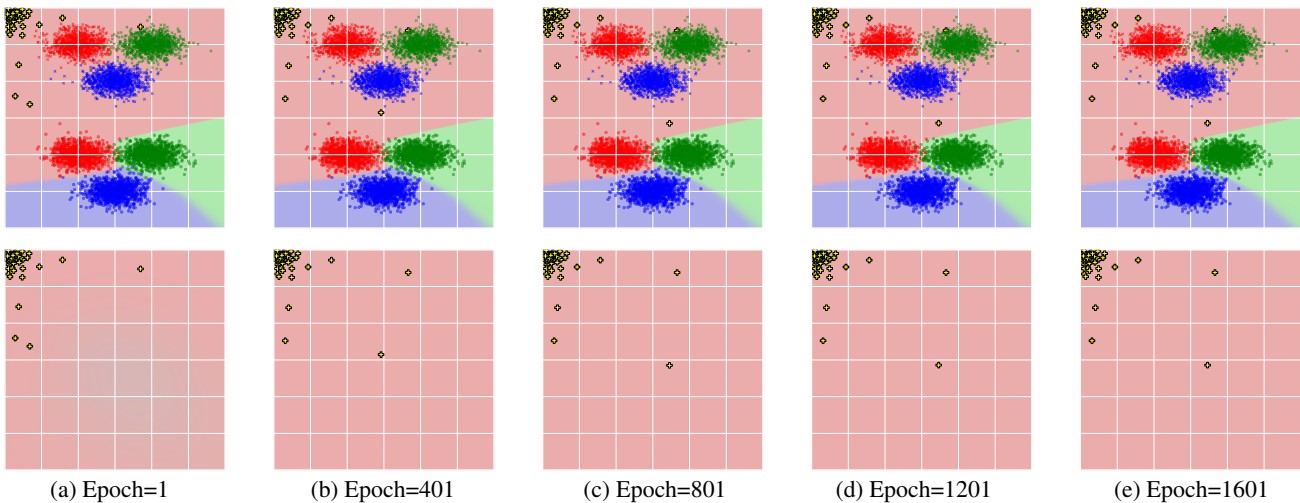

| (a) Epoch=1 | (b) Epoch=401 | (c) Epoch=801 | (d) Epoch=1201 | (e) Epoch=1601 |

*Figure 38.* Toy experiment for **data-free knowledge distillation (w/o BN loss)** on an **NTL teacher w/o BN layers** (initial 2).

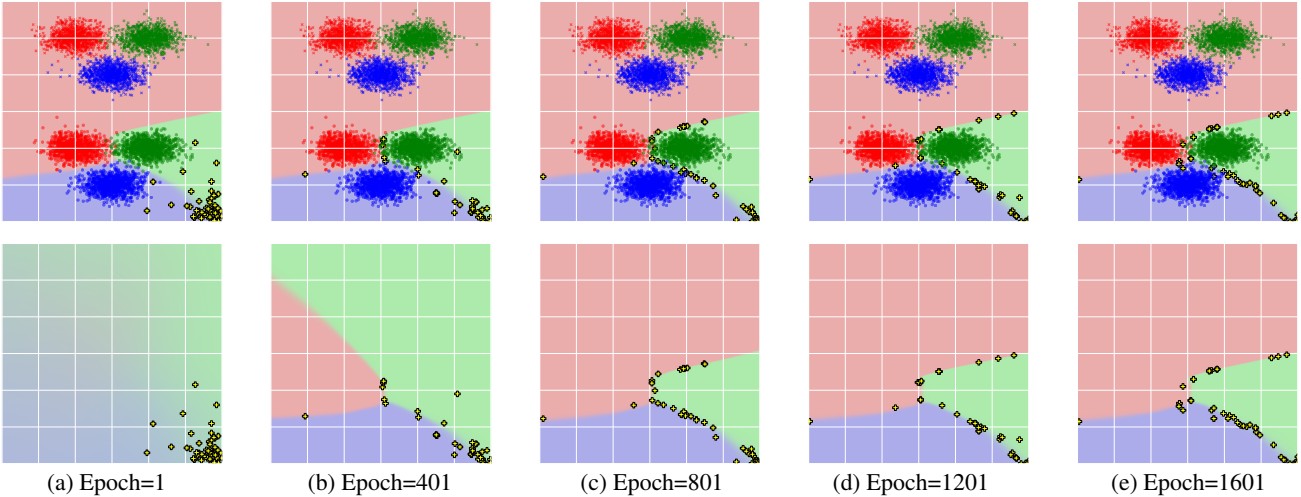

| (a) Epoch=1 | (b) Epoch=401 | (c) Epoch=801 | (d) Epoch=1201 | (e) Epoch=1601 |

*Figure 39.* Toy experiment for **data-free knowledge distillation (w/o BN loss)** on an **NTL teacher w/o BN layers** (initial 3).

**The effect of NTL components.** In addition, we also investigate the effect of different variants of NTL teachers (the same as variants in Appendix D.1). Visualization results of each NTL variant and the SL teacher are shown in Figure 40. The corresponding accuracies on ID and OOD domains are shown in Table 9.

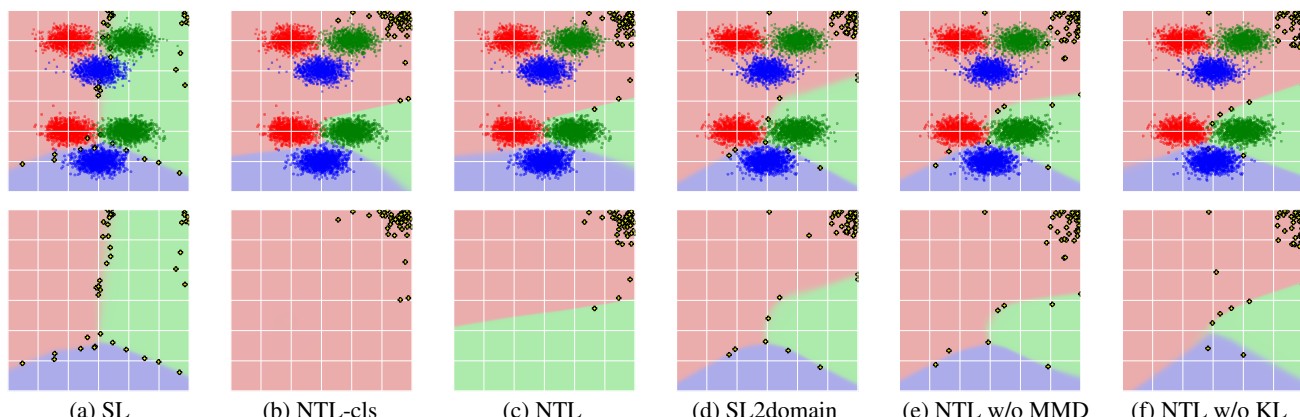

| (a) SL | (b) NTL-cls | (c) NTL | (d) SL2domain | (e) NTL w/o MMD | (f) NTL w/o KL |

*Figure 40.* Comparison of **DFKD w/o BN loss** across variants of **teachers w/o BN layers** (Epoch=1801).

*Table 9.* Results on variants of NTL teachers. We report the ID domain accuracy (IAcc) in blue and OOD domain accuracy (OAcc) in red.

|  | SL | | NTL-cls | | NTL | | SL2domain | | NTL w/o MMD | | NTL w/o KL | |
|---|---|---|---|---|---|---|---|---|---|---|---|---|
|  | IAcc ↑ | OAcc ↑ | IAcc ↑ | OAcc ↑ | IAcc ↑ | OAcc ↑ | IAcc ↑ | OAcc ↑ | IAcc ↑ | OAcc ↑ | IAcc ↑ | OAcc ↑ |
| Teacher | 99.44% | 65.67% | 99.22% | 34.78% | 99.11% | 34.78% | 99.67% | 34.78% | 99.56% | 34.78% | 99.56% | 34.78% |
| Student | 99.56% | 65.67% | 34.78% | 34.78% | 32.67% | 34.78% | 99.67% | 34.78% | 99.67% | 34.78% | 99.33% | 34.78% |

Similarly, all five NTL variants can transfer OOD misleading knowledge to students. Notably, compared to the results of teachers with BN layers (Figure 35 and Table 9), when the teacher lacks BN layers, the full NTL can completely block ID knowledge transfer, leading the student to exhibit random guess-like performance on the ID domain.

# E. Additional Experiments

In this section, we present more experimental results and analysis. We provide more detailed experimental setups in Appendix E.1. In Appendix E.2, we analyze the influence of adversarial perturbation bound to the synthetic data grouping. In Appendix E.3, we analyze the influence of forgetting loss. In Appendix E.4, we conduct ATEsc by using alternative adversarial attack. In Appendix E.5, we compare our proposed ATEsc with more baselines. In Appendix E.7, we plot visualization regarding the feature distribution and synthetic samples.

### E.1. Experiment Setup and Details

**Datasets.** We choose the common datasets shared by both NTL and DFKD, including Digits (MNIST (Deng, 2012), USPS (Hull, 1994), SVHN (Netzer et al., 2011), MNIST-M (Ganin et al., 2016), and SYN-D (Roy et al., 2018)), CIFAR10 (Krizhevsky et al., 2009), STL10 (Coates et al., 2011).

**NTL tasks.** For widely evaluation, we conduct our experiments on three different NTL tasks:

- **(i) Close-set NTL**. In this setting, the ID domain and OOD domain have the same label space, but the two domains have a distribution shift. We choose the dataset pairs of: SVHN→MNIST-M and CIFAR10→STL10;

- **(ii) Open-set NTL**. In this setting, the label space of ID domain and OOD domain are disjoint. We perform experiments on SVHN→CIFAR10 and CIFAR10→Digits, where Digits is the combination of 5 digits datasets: MNIST, USPS, SVHN, MNIST-M, and SYN-D;

- **(iii) NTL-based Backdoor**. We additionally consider using NTL in conducting training controllable backdoor attack (Wu et al., 2022). We use four triggers: BadNets(sq), BadNets(grid) (Gu et al., 2019), Blended (Chen et al., 2017), Sig (Barni et al., 2019) on CIFAR10 and see them as the OOD domain. The clean CIFAR10 is regarded as the ID domain.

Our aim is to address the malign OOD-Trap effect introduced by NTL teachers in DFKD, and thus, regardless of NTL teacher's tasks, our goal is to transfer only the ID domain knowledge from NTL teacher to student.

**DFKD baselines.** We involve the state-of-the-art (SOTA) non-memory bank method DFQ (Choi et al., 2020) and two memory bank methods: CMI (Fang et al., 2021b), NAYER (Tran et al., 2024). We follow the implementation of DFQ and CMI in `https://github.com/zju-vipa/CMI` and NAYER in `https://github.com/tmtuan1307/NAYER`. For each baseline, we evaluate two variants of our method: using only the CKD in Section 4.3 (denoted as + CKD) and the full method (denoted as + ATEsc). All teachers are considered as white-box models during DFKD.

**Implementation details.** For each task, NTL teachers are pre-trained up to 50 epochs (input images resize to 64×64). For DFKD, we train each method up to 200 epochs, with all hyper-parameters following their original implementations. VGG-13 (Simonyan, 2014) and ResNet-34 (He et al., 2016) are used as teacher architectures, while VGG-11 and ResNet-18 are as student architectures. We conduct all experiments on a single RTX 4090 GPU (24G).

**Evaluation metrics.** For **Close-set NTL**, we use ID domain accuracy (IAcc) to measure the performance of ID knowledge transfer. Because ID and OOD domains have the same label space in close-set NTL, we present OOD domain accuracy (OAcc) to their real labels to show the generalization ability from ID to OOD domain. Higher OAcc means better ID-to-OOD generalization, and also means better resistance to misleading OOD knowledge transfer during DFKD. For **Open-set NTL**, we calculate the accuracy between the prediction and the OOD-label on OOD domain (denoted as OLAcc) to evaluate the transfer of misleading OOD knowledge. A lower OLAcc indicates better resistance to OOD knowledge transfer. Moreover, we still use IAcc on ID domains. For **NTL-based Backdoor**, we follow backdoor learning to use clean label accuracy on ID domain (i.e., the same to IAcc) and attack success rate (ASR) (Li et al., 2022) on backdoor data, where the ASR is the accuracy of the prediction and the OOD-label (i.e., the backdoor target label) on trigger OOD data.

### E.2. Analysis of Adversarial Perturbation Bound

To evaluate the influence of the adversarial perturbation bound $\epsilon$, we conduct experiments on CIFAR10→STL10 and CIFAR10→Digits using two DFKD baselines: DFQ and NAYER. The results are presented in Figure 41 and Figure 42. For the CIFAR10→STL10 task (Figure 41), we observe that the performance are generally stable, except for DFQ + ATEsc. As $\epsilon$ decreases, the student's OAcc in DFQ + ATEsc experiences a significant drop. This suggests that an overly small perturbation bound is ineffective in distinguishing ID-like and OOD-like synthetic samples, allowing the transfer of misleading OOD knowledge from the teacher to the student. For the open-set task CIFAR10→Digits (Figure 42), a

higher OLAcc indicates a more severe transfer of misleading OOD knowledge. We observe a general trend: increasing the perturbation bound $\epsilon$ improves resistance to the transfer of misleading OOD knowledge from the teacher to the student.

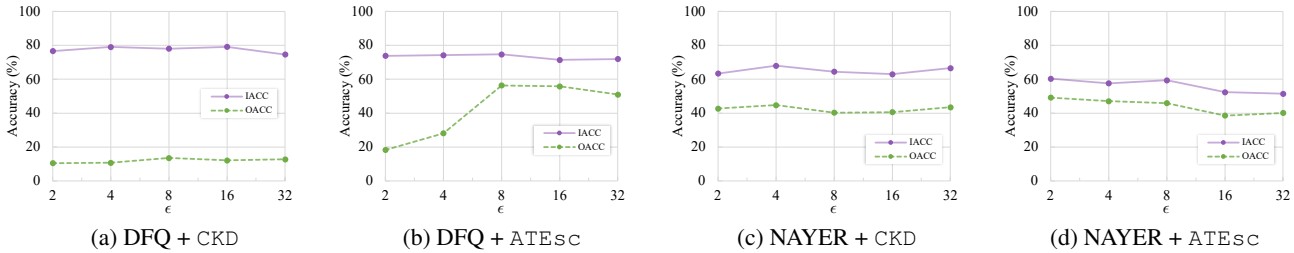

*Figure 41.* Hyperparamater analysis of $\epsilon$. CIFAR10→STL10. $T$: ResNet34 and $S$: ResNet18.

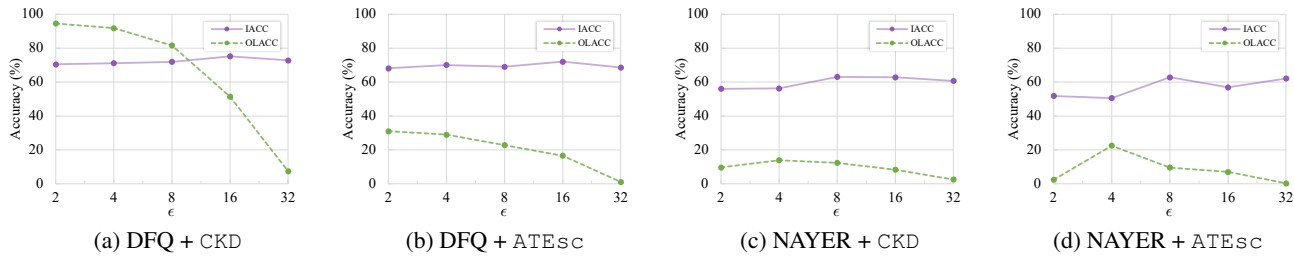

*Figure 42.* Hyperparamater analysis of $\epsilon$. CIFAR10→Digits. $T$: ResNet34 and $S$: ResNet18.

### E.3. Analysis of Forgetting Loss

To analyze the effect of forgetting loss, we change the weight of $\lambda$ in Equation 14 from $10^{-7}$ to $10^{-2}$ and conduct experiments on a close-set task CIFAR10→STL10 and an open-set task CIFAR10→Digits with DFQ and NAYER serving as DFKD baselines. The results are shown in Figure 43.

For the close-set CIFAR10→STL10 (Figure 43 (a)), higher OAcc indicates better resistance to OOD knowledge transfer. We observe that setting $\lambda$ either too high or too low results in poor performance for DFQ on both ID and OOD domain, whereas NAYER exhibits greater robustness to changes in $\lambda$.

For the open-set task CIFAR10→Digits (Figure 43 (b)), a lower OLAcc indicates better performance. We observe that both DFQ + ATEsc and NAYER + ATEsc demonstrate robustness to variations in $\lambda$, exhibiting strong resistance to OOD knowledge transfer. However, setting a too large $\lambda$ can degrade ID performance in DFQ.

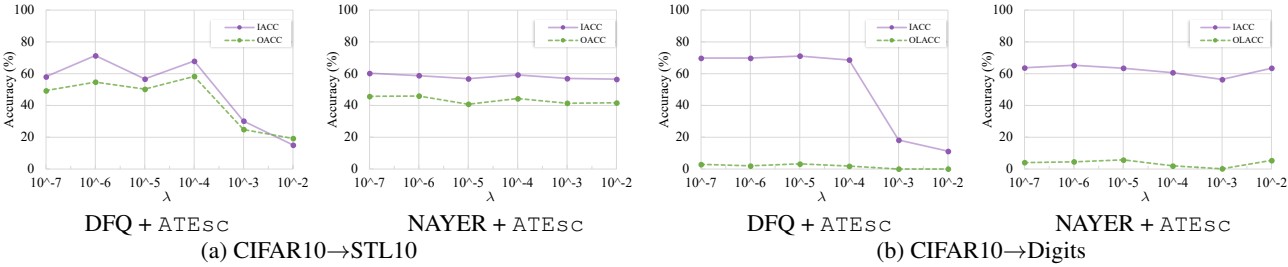

*Figure 43.* Hyperparamater analysis of $\lambda$. $T$: ResNet34 and $S$: ResNet18.

### E.4. Analysis of Adversarial Attack Method

The PGD (Madry, 2017) attack in Equation 10 can be replaced with any adversarial attack method. In this section, we use FGSM (Goodfellow et al., 2014) to perform adversarial attack, thus obtaining fragile and robust groups, and compare the final results with PGD. As shown in Table 10, we compare the results of CKD and the full ATEsc on close-set NTL tasks and the open set NTL tasks, with DFQ (Choi et al., 2020) and NAYER (Tran et al., 2024) serving as DFKD baselines. We find that by replacing PGD with FGSM, the results generally do not exhibit a significant difference, demonstrating the broad compatibility of our method with different adversarial attack methods.

*Table 10.* Different adversarial attack. We report the ID domain accuracy in blue and OOD domain accuracy in red. Results from the DFKD baselines are highlighted with gray rows. The accuracy drop compared to the pre-trained model is shown in brackets.

| | Close-set NTL | | | | Open-set NTL | | | |
|---|---|---|---|---|---|---|---|---|
| | ID: **CIFAR10** OOD: **STL** | | ID: **CIFAR10** OOD: **STL** | | ID: **CIFAR10** OOD: **Digits** | | ID: **CIFAR10** OOD: **Digits** | |
| | **VGG-13→VGG-11** | | **ResNet-34→ResNet-18** | | **VGG-13→VGG-11** | | **ResNet-34→ResNet-18** | |
| | IAcc (%) ↑ | OACC (%) ↑ | IAcc (%) ↑ | OACC (%) ↑ | IAcc (%) ↑ | OLACC (%) ↓ | IAcc (%) ↑ | OLACC (%) ↓ |
| NTL Teacher | $92.7_{\pm0.1}$ | $10.6_{\pm0.1}$ | $90.8_{\pm0.1}$ | $10.4_{\pm0.0}$ | $92.9_{\pm0.0}$ | $99.5_{\pm0.3}$ | $91.1_{\pm0.1}$ | $100.0_{\pm0.0}$ |
| DFQ | $42.7_{\pm3.6}$ (-50.0) | $10.4_{\pm0.0}$ (-0.2) | $52.5_{\pm9.5}$ (-38.3) | $11.8_{\pm1.3}$ (+1.4) | $62.8_{\pm2.3}$ (-30.1) | $99.8_{\pm0.1}$ (+0.3) | $57.4_{\pm8.8}$ (-33.7) | $85.8_{\pm8.6}$ (-14.2) |
| + CKD$_{PGD}$ | $63.0_{\pm3.3}$ (-29.7) | $10.4_{\pm0.0}$ (-0.2) | $68.0_{\pm10.1}$ (-22.8) | $22.5_{\pm7.3}$ (+12.1) | $68.5_{\pm1.4}$ (-24.4) | $98.6_{\pm0.4}$ (-0.9) | $70.1_{\pm4.0}$ (-21.0) | $5.5_{\pm3.6}$ (-94.5) |
| + CKD$_{FGSM}$ | $63.5_{\pm5.5}$ (-29.2) | $10.4_{\pm0.0}$ (-0.2) | $67.1_{\pm14.8}$ (-23.7) | $19.3_{\pm5.2}$ (+8.9) | $66.0_{\pm2.6}$ (-26.9) | $99.3_{\pm0.6}$ (-0.2) | $58.1_{\pm14.9}$ (-33.0) | $4.6_{\pm6.0}$ (-95.4) |
| + ATEsc$_{PGD}$ | $62.3_{\pm2.6}$ (-30.4) | $31.5_{\pm1.6}$ (+20.9) | $66.2_{\pm0.7}$ (-24.6) | $51.6_{\pm6.5}$ (+41.2) | $60.3_{\pm4.5}$ (-32.6) | $78.4_{\pm20.9}$ (-21.1) | $65.1_{\pm7.9}$ (-26.0) | $2.5_{\pm2.9}$ (-97.5) |
| + ATEsc$_{FGSM}$ | $59.2_{\pm3.4}$ (-33.5) | $27.9_{\pm9.7}$ (+17.3) | $53.3_{\pm5.1}$ (-37.5) | $47.0_{\pm9.7}$ (+36.6) | $60.8_{\pm3.4}$ (-32.1) | $80.1_{\pm33.6}$ (-19.4) | $52.1_{\pm13.8}$ (-39.0) | $0.9_{\pm0.1}$ (-99.1) |
| NAYER | $10.6_{\pm0.0}$ (-82.1) | $10.4_{\pm0.0}$ (-0.2) | $48.7_{\pm1.1}$ (-42.1) | $10.4_{\pm0.0}$ (-0.0) | $10.6_{\pm0.0}$ (-82.3) | $100.0_{\pm0.0}$ (+0.5) | $49.4_{\pm2.6}$ (-41.7) | $97.9_{\pm0.8}$ (-2.1) |
| + CKD$_{PGD}$ | $62.7_{\pm6.1}$ (-30.0) | $23.7_{\pm1.1}$ (+13.1) | $64.4_{\pm0.8}$ (-26.4) | $40.2_{\pm3.9}$ (+29.8) | $70.4_{\pm1.4}$ (-22.5) | $71.6_{\pm10.9}$ (-27.9) | $61.2_{\pm3.9}$ (-29.9) | $6.3_{\pm1.3}$ (-93.7) |
| + CKD$_{FGSM}$ | $65.3_{\pm3.1}$ (-27.4) | $24.2_{\pm9.9}$ (+13.6) | $64.7_{\pm2.0}$ (-26.1) | $40.9_{\pm4.0}$ (+30.5) | $71.4_{\pm1.6}$ (-21.5) | $74.4_{\pm10.1}$ (-25.1) | $63.6_{\pm4.6}$ (-27.5) | $1.6_{\pm0.7}$ (-98.4) |
| + ATEsc$_{PGD}$ | $69.7_{\pm5.4}$ (-23.0) | $42.3_{\pm2.7}$ (+31.7) | $52.7_{\pm2.4}$ (-38.1) | $39.2_{\pm1.3}$ (+28.8) | $69.8_{\pm1.9}$ (-23.1) | $2.8_{\pm3.8}$ (-96.7) | $59.0_{\pm3.0}$ (-32.1) | $3.6_{\pm0.6}$ (-96.4) |
| + ATEsc$_{FGSM}$ | $66.7_{\pm13.1}$ (-26.0) | $40.9_{\pm9.7}$ (+30.3) | $50.9_{\pm7.2}$ (-39.9) | $39.3_{\pm4.0}$ (+28.9) | $67.6_{\pm6.0}$ (-25.3) | $3.3_{\pm2.6}$ (-96.2) | $63.0_{\pm2.0}$ (-28.1) | $0.6_{\pm0.2}$ (-99.4) |

## E.5. Compare with Backdoor Baseline

Hong et al. (2023) first identified the risk of backdoor transfer from teacher to student via DFKD. To mitigate this risk, they proposed a defense method ABD, including Shuffling Vaccine (SV) and Self-Retrospection (SR) modules. Briefly, the SV mitigates the backdoor participating in the knowledge distillation stage by regularizing the generator, and SR synthesizes the potential backdoor and unlearns them. We compare our ATEsc with the ABD on close-set, open-set and NTL-based backdoor tasks. Specifically, we use CMI (Fang et al., 2021b) as DFKD baseline, which was also used in ABD's experiments. The implementation of ABD is follow: `https://github.com/illidanlab/ABD`. We consider two configurations of ABD, i.e, CMI + ABD (SV), and CMI + ABD (SV+SR).

The results are shown in Table 11. Although ABD is effective in defending conventional backdoor learning teachers, as reported in their paper, it struggles to address the OOD trap effect introduced by NTL teachers. This leads to the poor performance of using CMI + ABD to distilling close-set NTL teachers, open-set NTL teachers, and defend the NTL-based backdoor transfer during the DFKD. In contrast, our ATEsc can better mitigate OOD trap effect in NTL teachers, enhancing ID knowledge transfer while resisting OOD knowledge transfer.

*Table 11.* Compare to ABD. We report the ID domain accuracy in blue and OOD domain accuracy in red. Results from the DFKD baselines are highlighted with gray rows. The accuracy drop compared to the pre-trained model is shown in brackets.

| | Close-set NTL | | Open-set NTL | | NTL-based backdoor | | | |
|---|---|---|---|---|---|---|---|---|
| | ID: **CIFAR10** OOD: **STL** | | ID: **CIFAR10** OOD: **Digits** | | ID: **CIFAR10** OOD: **Blended** | | ID: **CIFAR10** OOD: **Sig** | |
| | IAcc (%) ↑ | OACC (%) ↑ | IAcc (%) ↑ | OLACC (%) ↓ | IAcc (%) ↑ | ASR (%) ↓ | IAcc (%) ↑ | ASR (%) ↓ |
| NTL Teacher | $90.8_{\pm0.1}$ | $10.4_{\pm0.0}$ | $91.1_{\pm0.1}$ | $100.0_{\pm0.0}$ | $90.5_{\pm0.2}$ | $100.0_{\pm0.0}$ | $90.2_{\pm0.2}$ | $100.0_{\pm0.0}$ |
| CMI | $37.7_{\pm3.5}$ (-53.1) | $11.0_{\pm0.4}$ (+0.6) | $34.0_{\pm1.3}$ (-57.1) | $76.1_{\pm5.7}$ (-23.9) | $50.5_{\pm1.1}$ (-40.0) | $99.9_{\pm0.1}$ (-0.1) | $46.0_{\pm2.2}$ (-44.2) | $100.0_{\pm0.0}$ (-0.0) |
| + ABD (SV) | $25.2_{\pm5.4}$ (-65.6) | $10.5_{\pm0.0}$ (+0.1) | $31.1_{\pm2.7}$ (-60.0) | $77.7_{\pm3.5}$ (-22.3) | $37.5_{\pm3.0}$ (-53.0) | $99.8_{\pm0.1}$ (-0.2) | $34.8_{\pm7.9}$ (-55.4) | $100.0_{\pm0.0}$ (-0.0) |
| + ABD (SV+SR) | $12.5_{\pm3.3}$ (-78.3) | $10.3_{\pm0.1}$ (-0.1) | $12.1_{\pm2.7}$ (-79.0) | $73.0_{\pm12.0}$ (-27.0) | $16.4_{\pm6.6}$ (-74.1) | $98.1_{\pm1.4}$ (-1.9) | $37.8_{\pm5.3}$ (-52.4) | $100.0_{\pm0.0}$ (-0.0) |
| + CKD (Ours) | $49.8_{\pm3.6}$ (-41.0) | $33.9_{\pm2.3}$ (+23.5) | $50.1_{\pm3.9}$ (-41.0) | $5.4_{\pm6.4}$ (-94.6) | $68.3_{\pm8.8}$ (-22.2) | $48.0_{\pm39.3}$ (-52.0) | $67.1_{\pm14.3}$ (-23.1) | $100.0_{\pm0.0}$ (-0.0) |
| + ATEsc (Ours) | $44.7_{\pm5.8}$ (-46.1) | $32.9_{\pm5.5}$ (+22.5) | $43.9_{\pm6.4}$ (-47.2) | $3.7_{\pm2.4}$ (-96.3) | $59.7_{\pm16.4}$ (-30.8) | $19.0_{\pm5.5}$ (-81.0) | $46.9_{\pm3.8}$ (-43.3) | $32.2_{\pm15.1}$ (-67.8) |

## E.6. Experiments on More Backbones

The results with different series of network architecture for teachers and students are shown in Table 12. From the presented results, the proposed ATEsc is consistently effective in mitigating the OOD trap effect caused by NTL teachers.

*Table 12.* Experiments on more backbones. We report the ID domain accuracy in blue and OOD domain accuracy in red. Results from the DFKD baselines are highlighted with gray rows. The accuracy drop compared to the pre-trained model is shown in brackets.

| | ID: **CIFAR10** OOD: **STL** | | | | ID: **CIFAR10** OOD: **Digits** | | | |
|---|---|---|---|---|---|---|---|---|
| | **ResNet-34→VGG-11** | | **VGG-13→ResNet-18** | | **ResNet-34→VGG-11** | | **VGG-13→ResNet-18** | |
| | IAcc (%) ↑ | OAcc (%) ↑ | IAcc (%) ↑ | OAcc (%) ↑ | IAcc (%) ↑ | OLAcc (%) ↓ | IAcc (%) ↑ | OLAcc (%) ↓ |
| NTL Teacher | $90.8_{\pm0.1}$ | $10.4_{\pm0.0}$ | $92.7_{\pm0.1}$ | $10.6_{\pm0.1}$ | $91.1_{\pm0.1}$ | $100.0_{\pm0.0}$ | $92.9_{\pm0.0}$ | $99.5_{\pm0.3}$ |
| NAYER | $10.3_{\pm0.4}$ (-80.5) | $9.7_{\pm1.3}$ (-0.7) | $18.1_{\pm0.3}$ (-74.6) | $10.6_{\pm0.2}$ (+0.0) | $10.6_{\pm0.0}$ (-80.5) | $100.0_{\pm0.0}$ (-0.0) | $23.7_{\pm0.0}$ (-69.2) | $98.0_{\pm1.1}$ (-1.5) |
| + CKD | $79.1_{\pm0.1}$ (-11.7) | $40.0_{\pm1.1}$ (+29.6) | $30.4_{\pm7.2}$ (-62.3) | $20.8_{\pm5.8}$ (+10.2) | $76.0_{\pm5.9}$ (-15.1) | $5.1_{\pm0.4}$ (-94.9) | $36.2_{\pm19.9}$ (-56.7) | $7.8_{\pm6.8}$ (-91.7) |
| + ATEsc | $68.4_{\pm2.1}$ (-22.4) | $51.7_{\pm2.4}$ (+41.3) | $28.6_{\pm4.2}$ (-64.1) | $18.7_{\pm1.9}$ (+8.1) | $74.5_{\pm3.9}$ (-16.6) | $5.5_{\pm0.1}$ (-94.5) | $48.1_{\pm1.7}$ (-44.8) | $1.3_{\pm0.9}$ (-98.2) |

## E.7. Visualization

**t-SNE visualization.** We plot the feature distribution of the identified ID-like synthetic samples and OOD-like synthetic samples by using t-SNE feature visualization (Van der Maaten & Hinton, 2008). Results on open-set, close-set, and backdoor tasks are shown in Figures 44 and 46. In each subfigure, we compare the features of synthetic samples with those of real ID and real OOD domain features. Before synthetic data grouping, we observe that synthetic samples' features partially overlap with both real ID and OOD domain features. In contrast, after synthetic data grouping, the features of identified fragile samples (regarded as ID-like samples) predominantly merge with real ID samples, as shown in Figures 44 to 46 (b). Similarly, the features of identified robust samples (regarded as OOD-like samples) align more closely with real OOD samples, as shown in Figures 44 to 46 (c). These results demonstrate the effectiveness of our synthetic data grouping strategy and further validate the adversarial robustness difference.

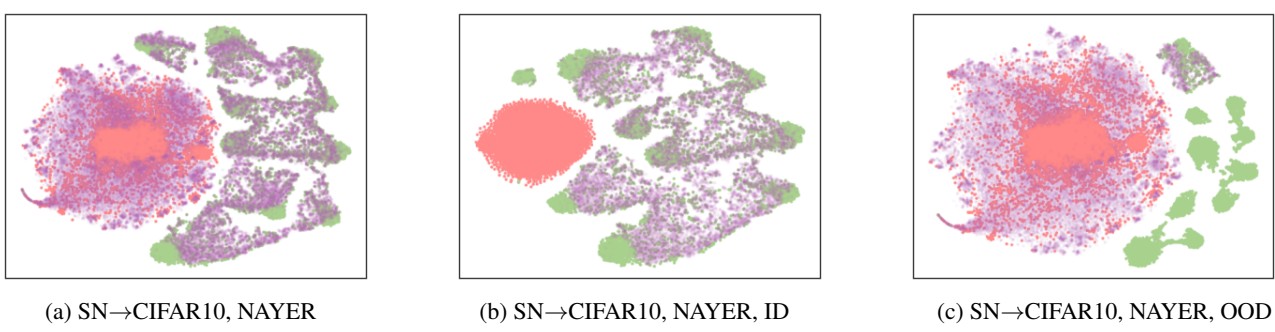

(a) SN→CIFAR10, NAYER          (b) SN→CIFAR10, NAYER, ID          (c) SN→CIFAR10, NAYER, OOD

*Figure 44.* Comparison of synthetic samples before and after ATEsc (DFKD by NAYER. $T$: ResNet34 and $S$: ResNet18). Green dots and red dots represent real ID and OOD domain samples, respectively. Purple transparent dots represent the synthetic samples.

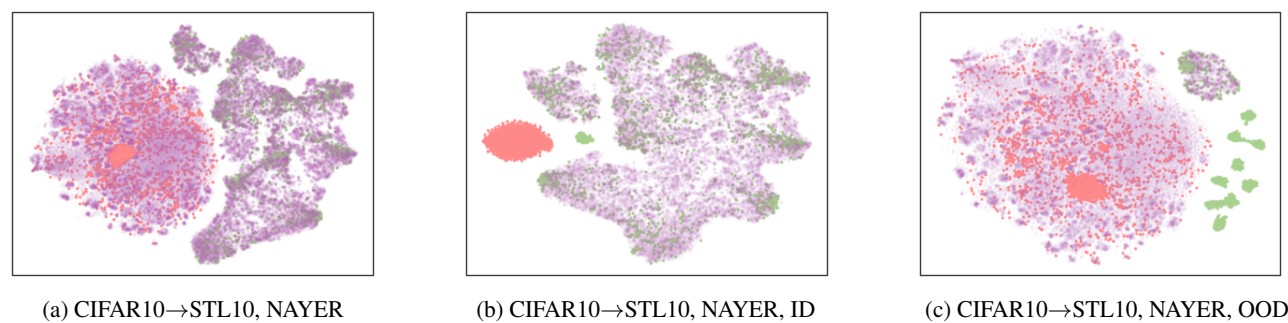

(a) CIFAR10→STL10, NAYER          (b) CIFAR10→STL10, NAYER, ID          (c) CIFAR10→STL10, NAYER, OOD

*Figure 45.* Comparison of synthetic samples before and after ATEsc (DFKD by NAYER. $T$: ResNet34 and $S$: ResNet18). Green dots and red dots represent real ID and OOD domain samples, respectively. Purple transparent dots represent the synthetic samples.

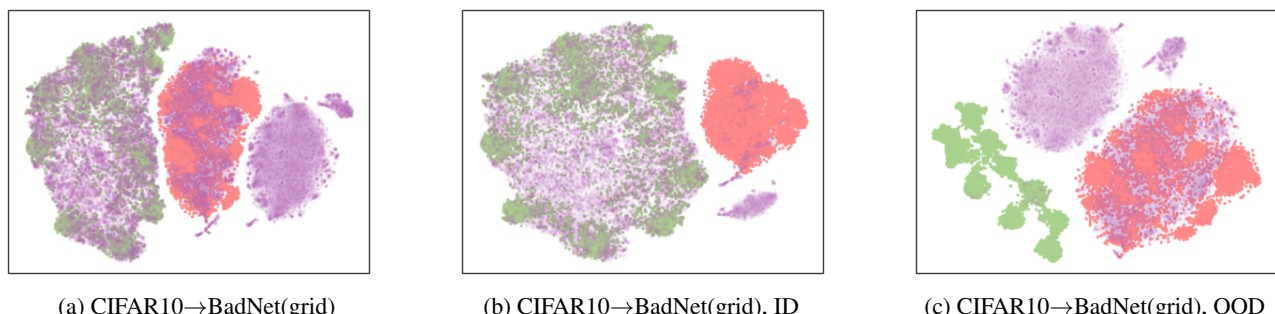

(a) CIFAR10→BadNet(grid)          (b) CIFAR10→BadNet(grid), ID          (c) CIFAR10→BadNet(grid), OOD

*Figure 46.* Comparison of synthetic samples before and after ATEsc (DFKD by NAYER. $T$: ResNet34 and $S$: ResNet18). Green dots and red dots represent real ID and OOD domain samples, respectively. Purple transparent dots represent the synthetic samples.

**Synthetic samples by using ATEsc.** In addition, we visualize the synthetic samples during distilling NTL teachers before and after using synthetic data grouping. The visualization results are shown in Figures 47 and 49. We can observe that original synthetic samples derived by DFKD on NTL teachers are likely to integrate both the ID domain and OOD domain

information (see Figures 47 to 49 (a)). After synthetic data grouping, the identified fragile samples, which are regarded as ID-like samples, resemble real ID samples more closely (see Figures 47 to 49 (b)), while those robust samples appear more similar to real OOD samples, as shown in Figures 47 to 49 (c). These observations are consistent with expectations, demonstrating the effectiveness of our method.

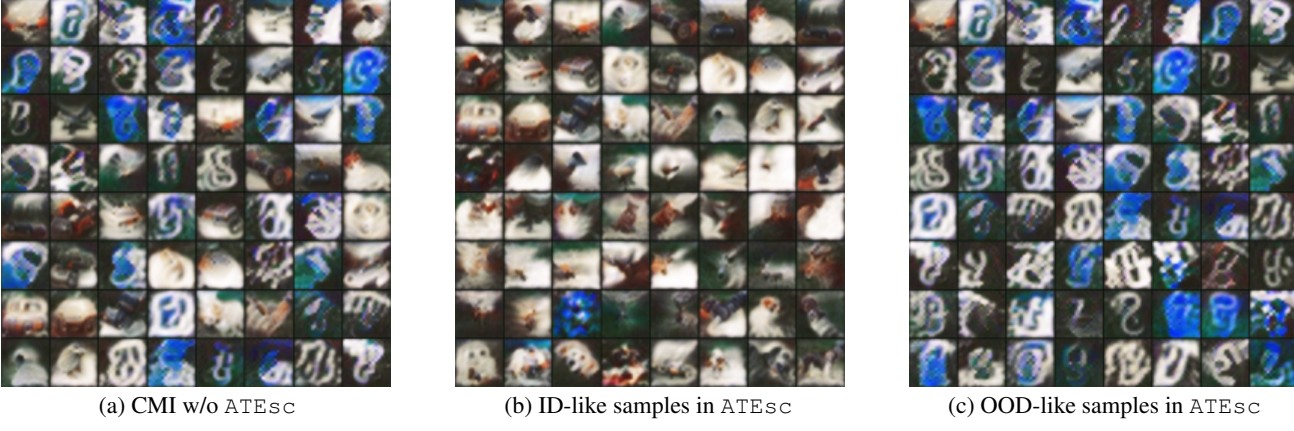

| (a) CMI w/o `ATEsc` | (b) ID-like samples in `ATEsc` | (c) OOD-like samples in `ATEsc` |

*Figure 47.* Comparison of synthetic samples (CMI on NTL CIFAR10→Digits. $T$: ResNet34 and $S$: ResNet18).

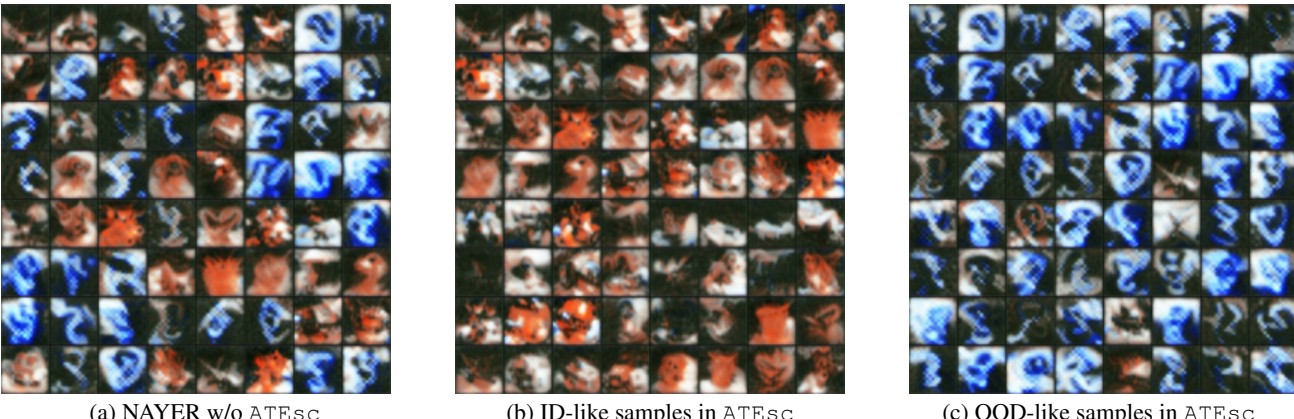

| (a) NAYER w/o `ATEsc` | (b) ID-like samples in `ATEsc` | (c) OOD-like samples in `ATEsc` |

*Figure 48.* Comparison of synthetic samples (NAYER on NTL CIFAR10→Digits. $T$: ResNet34 and $S$: ResNet18).

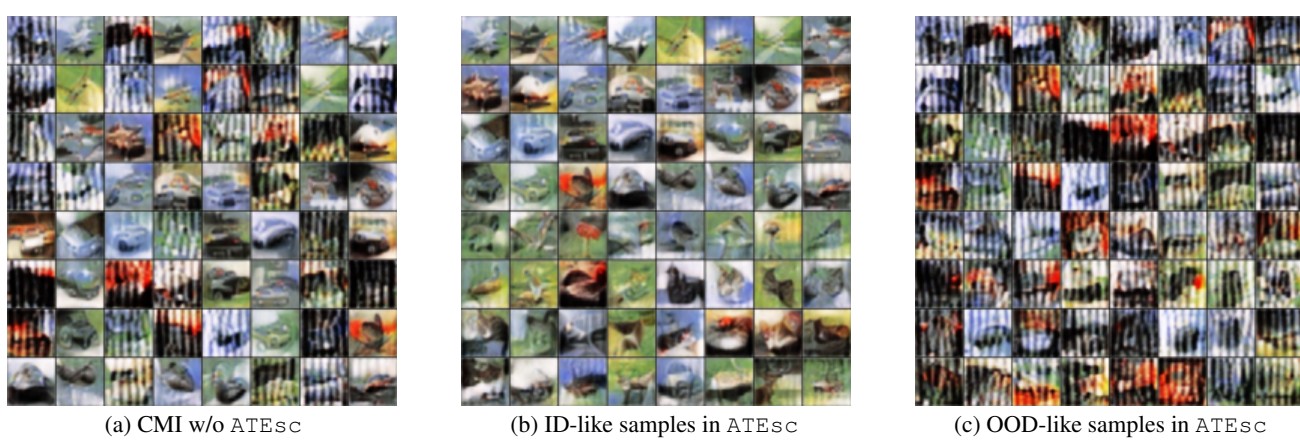

| (a) CMI w/o `ATEsc` | (b) ID-like samples in `ATEsc` | (c) OOD-like samples in `ATEsc` |

*Figure 49.* Comparison of synthetic samples (CMI on NTL CIFAR10→Sig. $T$: ResNet34 and $S$: ResNet18).

# F. Further Discussion

## F.1. Distribution Shift in DFKD

There are some other works focus on the issue of distribution shift in DFKD, such as Do et al. (2022); Binici et al. (2022b;a); Wang et al. (2024b). In this section, we discuss the similarities and differences between our work and these previous works.

Do et al. (2022); Binici et al. (2022b;a); Wang et al. (2024b) investigate various types of distribution shifts that arise when performing DFKD on a standard supervised learning (SL) teacher trained solely on the ID domain:

- Do et al. (2022); Binici et al. (2022b;a) focus on the non-stationary distribution problem, where shifts between synthetic distribution in different training stages (i.e., epochs) cause the catastrophic forgetting in the student model. This issue can be mitigated by memory bank-based strategies, where old synthetic samples are stored and will be replayed in future epochs (Do et al., 2022; Binici et al., 2022b) or by using an additional VAE (Kingma & Welling, 2013) to model the previously observed synthetic samples (Binici et al., 2022a).

- Wang et al. (2024b) addresses the distribution shift between synthetic data and real training data. They propose a novel perspective with causal inference (Von Kügelgen et al., 2021; Schölkopf et al., 2021; Lin et al., 2023; Zheng et al., 2024; Yao et al., 2025; Chen et al., 2024) to disentangle the student model from the impact of such shifts.

Our work explores the DFKD under NTL teachers. We find that NTL teachers (pretrained on an ID domain and an OOD domain) result in OOD trap effect for DFKD. One key reason is the ID-to-OOD synthetic distribution shift, which is caused by the joint effect of the BN loss and adversarial exploration loss when training the generator in DFKD. Such a shift will not occur when distilling an SL teacher trained on the ID domain by using DFKD.

## F.2. The Threat of Latent Inseparability

The concept of the latent inseparability was originally introduced in the adaptive backdoor attack (Qi et al., 2023), which aims to reduce the distinguishability between backdoor and clean features, thereby evading defenses that rely on the latent separability assumption (Tran et al., 2018; Chen et al., 2018; Tang et al., 2021; Hayase et al., 2021). In this section, we discuss how latent inseparability may emerge when performing DFKD on NTL-based backdoor teachers.

**Synthesized ID-like data in DFKD may lead to adaptive backdoor attacks.** Although the proposed calibrated knowledge distillation (CKD, Section 4.2) can filter out OOD-like samples (i.e., backdoor samples for NTL-based backdoor teachers), some intermediate samples inevitably exist. These samples are not robust but can still activate the teacher's backdoor knowledge, enabling a certain degree of backdoor knowledge transfer from teacher to student. More seriously, the backdoor features in the student may exhibit latent inseparability, potentially leading to adaptive backdoor attacks (like (Qi et al., 2023)) on the student. Specifically,

- Some ID-like samples may still contain minor backdoor triggers.
- During student training, such samples can be inadvertently perform adaptive backdoor attacks for the student. The attack manner is very similar to Qi et al. (2023) in two ways:
  - **Weakened triggers**: In Qi et al. (2023), weakened triggers are used for training-time poisoning, while the original standard trigger would only be used during test time to activate the backdoor. Similarly, in our scenario, minor backdoor triggers are similar to weakened triggers.
  - **Regularized poisoning**: In Qi et al. (2023), not all samples with backdoor triggers are predicted to the target class. Similarly, in our scenario, since only minor trigger patterns are present, the NTL teacher may not always predict these samples to the target class.
- Therefore, during DFKD, if we using the selected ID-like samples to training the student (i.e., the proposed CKD method), the NTL-based backdoor teacher potentially causes latent inseparability of clean and backdoor features for the student.

**While our method does not explicitly identify adaptive backdoor features within the synthetic data, it does include a defense mechanism that mitigates their influence, i.e., the forgetting term $\mathcal{L}_{\text{forget}}$.** Actually, we have considered that some ID-like samples can still activate the teacher's misleading backdoor knowledge, enabling a certain degree of backdoor knowledge transfer from teacher to student (see Section 4.4). To further suppress misleading backdoor knowledge transfer, we introduce a forgetting term $\mathcal{L}_{\text{forget}}$ to *maximize the divergence* between the student's and the teacher's predictions on backdoor-like samples. Intuitively,

- Backdoor-like samples contain strong backdoor triggers. NTL teacher will predict them to the target class.

- Therefore, maximizing the divergence between the student's and the teacher's predictions on backdoor-like samples can directly prevent the student from learning backdoor knowledge. This has the opposite effect to the adaptive-backdoor-attack formed by NTL teacher (which could potentially plant the backdoor to the student via weakened triggers).

As a result, even if some synthetic ID-like samples contain adaptive backdoor features, the student is hard to still encode inseparable latent representations. The forgetting term $\mathcal{L}_{\text{forget}}$ effectively mitigates the risk of adaptive backdoor formation during DFKD.

**Empirically, the effectiveness of the forgetting term is demonstrated in Tables 1 to 3.** Compared to using CKD alone, our full ATEsc (i.e., CKD+$\mathcal{L}_{\text{forget}}$) achieves a significantly lower ASR. Considering that the adaptive backdoor attack in Qi et al. (2023) still uses the original standard trigger during test time to activate the backdoor, our results already show its effectiveness against the adaptive backdoor attack (Qi et al., 2023) which is inadvertently formed during DFKD. More importantly, this forgetting term shows its effectiveness across all three NTL settings, highlighting its generality.

