# OpenReview forum: "When Data-Free Knowledge Distillation Meets Non-Transferable Teacher: Escaping Out-of-Distribution Trap is All You Need"
_ICML.cc/2025/Conference — ICML 2025 poster_

### Official Review · Reviewer_gtda · 2025-03-13

**Overall Recommendation:** 2

**Summary:**

This work investigates the vulnerability of data-free knowledge distillation (DFKD) when the teacher model is untrusted, particularly in non-transferable learning (NTL) scenarios where knowledge transfer is blocked. It finds that NTL teachers mislead DFKD by shifting the generator’s focus from useful in-distribution (ID) knowledge to misleading OOD knowledge. To address this, the authors propose Adversarial Trap Escaping (ATEsc), which distinguishes between ID-like (fragile) and OOD-like (robust) synthetic samples based on adversarial robustness. Fragile samples aid normal distillation, while robust ones help forget misleading OOD knowledge, improving DFKD's effectiveness against NTL teachers.

**Claims And Evidence:**

The claims made in the submission are partially supported by evidence, but certain aspects require further validation. Specifically, the use of fragile samples for ID knowledge transfer is not entirely convincing. The paper assumes that fragile samples (i.e., those with low adversarial robustness) are ID-like and thus beneficial for knowledge distillation. However, a model’s confidence on a sample does not necessarily correlate with its true distribution alignment—low-confidence predictions can be arbitrarily adjusted to fit almost any data. This raises concerns about whether the fragile samples truly retain ID knowledge or if they are simply artifact samples that do not generalize well to real ID data.

**Essential References Not Discussed:**

The authors have discussed most of the relevant prior work, covering key research on data-free knowledge distillation (DFKD), non-transferable learning (NTL), adversarial robustness, and model security. There do not appear to be major omissions in citing essential references.

**Ethics Expertise Needed:**

["Privacy and Security"]

**Experimental Designs Or Analyses:**

I have checked all the experiments and designs, but there are significant issues. The authors lack an in-depth analysis of fragile samples, particularly why they are considered high-confidence ID data. In my view, the fragile sample space is infinitely large, making their selection potentially unreliable.

**Methods And Evaluation Criteria:**

The datasets, problem, and application in this paper make sense and are relevant.

**Other Comments Or Suggestions:**

I consider the manuscript quality needs improvement, and the experimental analysis should be more comprehensive.

**Other Strengths And Weaknesses:**

A key strength of the paper is its originality, as it is the first to explore attacking non-transferable learning (NTL) teachers in the context of data-free knowledge distillation (DFKD). This makes it a novel and significant contribution to the field of model security and adversarial robustness.

However, the paper lacks a thorough analysis of the effectiveness of the proposed method. Specifically, the assumption that low-confidence (fragile) samples represent ID knowledge is not well-supported. A more detailed theoretical or empirical analysis of why fragile samples align with ID data would strengthen the claims. Additionally, the paper does not provide sufficient justification for why ATEsc can effectively prevent backdoor attacks.

**Questions For Authors:**

Q1. This paper is not clearly written. The authors should specify what knowledge is being extracted, which datasets can be used for evaluation, and whether the attacked model is in a white-box or black-box setting. Additionally, can the training data distribution of the model be inferred?
Q2. The second question is about ID-OOD learning task conflicts. If the goal is to create conflicts between ID and OOD learning, would this lead to a significant accuracy loss for the NTL model, especially under the same label space? Has the paper analyzed datasets with different data distributions?
Q3. Would adding noise affect the model's performance? Additionally, why is this method effective in defending against backdoor attacks?
Q4. High-confidence data could be either OOD data or ID data. How do you distinguish between the two?

**Relation To Broader Scientific Literature:**

The authors are the first to propose an attack on non-transferable learning (NTL) teachers, making this work a novel contribution to the field of data-free knowledge distillation (DFKD) and model security.

**Theoretical Claims:**

Yes, the mathematical formulations in the paper are generally correct and align with the theoretical claims.

---

> ### Author Rebuttal · Authors · 2025-04-01
>
> Dear Reviewer gtda,
>
> Thanks for your valuable comments! We address the weaknesses below. Please kindly let us know if you have further concerns.
>
> >**Q1: Why do fragile samples align with ID data, and how do we distinguish ID and OOD data.**
>
> We are sorry for the confusion. This assumption of "fragile synthetic samples align with ID data" is grounded in the widely accepted consensus within DFKD that BN loss is a necessary component. Due to the BN loss for training the generator in SOTA DFKD methods (like DFQ, CMI, and NAYER), the synthetic distribution will be constrained to have similar BN statistics with the teacher's training distribution. For more details on the BN loss, please find the answer to **Reviewer-saQK@Q1** (sorry for the limited space).
>
> As such, when distilling NTL teachers, joint effects of BN loss and adversarial exploration loss let the synthetic distribution become similar to the mixture distribution of real ID and OOD domains. Therefore, according to our findings of the adversarial robustness difference of real ID and OOD domain samples against NTL teachers (refer to Section 4.1 in the main paper), we make the assumption that fragile synthetic samples are ID-like samples, and robust synthetic samples are OOD-like.
>
> Empirically, the DFKD performance improvement by the introduced CKD (Tab. 1-3 in the main paper), visualization of original synthetic samples and ID-/OOD-like synthetic samples (Figs. 30-31 in main paper and Figs. R6-R7 in https://anonymous.4open.science/r/icml408/figs.pdf), t-SNE feature visualizations (Figs. 28-29 in main paper) can verify the effectiveness of our assumption.
>
> >**Q2: Our settings**
>
> We sincerely apologize for the confusion. According to the tasks NTL can handle, we consider DFKD for three types of NTL teachers: **closed-set NTL**, **open-set NTL**, and **NTL-based backdoor**. The difference between these tasks lies in the data used for ID and OOD domains in training NTL teachers. Specifically,
> - **closed-set NTL**: In this setting, the ID and OOD domains have the same label space, but the two domains have a distribution shift. We choose the dataset-pairs of: SVHN→MNIST-M and CIFAR10→STL10;
> - **open-set NTL**: In this setting, the label spaces for the ID domain and OOD domain are disjointed. We perform experiments on SVHN→CIFAR10 and CIFAR10→Digits, where Digits is the combination of 5 digits datasets: MNIST, USPS, SVHN, MNIST-M, and SYN-D;
> - **NTL-based backdoor**: We additionally consider using NTL to conduct training controllable backdoor attacks. We use four triggers: BadNets(sq), BadNets(grid), Blended, Sig on CIFAR10 and see them as the OOD domain. The clean CIFAR10 is regarded as the ID domain.
>
>
> Our aim is to address the malign *OOD-Trap effect* introduced by NTL teachers in DFKD, and thus, regardless of NTL teacher's tasks, our goal is to transfer only the ID domain knowledge from NTL teacher to student. In addition, all NTL teachers are considered as **white-box** models during DFKD.
>
> >**Q3: Would creating conflicts between ID and OOD learning lead to a significant accuracy loss for the NTL model?**
>
> It's true that NTL teachers have a certain degree of accuracy loss on either ID domain and any unseen domains. Empirically, we compare the performance of SL and Close-/Open-set NTLs on seen and unseen domains, as well as corrupt seen domain (with additive gaussian noise):
>
> **Table C1:** Accuracy loss of NTL. NetArchs: ResNet-34→ResNet-18
> ||SL(CIFAR10)|Close-set NTL(ID: CIFAR10 OOD: STL)|Open-set NTL(ID: CIFAR10 OOD: Digits)|
> |-|-|-|-|
> |CIFAR10(test)|92.9|90.8|91.2|
> |CIFAR10(test)+gaussian(std=0.1)|90.6|45.7|76.7|
> |CIFAR10(test)+gaussian(std=0.2)|80.7|10.6|28.5|
> |STL(test)|68.8|10.4|17.3|
>
> >**Q4: Why is this method effective in defending against backdoor attacks?**
> - **NTL-based backdoor.** In our experiments, we consider using NTL in conducting training controllable backdoor attacks, where clean data (e.g., CIFAR10) is regarded as ID domain and clean data with backdoor triggers (e.g., CIFAR10 with BadNets) is seen as the OOD domain. Through our empirical findings (Fig. 4 (b-c) and Tab. 3 in our main paper), NTL-based backdoored teachers can transfer backdoors to students. This is also because of the *OOD-trap effects* as we mainly analyzed in Section 3 of our main paper.
> - **Defending against NTL-based backdoor.** Our ATEsc can effectively defend the backdoor transfer from teacher to student through the DFKD process. This is because in the NTL-based backdoor task, the OOD domain (e.g., CIFAR10+BadNets) and the ID domain (original CIFAR10) exhibit significant differences in their adversarial robustness against NTL teachers. As such, the proposed ATEsc can distinguish ID-like and OOD-like synthetic samples and let the student learn ID domain knowledge. Visualization results in Fig. 29 and Fig. 31 provide evidence for the effectiveness of distinguishing between ID and OOD synthetic data for NTL-based backdoor teachers.

---

> > ### Comment · Reviewer_gtda · 2025-04-06
> >
> > Thank you for the detailed response. However, I would like to further clarify the assumption made in both Q1 and Q4 regarding the alignment between adversarial robustness and the ID/OOD (or backdoor) categorization. I appreciate the discussion, but I will maintain my current evaluation scores.
> >
> > In your response, fragile synthetic samples are assumed to be ID-like, while robust ones are assumed to be OOD-like. This assumption appears to be rooted in the observed adversarial robustness differences between ID and OOD samples under NTL teachers. However, this premise becomes more uncertain in the backdoor setting, where backdoored samples may be confidently and robustly classified by the teacher model.
> >
> > Specifically:
> >
> > - Backdoored samples, although manipulated, can be confidently and robustly classified into the target class by the backdoored NTL teacher.
> > - Therefore, they may behave similarly to ID samples in terms of confidence and possibly even adversarial fragility.
> > - Consequently, it seems possible that some fragile samples could in fact be backdoored OOD samples, especially if they lie close to the decision boundary but are still confidently classified.
> >
> > Could you clarify why these fragile samples are unlikely to be backdoor samples, and why the ATEsc framework can reliably distinguish between ID and backdoor-induced OOD samples under this ambiguity?

---

> > > ### Author Response · Authors · 2025-04-08
> > >
> > > Thanks for your reply. We would like to clarify more about *distinguishing clean/backdoor samples according to their adversarial robustness against NTL-based backdoor teachers*.
> > >
> > > >**1. Directly evidence for the *dissimilarly adversarial fragility and confidence* of backdoor/clean samples**.
> > >
> > > We first provide direct evidence regarding the significant difference of **adversarial robustness** and **confidence** between clean and backdoor data in https://anonymous.4open.science/r/icml408/figs.pdf (**Fig. R12-R15**). We argue that such a difference is caused by the training manner of NTL, which we will discuss in the following parts of this response.
> > >
> > > >**2. The NTL training loss leads to significantly different behavior for clean and backdoor samples.**
> > >
> > > **2.1 How we conduct NTL-based backdoored teacher training**
> > >
> > > We use NTL to conduct *training controllable backdoor attacks* [C1] for a teacher model. Formally, we have a clean ID domain $\mathcal{D_{\text{id}}}=\\{(x_i,y_i)\\}_{i=1}^{N _\text{id}}$, and we can get an OOD domain by adding trigger $\delta$ on $\mathcal{D _{\text{id}}}$, i.e., $\mathcal{D} _{\text{ood}}=\\{(x _i+\delta,y _\text{bd})\\} _{i=1}^{N _\text{id}}$, where $y _\text{bd}$ is the target class. By minimizing $\mathcal{L} _{\text{NTL-cls}}$, we train a NTL-based backdoor teacher $f$:
> > >
> > > $\mathcal{L} _{\text{NTL-cls}}=\mathcal{L} _{\text{id}}-\underbrace{\min(1,\alpha\cdot\mathcal{L} _{\text{out}}\cdot\mathcal{L} _{\text{feat}})} _{\mathcal{L} _{\text{NT}}}+\lambda _{\text{cls}}\cdot\mathcal{L} _{\text{cls}},$
> > >
> > > $=\mathbb{E} _{(x,y)\sim \mathcal{D} _{\text{id}}}\Big\\{D _{\text{KL}}(f(x), y)-\underbrace{\min(1,\alpha\cdot D _{\text{KL}}(f(x+\delta),f(x)) \cdot \text{MMD}(f_e(x),f_e(x+\delta)))} _{\mathcal{L} _{\text{NT}}}+\lambda _{\text{cls}}\cdot\mathcal{L} _{\text{CE}}(f(x+\delta),y _{\text{bd}})\Big\\}$
> > >
> > > where $\alpha$ and $\lambda _\text{cls}$ are weights, $f _e$ is the feature extractor in $f$.
> > >
> > > The **major difference** between NTL-based backdoor learning and conventional backdoor learning [C1,C2] is the additional term of $\mathcal{L} _{\text{NT}}$. Under the combined effect of $\mathcal{L} _{\text{cls}}$ and $\mathcal{L} _{\text{NT}}$, *backdoor samples* will not only be classified to the target class $y _\text{bd}$ (like conventional backdoor learning), but also have a far distance to its clean version in feature and output space. Existing defense ABD [C2], although successful in defense of conventional backdoor teacher, fails to deal with the transfer of backdoor from NTL-based backdoored teacher (Sec. C.5 in our main paper).
> > >
> > > **2.2 A margin perspective for why the difference of adversarial robustness is hold for NTL-based backdoor task**
> > >
> > > NTL training will cause significantly different **margins** (defined as the minimal distance from a sample point to decision boundaries [C3]) for *clean samples* and *backdoored samples*.
> > >
> > > - *For clean samples*, learning correct classification results in relatively **complex decision boundaries** between clean data classes and **small margins** for clean data points.
> > >
> > > - *For backdoor samples*:
> > >     - minimizing $\mathcal{L} _{\text{cls}}$ forces all backdoor data to be predicted as a single class, which is simple, and **no boundaries** will go through the backdoor samples.
> > >     - maximizing $\mathcal{L} _{\text{NT}}$ further pushes the backdoor cluster far away from the clean sample clusters, resulting in **very large margins** for backdoor samples (i.e., **no backdoor samples will close to decision boundary**).
> > >
> > > As a larger margin corresponds to stronger robustness [C3], we can assume NTL teachers exhibit strong/fragile robustness against adversarial attacks on backdoored/clean data.
> > >
> > > Besides, **if w/o $\mathcal{L} _{\text{NT}}$**, the teacher training is degraded to *conventional backdoor training*, and the margins of backdoor samples are not constrained to become large enough. In such a situation, we agree that some fragile samples could be backdoored samples.
> > > *However, we emphasize that we are not aimed at conventional backdoor situations. The proposed ATEsc are designed to defend against backdoor teachers trained by NTL*.
> > >
> > > **2.3 Ablation studies for the influence of $\mathcal{L} _{\text{NT}}$**.
> > >
> > > The ablation studies for the influence of $\mathcal{L} _{\text{NT}}$ for the **adversarial robustness** and **confidence** difference are shown in https://anonymous.4open.science/r/icml408/figs.pdf **Fig. R16-R19** and **Fig. R20-R23**, respectively. These results provide empirical support for our analysis.
> > >
> > > ---
> > >
> > > [C1] Backdoorbench: A comprehensive benchmark of backdoor learning, NeurIPS'22\
> > > [C2] Revisiting data-free knowledge distillation with poisoned teachers. ICML'23\
> > > [C3] Exploring and Exploiting Decision Boundary Dynamics for Adversarial Robustness. ICLR'23
> > >
> > > ---
> > > Thank you for recognizing our original and significant contribution. We’d greatly appreciate a score reconsideration and the chance to share our work with the community.

---

### Official Review · Reviewer_saQK · 2025-03-13

**Overall Recommendation:** 2

**Summary:**

This paper investigates what would happen if one tried to apply data-free knowledge distillation (DFKD) on non-transferable learning (NTL) teachers. The paper proposes the OOD trap effect that the generator shifts to generate OOD samples, causing the student model only learns misleading OOD knowledge. To solve this problem, this paper proposes to first identify and filter out those OOD samples as they are more robust to untargeted adversarial attack. This simple strategy is called CKD and is fairly effective, as shown by the experiments. The authors further propose ATEsc which includes a regularization term aiming to further suppress OOD knowledge transfer. Experiments demonstrate the effectiveness of the proposed method.

**Claims And Evidence:**

The proposed method heavily builds on the proposed OOD trap effect. Although empirical evidence demonstrates the OOD trap effect, the proposed reason behind such effect is not adequately supported.
* In order for the OOD trap effect to happen, the generator has to generator samples closer and closer to the OOD samples. This paper explains this by "the [generator] will be optimized toward synthesizing OOD-like samples to satisfy the maximization of the distribution discrepancy between [student] and [teacher]." However, following this logic, the generator could just generate imperceptible samples to maximize the discrepancy (like adversarial examples), and it is not clear why the generator has to output OOD-like samples to maximize the discrepancy.
* In line 195 right column: "We assume a student S in DKFD currently only has ID domain knowledge." It seems to be the goal is to let the student learn ID domain knowledge. So why can we assume this in the first place?
* The second explanation provide by this paper is the "task conflicts." I.e., the teacher's outputs on ID and OOD data are very different, and thus the student faces "task conflicts" when trying to distill the teacher's knowledge. However, if there is indeed a "task conflict," i.e., the ID and OOD data are close enough to be considered as conflicting task, why they have vastly different adversarial robustness?
* The introduction of L_forget (12) seems ad-hoc, and that why maximizing L_forget can push the student model to forget about misleading knowledge of the teacher model? The with this term (ATEsc), it doesn't seem to be better than the vanilla CKD method.

**Essential References Not Discussed:**

N/A

**Experimental Designs Or Analyses:**

See previous discussions

**Methods And Evaluation Criteria:**

Minor question: Although this paper is about data-free knowledge distillation, where a generator must be used. I'm curious to see how regular knowledge distillation would perform with NTL teachers.

**Other Comments Or Suggestions:**

N/A

**Other Strengths And Weaknesses:**

Strengths:
this paper investigates an interesting problem, and the proposed OOD trap effect is very interesting. Especially there is potential impact if the phenomenon is well understood and well utilized.

Other Weaknesses:
This paper is not well-written. Many grammar issues.

For example,
* (line 14 right) "due to the unavailable of...";
* (line 186 left) "We analysis on...";
* (line 189 right) "distribution D_s will occur distribution shift..."

**Questions For Authors:**

See previous discussions

**Relation To Broader Scientific Literature:**

N/A

**Theoretical Claims:**

N/A

---

> ### Author Rebuttal · Authors · 2025-04-01
>
> Dear Reviewer saQK,
>
> Thanks for your valuable reviews! We address your concerns as follows. Please let us know if anything remains unclear.
> >**Q1: Why generator has to output OOD-like samples**
>
> This is because DFKDs commonly use BN loss [B1] as regularization for training generator $G$. Specifically,
> - **When training NTL**, each batch contains a mixture of ID and OOD samples. This results in that BN layers in pre-trained NTL teachers record statistical information for the mixture distribution of ID and OOD domains (mean $\mu_l$ and var $\sigma_l^2$ for layer $l$).
> - **When training $G$**, BN loss is represented as the divergence between synthetic statistics $\mathcal{N}(\mu_l(x_{syn}), \sigma_l^2(x_{syn}))$ and teacher's BN statistics $\mathcal{N}(\mu_l, \sigma_l^2)$. Minimizing BN loss lets the synthetic samples follow similar statistical information of teacher's training samples.
>
> The joint effects of BN loss and adversarial exploration loss (Eq. 7 in the main paper) constraint the $G$ to:
> - (i) follow the statistics of NTL teacher's training data (i.e., a mixture of ID and OOD domains), and
> - (ii) maximize the discrepancy between student $S$ and NTL teacher.
>
> Thus, if the $S$ only has ID domain knowledge, the $G$ will be optimized to synthesize OOD-like samples.
>
> If w/o BN loss, the $G$ is unnecessary to generate OOD-like samples, and DFKD will not be influenced by OOD trap from NTL teachers. Unfortunately, their ID performance will be influenced by low quality of synthetic samples. Corresponding empirical evidence is shown in Tab. B1. More results and synthetic samples w/ and w/o BN are in https://anonymous.4open.science/r/icml408/figs.pdf (Tab. R1 and Fig. R1-R5).
>
> **Table B1:** ID: CIFAR10, OOD: STL; ResNet-34→ResNet-18
> ||IAcc↑|OAcc↑|
> |-|-|-|
> |NTL Teacher|90.8±0.1|10.4±0.0|
> |CMI w/ bn|37.7±3.5(-53.1)|11.0±0.4(+0.6)|
> |CMI w/o bn|31.1±3.2(-59.7)|27.4±3.1(+17.0)|
> |NAYER w/ bn|48.7±1.1(-42.1)|10.4±0.0(+0.0)|
> |NAYER w/o bn|40.8±2.9(-50.0)|29.3±2.2(+18.9)|
>
> [B1] Dreaming to distill: Data-free knowledge transfer via deepinversion, CVPR'20
> >**Q2: Why assume a student only has ID knowledge first**
>
> Sorry for the confusion. We aim to demonstrate that in adversarial exploration-based DFKD, even if a student $S$ learns only ID knowledge in *initial or some intermediate stage*, the $S$ will inevitably let the $G$ to synthesize OOD-like samples and then be taught to learn misleading OOD knowledge.
> >**Q3: Why there are vastly different adversarial robustness for close ID and OOD data**
>
> Because the training manner of NTL causes significantly different **margins** (i.e., the minimal distance from a sample point to decision boundaries [B2]) for samples from even two similar domains:
> - **For ID domain**, the objective is to learn correct classification (e.g., 10-class classification for CIFAR10). This results in relatively complex decision boundaries between ID domain classes and small margins for ID data points.
> - **For OOD domain**, the objective is to force all data to be predicted as a single class (Eq. 5), and to maximize the domain discrepancy on both logits and features (Eq. 4). Predicting all samples to one class is simple, and no boundaries will go through the OOD clusters. Besides, maximizing the discrepancy on logits/features between domains further pushes the OOD cluster away from the ID domain clusters, resulting in very large margins for OOD data.
>
> As a larger margin corresponds to stronger robustness [B2], we assume NTL teachers exhibit strong/fragile robustness against adversarial attacks on OOD/ID data. Empirical studies verify our assumption (Fig. 19-24 in main paper). We provide more evidence in https://anonymous.4open.science/r/icml408/figs.pdf (Fig. R8-R11) to analyze the effects of NTL's objectives for robustness.
>
> [B2] Exploring and Exploiting Decision Boundary Dynamics for Adversarial Robustness, ICLR'23
> >**Q4: The forget term**
>
> If the student $S$ and NTL teacher have similar predictions on OOD-like samples, it suggests that the $S$ has learned misleading OOD knowledge. By maximizing $\mathcal{L}_{\text{forget}}$, the $S$'s outputs on OOD-like samples are encouraged to diverge from the teacher's outputs, thus mitigating the misleading OOD knowledge transfer.
>
> This term is effective in suppressing OOD knowledge transfer, as evidenced by the decreased OLAcc (Tab. 2) and the reduced ASR (Tab. 3). But it also degrades ID knowledge transfer. Such trade-off depends on whether the priority is on preserving ID transfer or preventing OOD transfer.
> >**Q5: Regular KD results**
>
> Thanks! We present results of regular KD by ID data (KD-ID), OOD data  (KD-OOD) and ID+OOD data (KD-All) in Tab. B2. More results in https://anonymous.4open.science/r/icml408/figs.pdf (Tab. R2)
>
> **Table B2:** ID: CIFAR10, OOD: STL; ResNet-34→ResNet-18
> ||IAcc↑|OAcc↑|
> |-|-|-|
> |NTL Teacher|90.8±0.1|10.4±0.0|
> |KD-ID|80.4±0.2(-10.4)|58.1±1.0(+47.7)|
> |KD-OOD|10.6±0.0(-80.2)|10.4±0.0(+0.0)|
> |KD-All|76.6±0.4(-14.2)|10.5±0.0(+0.1)|

---

### Official Review · Reviewer_iM7M · 2025-03-16

**Overall Recommendation:** 3

**Summary:**

This paper identifies the OOD trap effect from NTL teachers to DFKD, i.e., misleading knowledge from OOD data may mislead students' learning process. The authors propose a plug-and-play ATEsc method to ensure that students can benefit from the NTL teacher model. This article can be considered as filling a gap in the field of DFKD, and the logic is clear, vivid, and easy to understand, accompanied by certain theoretical basis.

**Claims And Evidence:**

Most of the claims in this article have been validated.

**Essential References Not Discussed:**

I have learned about some other work on distribution offset issues in DFKD tasks. It would be even better to discuss the similarities and differences between this article and these methods (only discuss without experiments).

[1] Momentum adversarial distillation: Handling large distribution shifts in data-free knowledge distillation. NeurIPS 2022.

[2] Preventing catastrophic forgetting and distribution mismatch in knowledge distillation via synthetic data. WACV 2022.

[3] Robust and resource-efficient data-free knowledge distillation by generative pseudo replay. AAAI 2022.

[4] De-confounded Data-free Knowledge Distillation for Handling Distribution Shifts. CVPR 2024.

**Experimental Designs Or Analyses:**

The experimental results are relatively complete and the research is sufficient within the dataset scope mentioned by the author. However, to my knowledge, many DFKD methods can be applied to larger datasets, such as CIFAR-100, Tiny ImageNet, or ImageNet In addition, there are relatively few backbone combinations between teachers and students, and they are all homologous models, such as VGG or ResNet. This article introduces adversarial perturbations in training, which may introduce significant additional overhead, and the discussion in this section is missing. In addition, there are other methods to address the issue of data domain distribution offset. It would be even better if we could discuss the similarities and differences with these methods.

**Methods And Evaluation Criteria:**

Toy experiments and validation experiments can support textual conclusions.

**Other Comments Or Suggestions:**

* I suggest the author compare and discuss several works on data distribution shift in DFKD tasks, which will help readers further clarify the task setting and value of this article.
* What would happen if teachers and students choose different model series, such as ViTs and CNNs, or different CNN series models.

**Other Strengths And Weaknesses:**

Strengths:
* This article focuses on the impact of NTL teachers and data distribution differences on student learning in DFKD tasks.
* The writing is good, with clear logic and sufficient small-scale experiments to clarify the motivation and effectiveness of the method.

Weaknesses:
Weaknesses can refer to Essential References, Suggestions, and Questions.

**Questions For Authors:**

* The performance of the original versions of CMI and NAYER seems to be very unstable, such as IAcc. What do you think is the main reason. As far as I know, the generation process of CMI relies on the statistical information of the BN layer, so it seems not suitable for experiments across datasets. After combining with ATEsc, good results were achieved. Does distinguishing between ID and OOD data during training help solve the problem of mismatched statistical information？
* Is the default attack steps 5? How much training cost will this introduce？

**Relation To Broader Scientific Literature:**

The concept of teachers in this article was inspired by NTL (Wang et al., 2022).

**Theoretical Claims:**

Yes

---

> ### Author Rebuttal · Authors · 2025-04-01
>
> Dear Reviewer iM7M,
>
> Thanks for your positive opinions. We address your concerns as follows. If anything remains unclear, please do not hesitate to contact us.
> >**Q1: Discuss on distribution shift in DFKD**
>
> Thanks for this valuable suggestion! [1-4] focus on different types of distribution shifts when distilling an SL teacher trained on the ID domain using DFKD.
>
> - [1-3] focus on the non-stationary distribution problem, where **shifts between synthetic distribution in different training stages (i.e., epochs) cause the catastrophic forgetting of the student models**. This issue can be mitigated by memory bank-based strategies, where old synthetic is stored and will be replayed in future epochs [1-2] or by using an additional VAE to model the previously observed synthetic samples [3].
> - [4] address the **distribution shift between synthetic data and real training data**. They propose a novel perspective with causal inference to disentangle the student models from the impact of such shifts.
>
> Our work explores the DFKD under NTL teachers. We find that NTL teachers (pretrained on an ID domain and an OOD domain) result in OOD trap effect for DFKD. One key reason is the **ID-to-OOD synthetic distribution shift**, which is caused by the adversarial exploration of DFKD and the NTL's conflict learning target for ID and OOD domains. Such a shift will not occur when distilling an SL teacher trained on the ID domain using DFKD.
>
> >**Q2: Experiments on more backbones and larger datasets.**
>
> The results with different network arch series for teachers and students are shown in Tab. A1-A4. In addition, the results on larger datasets are shown in Tab. A5. From the presented results, the proposed ATEsc is consistently effective in mitigating the OOD trap effect caused by NTL teachers.
>
> **Table A1:** Datasets: ID: CIFAR10 OOD: STL; NetArchs: ResNet-34→VGG-11
> ||IAcc↑|OAcc↑|
> |-|-|-|
> |NTL Teacher|90.8±0.1|10.4±0.0|
> |NAYER|10.3±0.4(-80.5)|9.7±1.3(-0.7)|
> |+CKD|79.1±0.1(-11.7)|40.0±1.1(+29.6)|
> |+ATEsc|68.4±2.1(-22.4)|51.7±2.4(+41.3)|
>
> **Table A2:** Datasets: ID: CIFAR10 OOD: STL; NetArchs: VGG-13→ResNet-18
> ||IAcc↑|OAcc↑|
> |-|-|-|
> |NTL Teacher|92.7±0.1|10.6±0.1|
> |NAYER|18.1±0.3(-74.6)|10.6±0.2(+0.0)|
> |+CKD|30.4±7.2(-62.3)|20.8±5.8(+10.2)|
> |+ATEsc|28.6±4.2(-64.1)|18.7±1.9(+8.1)|
>
> **Table A3:** Datasets: ID: CIFAR10 OOD: Digits; NetArchs: ResNet-34→VGG-11
> ||IAcc↑|OLAcc↓|
> |-|-|-|
> |NTL Teacher|91.1±0.1|100.0±0.0|
> |NAYER|10.6±0.0(-80.5)|100.0±0.0(-0.0)|
> |+CKD|76.0±5.9(-15.1)|5.1±0.4(-94.9)|
> |+ATEsc|74.5±3.9(-16.6)|5.5±0.1(-94.5)|
>
> **Table A4:** Datasets: ID: CIFAR10 OOD: Digits; NetArchs: VGG-13→ResNet-18
> ||IAcc↑|OLAcc↓|
> |-|-|-|
> |NTL Teacher|92.9±0.0|99.5±0.3|
> |NAYER|23.7±0.0(-69.2)|98.0±1.1(-1.5)|
> |+CKD|36.2±19.9(-56.7)|7.8±6.8(-91.7)|
> |+ATEsc|48.1±1.7(-44.8)|1.3±0.9(-98.2)|
>
> **Table A5:** Datasets: ID: CIFAR100 OOD: STL; NetArchs: ResNet-34→ResNet-18
> ||IAcc↑|OLAcc↓|
> |-|-|-|
> |NTL Teacher|68.5±0.6|100.0±0.0|
> |NAYER|5.9±1.2(-62.6)|100.0±0.0(-0.0)|
> |+CKD|48.4±0.9(-20.1)|14.6±12.5(-85.4)|
> |+ATEsc|46.5±1.0(-22.0)|0.0±0.0(-100.0)|
>
> >**Q3: Stability of original CMI/NAYER and DFKD+ATEsc**
>
> The performance of DKFD is sensitive to network initialization, even when distilling SL teachers. The sensitivity can be further increased when distilling NTL teachers. Empirically, our toy experiment in Appendix A offers initial evidence supporting this point (please compare Fig. 9-11).
>
> For the BN layers, we mix the ID data and OOD data in a batch for training NTL teachers, resulting in the well-trained BN layers recording the mixture of statistical information of ID and OOD domains. All SOTA DFKD methods in our experiments rely on BN loss to *train the generator $G$*. Under the joint effect of BN loss and adversarial exploration loss, the synthetic distribution will face an ID-to-OOD synthetic distribution shift when distilling NTL teachers. This further leads to ID-OOD learning task conflicts for students, hindering the student's performance on ID domain. We argue that *distinguishing between ID-like and OOD-like synthetic data during the student's training* can directly solve the ID-OOD conflict issue for students. However, it may not help solve the problem of mismatched statistical information, as the student's training does not directly rely on the BN information from teachers.
>
> >**Q4: The cost of adversarial perturbation**
>
> We use **one-step** untargeted PGD attack in our method, with the perturbation bound $\epsilon=24$. Using NAYER as an example, in each epoch, we only synthesize 800 new samples after $G$'s training. We perform PGD on these samples and save them into ID-like or OOD-like pools. Thus, the additional time is only the time for performing PGD on 800*Epoch samples. As shown in Tab. A5, training NAYER 200 epochs on one RTX 4090 has only an additional 2 mins.
>
> **Table A5:** Datasets: ID: CIFAR10 OOD: STL; NetArchs: ResNet-34→ResNet-18
> ||NAYER|+one-step PGD|
> |-|-|-|
> |Time|1h55m|1h57m|

---

### Decision · Program_Chairs · 2025-05-01

**Decision:**

Accept (poster)

**Comment:**

This paper studies what would happen if one tried to apply data-free knowledge distillation (DFKD) on non-transferable learning (NTL) teachers. The authors propose a new approach named Adversarial Trap Escaping (ATEsc), which benefits DFKD by identifying and filtering out OOD-like synthetic samples.

Reviewers agreed that this paper studies an important problem, and the proposed method is novel. One reviewer pointed out that this paper can be considered as filling a gap in the field of DFKD, and the logic is clear, vivid, and easy to understand. Reviewers also raised some concerns regarding motivation, experiments, technical details, adversarial robustness, etc. The authors have provided very detailed responses to address these concerns.

After the Author-Reviewer discussion period, one reviewer raised a new concern about adaptive backdoor attacks, and the authors provided detailed explanations and justifications that are reasonable.

The authors are strongly encouraged to incorporate the comments and suggestions from reviewers to the camera-ready version, such as explaining why the generator generates samples increasingly closer to the OOD samples, discussing the potential scenarios with adaptive backdoor attacks, etc.